# Mean-Field Sampling for Cooperative Multi-Agent Reinforcement Learning

**Emile Anand**[*]
Georgia Institute of Technology
Atlanta, GA 30308
emile@gatech.edu

**Ishani Karmarkar**
Stanford University
Palo Alto, CA, 94305
ishanik@stanford.edu

**Guannan Qu**
Carnegie Mellon University
Pittsburgh, PA 94035
gqu@andrew.cmu.edu

## Abstract

Designing efficient algorithms for multi-agent reinforcement learning (MARL) is fundamentally challenging because the size of the joint state and action spaces grows exponentially in the number of agents. These difficulties are exacerbated when balancing sequential global decision-making with local agent interactions. In this work, we propose a new algorithm `SUBSAMPLE-MFQ` (**Subsample**-**M**ean-**F**ield-**Q**-learning) and a decentralized randomized policy for a system with $n$ agents. For any $k \leq n$, our algorithm learns a policy for the system in time polynomial in $k$. We prove that this learned policy converges to the optimal policy on the order of $\tilde{O}(1/\sqrt{k})$ as the number of subsampled agents $k$ increases. In particular, this bound is independent of the number of agents $n$.

## 1 Introduction

Reinforcement Learning (RL) has become a popular framework to solve sequential decision making problems in unknown environments and has achieved tremendous success in a wide array of domains such as playing the game of Go [Silver et al., 2016], robotic control [Kober et al., 2013], and autonomous driving [Kiran et al., 2022, Lin et al., 2023a]. A key feature of most real-world systems is their uncertain nature, and thus, RL has emerged as a powerful tool for learning optimal policies for multi-agent systems to operate in unknown environments [Kim and Giannakis, 2017, Zhang et al., 2021, Lin et al., 2024, Anand and Qu, 2024]. While early RL works focused on the single-agent setting, multi-agent RL (MARL) has recently achieved impressive success in many applications, such as coordinating robotic swarms [Preiss et al., 2017, DeWeese and Qu, 2024], real-time bidding [Jin et al., 2018], ride-sharing [Li et al., 2019], and stochastic games [Jin et al., 2020].

Despite growing interest in MARL, extending RL to multi-agent settings poses significant computational challenges due to the *curse of dimensionality*. MARL is fundamentally difficult as agents in the real-world not only interact with the environment but also with each other [Shapley, 1953]: if each of the $n$ agents has a state space $S$ and action space $A$, the global state-action space has size $(|S||A|)^n$, which is exponential in $n$. Thus, many RL algorithms (such as temporal difference learning and tabular Q-learning) require computing and storing an $(|S||A|)^n$-sized Q-table [Sutton et al., 1999, Bertsekas and Tsitsiklis, 1996]. This scalability issue has been observed in a variety of MARL settings [Blondel and Tsitsiklis, 2000, Papadimitriou and Tsitsiklis, 1999, Littman, 1994].

---

[*]Work partially done while a visiting student at Carnegie Mellon University and intern at Cognition AI.

39th Conference on Neural Information Processing Systems (NeurIPS 2025).

An exciting line of work that addresses this intractability is mean-field MARL [Lasry and Lions, 2007, Yang et al., 2018, Gu et al., 2021, 2022, Hu et al., 2023]. The mean-field approach assumes agents are homogeneous in their state-action spaces, enabling their interactions to be approximated by a two-agent setting: here, each agent interacts with a representative "mean agent" which evolves as the empirical distribution of states of all other agents. With these assumptions, mean-field MARL learns an optimal policy with sample complexity $O(n^{|S||A|}|S||A|)$, which is polynomial in the number of agents. However, if $n$ is large, this remains prohibitive even for moderate values of $|S|$ and $|A|$.

Motivated by this problem, in this paper we study the cooperative setting–where agents work collaboratively to maximize a structured global reward–and ask: *Can we design a scalable MARL algorithm for learning an approximately optimal policy in a cooperative multi-agent system?*

**Contributions.** We answer this question affirmatively. Our key contributions are outlined below.

**Subsampling algorithm.** We model the problem as a Markov Decision Process (MDP) with a global agent and $n$ local agents. We propose SUBSAMPLE-MFQ to address the challenge of MARL with a large number of local agents. SUBSAMPLE-MFQ selects $k \leq n$ local agents to learn a deterministic policy $\hat{\pi}_k^{\text{est}}$ by applying mean-field value iteration on the $k$-local-agent subsystem to learn $\hat{Q}_k^{\text{est}}$, which can be viewed as a smaller $Q$ function. SUBSAMPLE-MFQ then deploys a stochastic policy $\pi_k^{\text{est}}$, where the global agent samples $k$ local agents uniformly at each step and uses $\hat{\pi}_k^{\text{est}}$ to determine its action, while each local agent samples $k-1$ other local agents and uses $\hat{\pi}_k^{\text{est}}$ to determine its action.

**Sample complexity and theoretical guarantee.** As the number of local agents increases, the size of $\hat{Q}_k$ scales polynomially with $k$, rather than polynomially with $n$ as in mean-field MARL. Analogously, when the size of the local agent's state space grows, the size of $\hat{Q}_k$ scales exponentially with $k$, rather than exponentially with $n$, as in traditional $Q$-learning. The key analytic technique underlying our results is a novel MDP sampling result. This result shows that the performance gap between $\pi_k^{\text{est}}$ and the optimal policy $\pi^*$ is at most $\tilde{O}(1/\sqrt{k})$, with high probability. The choice of $k$ reveals a fundamental trade-off between the size of the $Q$-table and the optimality of $\pi_k^{\text{est}}$. For example, if $k$ is set to $O(\log n)$, SUBSAMPLE-MFQ is the first centralized MARL algorithm to achieve a *polylogarithmic* run-time in $n$, representing an exponential speedup over the previously best-known polytime mean-field MARL methods, while maintaining a decaying optimality gap as $n$ gets large,

While our results are theoretical in nature, we hope SUBSAMPLE-MFQ will further exploration of sampling in Markov games, and potentially inspire new practical multi-agent algorithms.

**Related work.** MARL has a rich history, starting with early works on Markov games [Littman, 1994, Sutton et al., 1999], which are a multi-agent extension of MDPs. MARL has since been actively studied [Zhang et al., 2021] in a broad range of settings. MARL is most similar to the category of "succinctly described" MDPs [Blondel and Tsitsiklis, 2000], where the state/action space is a product space formed by the individual state/action spaces of multiple agents, and where the agents interact to maximize an objective. A recent line of research constrains the problem to sparse networked instances to enforce local interactions between agents [Qu et al., 2020a, Lin et al., 2020, Mondal et al., 2022]. In this formulation, the agents correspond to vertices on a graph who only interact with nearby agents. By exploiting Gamarnik's correlation decay property from combinatorial optimization [Gamarnik et al., 2009], they overcome the curse of dimensionality by simplifying the problem to only search over the policy space derived from the truncated graph to learn approximately optimal solutions. However, as the underlying network structure becomes dense with many local interactions, the neighborhood of each agent gets large, and these algorithms become intractable.

*Mean-Field RL.* Under assumptions of homogeneity in the state/action spaces of the agents, the problem of densely networked multi-agent RL was studied by Yang et al. [2018], Gu et al. [2021], who approximated the solution in polynomial time with a mean-field approach where the approximation error scales in $O(1/\sqrt{n})$. In contrast, our work achieves *subpolynomial* runtimes by directly sampling from this mean-field distribution. Cui and Koeppl [2022] introduce heterogeneity to mean-field MARL by modeling non-uniform interactions through graphons; however, these methods crucially assume the existence of graphon sequences that converge in cut-norm to the finite graph. In the cooperative setting, Subramanian et al. [2022], Cui et al. [2023] studies a mean-field setting with $q$ types of homogeneous agents; however, their learned policy does not provably converge to the optimum.

*Other related works.* Our work is related to factored MDPs, where there is a global action affecting every agent; however, in our case, each agent has its own action [Min et al., 2023, Lauer and

Riedmiller, 2000]. Jin et al. [2020] reduces the dependence of the product action space to an additive dependence with V-learning. Our work *further* reduces the complexity of the joint state space, which has not been previously accomplished. We add to the growing literature on the Centralized Training with Decentralized Execution regime [Zhou et al., 2023], as our algorithm learns a provably approximately optimal policy using centralized information, but makes decisions using only local information during execution. Finally, one can efficiently approximate the $Q$-table through function approximation [Jin et al., 2021]. However, achieving theoretical bounds on the performance loss due to function approximation is intractable without strong assumptions such as linear Bellman completeness or low Bellman-Eluder dimension [Golowich and Moitra, 2024]. While our work primarily studies the finite tabular setting, we extend it to non-tabular linear MDPs in Appendix J.

## 2 Preliminaries

**Notation.** For $k, n \in \mathbb{N}$ where $k \leq n$, let $\binom{[n]}{k}$ denote the set of $k$-sized subsets of $[n] = \{1, \ldots, n\}$. For any vector $z \in \mathbb{R}^d$, let $\|z\|_1$ and $\|z\|_\infty$ denote the standard $\ell_1$ and $\ell_\infty$ norms of $z$ respectively. Let $\|\mathbf{A}\|_1$ denote the matrix $\ell_1$-norm of $\mathbf{A} \in \mathbb{R}^{n \times m}$. Given variables $s_1, \ldots, s_n, s_\Delta := \{s_i : i \in \Delta\}$ for $\Delta \subseteq [n]$. We use $\tilde{O}(\cdot)$ to suppress polylogarithmic factors in all problem parameters except $n$. For a discrete measurable space $(\mathcal{X}, \mathcal{F})$, the total variation distance between probability measures $\rho_1, \rho_2$ is given by $\mathrm{TV}(\rho_1, \rho_2) = \frac{1}{2} \sum_{x \in \mathcal{X}} |\rho_1(x) - \rho_2(x)|$. Next, $x \sim \mathcal{D}(\cdot)$ denotes that $x$ is a random element sampled from a distribution $\mathcal{D}$, and we denote that $x$ is a random sample from the uniform distribution over a finite set $\Omega$ by $x \sim \mathcal{U}(\Omega)$. We include a detailed notation table in Table 1.

### 2.1 Problem formulation

We consider a system of $n + 1$ agents, where agent $g$ is a "global decision making agent" and the remaining $n$ agents, denoted by $[n]$, are "local agents." At time $t$, the agents are in state $s(t) = (s_g(t), s_1(t), ..., s_n(t)) \in \mathcal{S} := \mathcal{S}_g \times \mathcal{S}_l^n$, where $s_g(t) \in \mathcal{S}_g$ denotes the global agent's state, and for each $i \in [n]$, $s_i(t) \in \mathcal{S}_l$ denotes the state of the $i$'th local agent. The agents cooperatively select actions $a(t) = (a_g(t), a_1(t), ..., a_n(t)) \in \mathcal{A} := \mathcal{A}_g \times \mathcal{A}_l^n$, where $a_g(t) \in \mathcal{A}_g$ denotes the global agent's action and $a_i(t) \in \mathcal{A}_l$ denotes the $i$'th local agent's action. At time $t + 1$, the next state for all the agents is independently generated by stochastic transition kernels $P_g : \mathcal{S}_g \times \mathcal{S}_g \times \mathcal{A}_g \to \Delta(\mathcal{S}_g)$ and $P_l : \mathcal{S}_l \times \mathcal{S}_l \times \mathcal{S}_g \times \mathcal{A}_l \to \Delta(\mathcal{S}_l)$ as follows:

$$s_g(t + 1) \sim P_g(\cdot|s_g(t), a_g(t)), \quad s_i(t + 1) \sim P_l(\cdot|s_i(t), s_g(t), a_i(t)), \forall i \in [n]. \tag{1}$$

The system then collects a structured stage reward $r(s(t), a(t))$ where the reward $r : \mathcal{S} \times \mathcal{A} \to \mathbb{R}$ depends on $s(t)$ and $a(t)$ through Equation (2), and where $r_g$ and $r_l$ is typically application specific.

$$r(s, a) = \underbrace{r_g(s_g, a_g)}_{\text{global component}} + \frac{1}{n} \sum_{i \in [n]} \underbrace{r_l(s_i, s_g, a_i)}_{\text{local component}} \tag{2}$$

A policy $\pi : \mathcal{S} \to \mathcal{P}(\mathcal{A})$ maps from states to distributions of actions such that $a \sim \pi(\cdot|s)$. Given $\gamma \in (0, 1)$, we seek to learn a policy $\pi$ that maximizes the ($\gamma$-discounted) value for each $s \in S$:

$$V^\pi(s) = \mathbb{E}_{a(t) \sim \pi(\cdot|s)} \left[ \sum_{t=0}^\infty \gamma^t r(s(t), a(t))|s(0) = s \right]. \tag{3}$$

The cardinality of the search space simplex for the optimal policy is $|\mathcal{S}_g||\mathcal{S}_l|^n|\mathcal{A}_g||\mathcal{A}_l|^n$, which is exponential in the number of agents, underscoring the need for efficient approximation algorithms.

To efficiently learn policies that maximize the objective, we make the following standard assumptions:

**Assumption 2.1** (Finite state/action spaces). We assume that the state and action spaces of all the agents in the MARL game are finite: $|\mathcal{S}_l|, |\mathcal{S}_g|, |\mathcal{A}_g|, |\mathcal{A}_l| < \infty$. Appendix J of the supplementary material relaxes this assumption to the non-tabular setting with infinite continuous sets.

**Assumption 2.2** (Bounded rewards). The components of the reward function are bounded. Specifically, $\|r_g(\cdot, \cdot)\|_\infty \leq \tilde{r}_g$, and $\|r_l(\cdot, \cdot, \cdot)\|_\infty \leq \tilde{r}_l$. This implies $\|r(\cdot, \cdot)\|_\infty \leq \tilde{r}_g + \tilde{r}_l := \tilde{r}$.

**Definition 2.3** ($\epsilon$-optimal policy). Given a policy simplex $\Pi$, $\pi \in \Pi$ is an $\epsilon$-optimal policy if $V^\pi(s) \geq \sup_{\pi^* \in \Pi} V^{\pi^*}(s) - \epsilon$.

**Motivating examples.** Below we give examples of two cooperative MARL settings which are naturally modeled by our setting. Our experiments in Appendix B reveal a monotonic improvement in the learned policies as $k \to n$, while providing a substantial speedup over mean-field $Q$-learning[2].

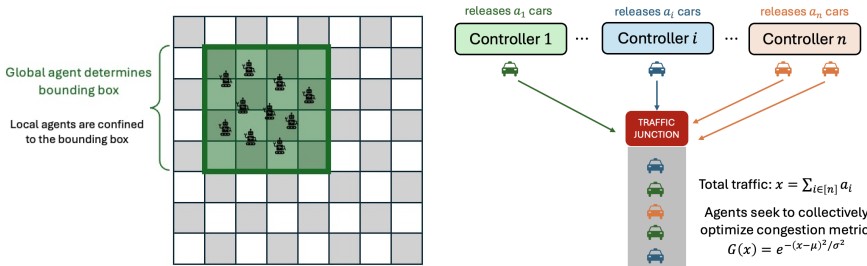

Figure 1: Bounded exploration in warehouse accidents, and traffic congestions with Gaussian squeeze.

- **Gaussian squeeze:** In this task, $n$ homogeneous agents determine individual actions $a_i$ to jointly maximize the objective $r(x) = xe^{-(x-\mu)^2/\sigma^2}$, where $x = \sum_{i=1}^{n} a_i$, and $\mu$ and $\sigma$ are the pre-defined mean and variance of the system. In scenarios of traffic congestion, each agent $i \in [n]$ is a controller trying to send $a_i$ vehicles into the main road, where controllers coordinate with each other to avoid congestion, hence avoiding either over-use or under-use, thereby contributing to the entire system. This GS problem is previously studied in Yang et al. [2018], and serves as an ablation study on the impact of subsampling for MARL.

- **Constrained exploration:** Consider an $M \times M$ grid. Each agent's state is a coordinate in $[M] \times [M]$. The state represents the *center of a $d \times d$ box* where the global agent constrains the local agents' movements. Initially, all agents are in the same location. At each time-step, the local agents take actions $a_i(t) \in \mathbb{R}^2$ (e.g., up, down, left, right) to transition between states and collect rewards. The transition kernel ensures that local agents remain within the $d \times d$ box dictated by the global agent, using knowledge of $a_i(t), s_g(t)$, and $s_i(t)$. In warehouse settings where shelves have collapsed, creating hazardous or inaccessible areas, we want agents to clean these areas. However, exploration in these regions may be challenging due to physical constraints or safety concerns, causing exploration in these regions to be *disincentivized* from the local agents' perspectives. Through an appropriate design of the reward and transition functions, the global agent could guide the local agents to focus on specific $d \times d$ grids, allowing efficient cleanup while minimizing risk.

**Capturing heterogeneity.** Following Mondal et al. [2022], Xu and Klabjan [2023], our model can capture heterogeneity in the local agents by modeling agent types as part of the state: to do this, we assign a type $\varepsilon_i \in \mathcal{E}$ to each local agent by letting $\mathcal{S}_l = \mathcal{E} \times \mathcal{S}'_l$, where $\mathcal{E}$ is a set that enumerates possible types that are treated as a fixed part of the agent's state, and $\mathcal{S}'_l$ is the latent state space of any local agent. The transition and reward functions can vary depending on the agent's type. The global agent can provide unique signals to local agents of each type by letting $s_g \in \mathcal{S}_g$ and $a_g \in \mathcal{A}_g$ denote a state/action vector where each element corresponds to a type.

## 3 Algorithmic Approach: Subsampled Value Iteration

**Q-learning.** Our starting point is the classic Q-learning framework [Watkins and Dayan, 1992] for offline-RL, which seeks to learn the $Q$-function $Q : \mathcal{S} \times \mathcal{A} \to \mathbb{R}$. For any policy $\pi$, $Q^\pi(s, a) = \mathbb{E}^\pi[\sum_{t=0}^{\infty} \gamma^t r(s(t), a(t))|s(0)=s, a(0)=a]$. Initially, $Q^0(s,a)=0$, for all $(s,a) \in \mathcal{S} \times \mathcal{A}$. Then, for all $t \in [T]$, it is updated as $Q^{t+1}(s,a) \leftarrow \mathcal{T}Q^t(s,a)$, where the *Bellman operator* $\mathcal{T}$ is

$$\mathcal{T}Q^t(s,a) = r(s,a) + \gamma \mathbb{E}_{s'_g \sim P_g(\cdot|s_g,a), s'_i \sim P_l(\cdot|s_i,s_g,a_i), \forall i \in [n]} \max_{a' \in \mathcal{A}} Q^t(s',a'). \quad (4)$$

It is well known that $\mathcal{T}$ is $\gamma$-contractive, ensuring that the above updates converge to a unique $Q^*$ such that $\mathcal{T}Q^* = Q^*$. The optimal policy $\pi^* : \mathcal{S} \to \mathcal{A}$ can then be computed greedily as

---

[2]We provide supporting code for the algorithm and experiments in `https://github.com/emiletimothy/Mean-Field-Subsample-Q-Learning`

$\pi^*(s) = \arg\max_{a \in \mathcal{A}} Q^*(s, a)$. However, the update complexity for the $Q$-function is $O(|\mathcal{S}||\mathcal{A}|) = O(|\mathcal{S}_g||\mathcal{S}_l|^n|\mathcal{A}_g||\mathcal{A}_l|^n)$, which is exponential in the number of local agents increases.

**Mean-field transformation.** To address this, mean-field MARL [Yang et al., 2018] (under homogeneity assumptions) studies the empirical distribution function $F_{z_{[n]}} : \mathcal{Z}_l \to \mathbb{R}$, for $\mathcal{Z}_l := \mathcal{S}_l \times \mathcal{A}_l$:

$$F_{z_{[n]}}(z) := \frac{1}{n} \sum_{i=1}^n \mathbf{1}\{s_i = z_s, a_i = z_a\}, \quad \forall z := (z_s, z_a) \in \mathcal{Z}_l := \mathcal{S}_l \times \mathcal{A}_l. \tag{5}$$

Let $\mu_n(\mathcal{Z}_l) = \{\frac{b}{n}|b \in \{0, \dots, n\}\}^{|\mathcal{Z}_l|}$ be the space of $|\mathcal{Z}_l|$-sized vectors (or $|\mathcal{S}_l| \times |\mathcal{A}_l|$-sized tables), where each entry is in $\{0, 1/n, 2/n, \dots, 1\}$. Intuitively, $\mu_n(\mathcal{Z}_l)$ is a discrete distribution over $(\mathcal{S}_l, \mathcal{A}_l)$ where each probability assigned is a multiple of $1/n$. Then, $F_{z_{[n]}} \in \mu_n(\mathcal{Z}_l)$ indicates the proportion of agents in each state-action pair.

As in Q-learning, in mean-field Q-learning initially $\hat{Q}^0(s_g, s_1, a_g, a_1, F_{z_{[n]\setminus 1}}) = 0$. At each time $t \in [T]$, we update $\hat{Q}$ as $\hat{Q}^{t+1} = \hat{\mathcal{T}}\hat{Q}^t$, where $\hat{\mathcal{T}}$ is the Bellman operator in *distribution space*:

$$\hat{\mathcal{T}}\hat{Q}^t(s_g, s_1, a_g, a_1, F_{z_{[n]\setminus 1}}) = r(s, a) + \gamma \mathbb{E}_{\substack{s_g' \sim P_g(\cdot|s_g, a_g) \\ s_i' \sim P_l(\cdot|s_i, s_g, a_i) \\ \forall i \in [n]}} \max_{\substack{(a_g', a_1', a_{[n]\setminus 1}') \\ \in \mathcal{A}_g \times \mathcal{A}_l \times \mathcal{A}_l^{n-1}}} \hat{Q}^t(s_g', s_1', a_g', a_1', F_{z_{[n]\setminus 1}'})$$

Since the $Q$-function is permutation-invariant in the homogeneous local agents, one sees that for each $t \geq 0$, $Q^t(s_g, s_{[n]}, a_g, a_{[n]}) = \hat{Q}^t(s_g, s_1, a_g, a_1, F_{z_{[n]\setminus 1}})$. In other words, mean-field Q-learning implements the same updates as standard Q-learning. However, the update complexity of $\hat{Q}$ is only $O(|\mathcal{S}_g||\mathcal{A}_g||\mathcal{Z}_l|n^{|\mathcal{Z}_l|})$, which scales polynomially in $n$ but exponentially in $|\mathcal{Z}_l|$.

**Remark 3.1.** Existing methods use sample complexity $\tilde{O}(\min\{|\mathcal{S}_g||\mathcal{A}_g||\mathcal{Z}_l|n, |\mathcal{S}_g||\mathcal{A}_g||\mathcal{Z}_l|n^{|\mathcal{Z}_l|}\})$: one uses $Q$-learning if $|\mathcal{Z}_l|^{n-1} < n^{|\mathcal{Z}_l|}$, and mean-field value iteration otherwise. In each regime, as $n$ scales, the update complexity becomes computationally infeasible.

To further reduce the update complexity, we propose `SUBSAMPLE-MFQ` to overcome the polynomial (in $n$) sample complexity of mean-field $Q$-learning and the exponential (in $n$) sample complexity of traditional $Q$-learning. We begin by motivating the intuition behind our algorithms.

### 3.1 Overview of approach.

**Offline Planning: Algorithm 1.** First, the global agent randomly samples a subset of local agents $\Delta \subseteq [n]$ such that $|\Delta| = k$, for $k \leq n$. It then ignores all other local agents $[n] \setminus \Delta$, and performs value iteration (using $m$ samples to update the $Q$-function in each iterate) to approximately learn the $Q$-function $\hat{Q}_{k,m}^{\text{est}}$ and policy $\hat{\pi}_{k,m}^{\text{est}}$ for this surrogate subsystem of $k$ local agents. We denote the surrogate reward gained by this subsystem at each time step by $r_\Delta : \mathcal{S} \times \mathcal{A} \to \mathbb{R}$, where

$$r_\Delta(s, a) = r_g(s_g, a_g) + \frac{1}{|\Delta|} \sum_{i \in \Delta} r_l(s_g, s_i, a_i). \tag{6}$$

In Theorem E.3, we show that $\|\hat{Q}_{k,m}^{\text{est}} - Q^*\|_\infty$ is Lipschitz continuous with respect to the TV-distance between the state/action pairs of the subsampled agents and the full set of $n$ agents. Equipped with this approximation guarantee, we show how to construct an approximately optimal policy on the *full* system on $n$ agents. In general, converting this policy on $k$ local agents to a policy on the full $n$-agent system without sacrificing error guarantees can be intractable, and there is a line on works on centralized-training decentralized-execution (CTDE) [Xu and Klabjan, 2023, Zhou et al., 2023] which shows that designing performant decentralized policies can be highly non-trivial.

**Online Execution: Algorithm 2.** In order to circumvent this obstacle and convert the optimality of each agent's action in the $k$ local-agent subsystem to an approximate optimality guarantee on the full $n$-agent system, we propose a randomized policy $\pi_{k,m}^{\text{est}}$, where the global agent samples $\Delta \in \binom{[n]}{k}$ at each time-step to derive an action $a_g \leftarrow \hat{\pi}_{k,m}^{\text{est}}(s_g, s_\Delta)$, and where each local $i$ agent samples $k-1$ other local agents $\Delta_i$ to derive an action $a_i \leftarrow \hat{\pi}_{k,m}^{\text{est}}(s_g, s_i, s_{\Delta_i})$. Finally, Theorem 4.4 shows that the policy $\pi_{k,m}^{\text{est}}$ converges to the optimal policy $\pi^*$ as $k \to n$ with rate $\tilde{O}(\frac{1}{\sqrt{k}})$.

## 3.2 Algorithm description.

We now formally describe the algorithms. Before describing Algorithm 1 (SUBSAMPLE-MFQ: Learning) and Algorithm 2 (SUBSAMPLE-MFQ: Execution) in detail, it will be helpful to first define the *empirical distribution function*:

**Definition 3.2** (Empirical Distribution Function). For any population $(z_1, \ldots, z_n) \in \mathcal{Z}_l^n$, where $\mathcal{Z}_l := \mathcal{S}_l \times \mathcal{A}_l$, define the empirical distribution function $F_{z_\Delta} : \mathcal{Z}_l \to \mathbb{R}_+$ for all $z := (z_s, z_a) \in \mathcal{S}_l \times \mathcal{A}_l$ and for all $\Delta \in \binom{[n]}{k}$ by $F_{z_\Delta}(x) := \frac{1}{k} \sum_{i \in \Delta} \mathbf{1}\{s_i = z_s, a_i = z_a\}$.

Let $\mu_k(\mathcal{Z}_l) := \left\{ \frac{b}{k} | b \in \{0, \ldots, k\} \right\}^{|\mathcal{Z}_l|}$ be the space of $|\mathcal{Z}_l|$-sized vectors (or $|\mathcal{S}_l| \times |\mathcal{A}_l|$-sized tables) where each entry is in $\{0, 1/k, 2/k, \ldots, 1\}$. Intuitively, $\mu_k(\mathcal{Z}_l)$ is a discrete distribution over $(\mathcal{S}_l, \mathcal{A}_l)$ where each probability assigned is a multiple of $1/k$. Here, $F_{z_\Delta} \in \mu_k(\mathcal{Z}_l)$ indicates the proportion of agents (in the $k$-local-agent subsystem) at each state/action pair.

**Algorithm 1** (Offline learning). Let $m \in \mathbb{N}$ denote the sample size for the learning algorithm with sampling parameter $k \leq n$. As in Remark 3.1, when $|\mathcal{Z}_l|^k \leq |\mathcal{Z}_l| k^{|\mathcal{Z}_l|}$, the algorithm uses traditional value-iteration, and when $|\mathcal{Z}_l|^k > |\mathcal{Z}_l| k^{|\mathcal{Z}_l|}$, it uses mean-field value iteration. We formally describe the procedure for each regime below:

**Regime with Large State/Action Space:** When $|\mathcal{Z}_l|^k \leq |\mathcal{Z}_l| k^{|\mathcal{Z}_l|}$, we iteratively learn the optimal $Q$-function for a subsystem with $k$-local agents denoted by $\hat{Q}_{k,m}^t : \mathcal{S}_g \times \mathcal{S}_l^k \times \mathcal{A}_g \times \mathcal{A}_l^k \to \mathbb{R}$, which is initialized to 0. At time $t$, we update

$$\hat{Q}_{k,m}^{t+1}(s_g, s_\Delta, a_g, a_\Delta) = \tilde{\mathcal{T}}_{k,m} \hat{Q}_{k,m}^t(s_g, s_\Delta, a_g, a_\Delta), \tag{7}$$

where $\tilde{\mathcal{T}}_{k,m}$ is the *empirically adapted Bellman operator* in Equation (8). Since the system is unknown, $\tilde{\mathcal{T}}_{k,m}$ cannot directly perform the Bellman update, so it instead uses $m$ random samples $s_g^j \sim P_g(\cdot|s_g, a_g)$ and $s_i^j \sim P_l(\cdot|s_i, s_g, a_i)$ for each $j \in [m]$, $i \in \Delta$ to approximate the system:

$$\tilde{\mathcal{T}}_{k,m} \hat{Q}_{k,m}^t(s_g, s_\Delta, a_g, a_\Delta) = r_\Delta(s, a) + \frac{\gamma}{m} \sum_{j \in [m]} \max_{a_g' \in \mathcal{A}_g, a_\Delta' \in \mathcal{A}_l^k} \hat{Q}_{k,m}^t(s_g^j, s_\Delta^j, a_g', a_\Delta').$$

Since $\tilde{\mathcal{T}}_{k,m}$ is $\gamma$-contractive, Algorithm 1 applies value iteration with $\tilde{\mathcal{T}}$ until $\hat{Q}_{k,m}$ converges to a fixed point satisfying $\tilde{\mathcal{T}}_{k,m} \hat{Q}_{k,m}^{\text{est}} = \hat{Q}_{k,m}^{\text{est}}$, yielding a deterministic policy $\hat{\pi}_{k,m}^{\text{est}}(s_g, s_\Delta)$ where

$$\hat{\pi}_{k,m}^{\text{est}}(s_g, s_\Delta) = \underset{a_g \in \mathcal{A}_g, a_\Delta \in \mathcal{A}_l^k}{\arg \max} \hat{Q}_{k,m}^{\text{est}}(s_g, s_\Delta, a_g, a_\Delta). \tag{8}$$

**Regime with Large Number of Agents:** When $|\mathcal{Z}_l|^k > |\mathcal{Z}_l| k^{|\mathcal{Z}_l|}$, we learn the optimal mean-field $Q$-function for a $k$ local agent system, denoted by $\hat{Q}_{k,m}^t : \mathcal{S}_g \times \mathcal{S}_l \times \mu_{k-1}(\mathcal{Z}_l) \times \mathcal{A}_g \times \mathcal{A}_l \to \mathbb{R}$, which is initialized to 0. At time $t$, we update

$$\hat{Q}_{k,m}^{t+1}(s_g, s_1, F_{z_{\tilde{\Delta}}}, a_1, a_g) = \hat{\mathcal{T}}_{k,m} \hat{Q}_{k,m}^t(s_g, s_1, F_{z_{\tilde{\Delta}}}, a_1, a_g), \tag{9}$$

where $\hat{\mathcal{T}}_{k,m}$ is the *empirically adapted mean-field Bellman operator* in Equation (10). Similarly, as the system is unknown, $\hat{\mathcal{T}}_{k,m}$ cannot directly perform the Bellman update and instead uses $m$ random samples $s_g^j \sim P_g(\cdot|s_g, a_g)$ and $s_i^j \sim P_l(\cdot|s_i, s_g, a_i), \forall j \in [m]$, $i \in \Delta$ to approximate the system:

$$\hat{\mathcal{T}}_{k,m} \hat{Q}_{k,m}^t(s_g, s_1, F_{z_{\tilde{\Delta}}}, a_1, a_g) = r_\Delta(s, a) + \frac{\gamma}{m} \sum_{j \in [m]} \max_{\substack{a_g' \in \mathcal{A}_g, a_1' \in \mathcal{A}_l, \\ F_{a_{\tilde{\Delta}}'} \in \mu_{k-1}(\mathcal{A}_l)}} \hat{Q}_{k,m}^t(s_g^j, s_1^j, F_{s_{\tilde{\Delta}}^j, a_{\tilde{\Delta}}'}, a_1', a_g') \tag{10}$$

Since $\hat{Q}_{k,m}^t$ depends on $s_\Delta$ and $a_\Delta$ through $F_{z_\Delta}$, $\hat{\mathcal{T}}_{k,m}$ is also $\gamma$-contractive. So, Algorithm 1 applies value iteration with $\hat{\mathcal{T}}$ until $\hat{Q}_{k,m}$ converges to a fixed point satisfying $\hat{\mathcal{T}}_{k,m} \hat{Q}_{k,m}^{\text{est}} = \hat{Q}_{k,m}^{\text{est}}$, yielding a deterministic policy $\hat{\pi}_{k,m}^{\text{est}}(s_g, s_1, F_{s_{\tilde{\Delta}}})$:

$$\hat{\pi}_{k,m}^{\text{est}}(s_g, s_1, F_{s_{\tilde{\Delta}}}) = \underset{a_g \in \mathcal{A}_g, a_1 \in \mathcal{A}_l, F_{a_{\tilde{\Delta}}} \in \mu_{k-1}(\mathcal{A}_l)}{\arg \max} \hat{Q}_{k,m}^{\text{est}}(s_g, s_1, F_{z_{\tilde{\Delta}}}, a_1, a_g)$$

For $\hat{\pi}_{k,m}^{\text{est}}(s_g, s_i, s_{\Delta \setminus i}) = a_g^*, a_i^*, a_{\Delta \setminus i}^*$, let $[\hat{\pi}_{k,m}^{\text{est}}(s_g, s_\Delta)]_g = a_g^*$ and $[\hat{\pi}_{k,m}^{\text{est}}(s_g, s_i, s_{\Delta \setminus i})]_l = a_i^*$.

**Algorithm 1** SUB-SAMPLE-MFQ: Learning

---

**Require:** A multi-agent system as in Section 2, number of iterations $T$, sampling parameters $k \in [n]$, $m \in \mathbb{N}$, and discount factor $\gamma \in (0, 1)$.

1: Let $\Delta = \{1, \ldots, k\}$, $\tilde{\Delta} = \{2, \ldots, k\}$, and $\mu_{k-1}(\mathcal{Z}_l) = \{\frac{b}{k-1} : b \in \{0, 1, \ldots, k-1\}\}^{|\mathcal{S}_l| \times |\mathcal{A}_l|}$.
2: **if** $|\mathcal{Z}_l|^k \leq |\mathcal{Z}_l| k^{|\mathcal{Z}_l|}$ **then**
3:     *// Sub-sampled version of standard Q-learning*
4:     Initialize $\hat{Q}^0_{k,m}(s_g, s_\Delta, a_g, a_\Delta) = 0, \forall (s_g, s_\Delta, a_g, a_\Delta)$.
5:     **for** $t = 1$ to $T$ **do**
6:         **for** $(s_g, s_\Delta, a_\Delta, a_g) \in \mathcal{S}_g \times \mathcal{S}^k_l \times \mathcal{A}_g \times \mathcal{A}^k_l$ **do**
7:             $\hat{Q}^{t+1}_{k,m}(s_g, s_\Delta, a_g, a_\Delta) = \tilde{\mathcal{T}}_{k,m} \hat{Q}^t_{k,m}(s_g, s_\Delta, a_g, a_\Delta)$
8:     Let the greedy policy be $\hat{\pi}^{\text{est}}_{k,m}(s_g, s_\Delta) := \arg\max_{a_g \in \mathcal{A}_g, a_\Delta \in \mathcal{A}^k_l} \hat{Q}^T_{k,m}(s_g, s_\Delta, a_g, a_\Delta)$.
9: **else**
10:     *// Sub-sampled version of mean-field Q-learning*
11:     Initialize $\hat{Q}^0_{k,m}(s_g, s_1, F_{z_{\tilde{\Delta}}}, a_1, a_g) = 0, \forall (s_g, s_\Delta, a_\Delta, a_g)$.
12:     **for** $t = 1$ to $T$ **do**
13:         **for** $(s_g, s_1, F_{z_{\tilde{\Delta}}}, a_1, a_g) \in \mathcal{S}_g \times \mathcal{S}_l \times \mu_{k-1}(\mathcal{Z}_l) \times \mathcal{A}_l \times \mathcal{A}_g$ **do**
14:             $\hat{Q}^{t+1}_{k,m}(s_g, s_1, F_{z_{\tilde{\Delta}}}, a_1, a_g) = \hat{\mathcal{T}}_{k,m} \hat{Q}^t_{k,m}(s_g, s_1, F_{z_{\tilde{\Delta}}}, a_1, a_g)$
15:     Let the greedy policy be $\hat{\pi}^{\text{est}}_{k,m}(s_g, s_i, F_{s_{\tilde{\Delta}}}) := \arg\max_{\substack{a_g \in \mathcal{A}_g, a_i \in \mathcal{A}_l, \\ F_{a_{\tilde{\Delta}}} \in \mu_{k-1}(\mathcal{A}_l)}} \hat{Q}^T_{k,m}(s_g, s_1, F_{z_{\tilde{\Delta}}}, a_1, a_g)$.

---

**Remark 3.3.** Since mean-field $Q$-learning maintains the same updates as standard $Q$-learning (from Lemma C.19 which follows by noting that the $Q$-function is permutation-invariant in the homogeneous local agents), the deterministic policies $\hat{\pi}^{\text{est}}_{k,m}(s_g, s_\Delta)$ and $\hat{\pi}^{\text{est}}_{k,m}(s_g, s_1, F_{s_{\tilde{\Delta}}})$ are equivalent.

**Algorithm 2** (Online implementation). In Algorithm 2, (SUBSAMPLE-MFQ: Execution) the global agent samples local agents $\Delta(t) \sim \mathcal{U}\binom{[n]}{k}$ at each step to derive action $a_g(t) = [\hat{\pi}^{\text{est}}_{k,m}(s_g, s_\Delta(t))]_g$, and each local agent $i$ samples other local agents $\Delta_i(t) \sim \mathcal{U}\binom{[n] \setminus i}{k-1}$ to derive action $a_i(t) = [\hat{\pi}^{\text{est}}_{k,m}(s_g, s_i, s_\Delta(t))]_l$. The system then incurs a reward $r(s, a)$. This procedure of first sampling agents and then applying $\hat{\pi}^{\text{est}}_{k,m}$ is denoted by a stochastic policy $\pi^{\text{est}}_{k,m}(a|s)$, where $\pi^{\text{est}}_{k,m}(a_g|s)$ is the global agent's action distribution and $\pi^{\text{est}}_{k,m}(a_l|s)$ is the local agent's action distribution:

$$\pi^{\text{est}}_{k,m}(a_g|s) = \frac{1}{\binom{n}{k}} \sum_{\Delta \in \binom{[n]}{k}} \mathbf{1}(\hat{\pi}^{\text{est}}_{k,m}(s_g, s_\Delta) = a) \tag{11}$$

$$\pi^{\text{est}}_{k,m}(a_i|s) = \frac{1}{\binom{n-1}{k-1}} \sum_{\tilde{\Delta} \in \binom{[n] \setminus i}{k-1}} \mathbf{1}(\pi^{\text{est}}_{k,m}(s_g, s_i, F_{s_{\tilde{\Delta}}}) = a_i) \tag{12}$$

The agents then transition to their next states.

## 4 Theoretical Guarantees and Analysis Approach

We now show the value of the expected discounted cumulative reward produced by $\pi^{\text{est}}_{k,m}$ is approximately optimal, where the optimality gap decays as $k \to n$ and $m$ grows.

**Bellman noise.** We introduce the notion of Bellman noise, which is used in the main theorem. Note that $\hat{\mathcal{T}}_{k,m}$ is an unbiased estimator of the adapted Bellman operator $\hat{\mathcal{T}}_k$,

$$\hat{\mathcal{T}}_k \hat{Q}_k(s_g, s_\Delta, a_g, a_\Delta) = r_\Delta(s, a) + \gamma \mathbb{E}_{\substack{s'_g \sim P_g(\cdot|s_g, a_g), \\ s'_i \sim P_l(\cdot|s_i, s_g, a_i), \forall i \in \Delta}} \max_{a'_g \in \mathcal{A}_g, a'_\Delta \in \mathcal{A}^k_l} \hat{Q}_k(s'_g, s'_\Delta, a'_g, a'_\Delta). \tag{13}$$

Let $\hat{Q}^0_k(s_g, s_\Delta, a_g, a_\Delta) = 0$. For $t > 0$, let $\hat{Q}^{t+1}_k = \hat{\mathcal{T}}_k \hat{Q}^t_k$, where $\hat{\mathcal{T}}_k$ is defined for $k \leq n$ in Equation (13). Then, $\hat{\mathcal{T}}_k$ is also a $\gamma$-contraction with fixed-point $\hat{Q}^*_k$. By the law of large numbers, $\lim_{m \to \infty} \hat{\mathcal{T}}_{k,m} = \hat{\mathcal{T}}_k$ and $\|\hat{Q}^{\text{est}}_{k,m} - \hat{Q}^*_k\|_\infty \to 0$ as $m \to \infty$. For finite $m$, $\epsilon_{k,m} := \|\hat{Q}^{\text{est}}_{k,m} - \hat{Q}^*_k\|_\infty$

**Algorithm 2** SUBSAMPLE-MFQ: Execution

---

**Require:** Parameter $T'$ for the number of iterations for the decision-making sequence. Sampling parameter $k \in [n], m \in \mathbb{N}$. Discount factor $\gamma$. Policy $\hat{\pi}_{k,m}^{\text{est}}(s_g, F_{s_\Delta})$.
1: Learn $\hat{\pi}_{k,m}^{\text{est}}$ from Algorithm 1.
2: Sample $(s_g(0), s_{[n]}(0)) \sim s_0$, where $s_0$ is a distribution on the initial global state $(s_g, s_{[n]})$
3: Initialize the total reward $R_0 = 0$.
4: **Policy** $\pi_{k,m}^{\text{est}}(s)$ is defined as follows:
5: **for** $t = 0$ to $T'$ **do**
6:    Choose $\Delta$ uniformly at random from $\binom{[n]}{k}$ and let $a_g(t) = [\hat{\pi}_{k,m}^{\text{est}}(s_g(t), s_\Delta(t))]_g$.
7:    **for** $i = 1$ to $n$ **do**
8:       Choose $\Delta_i$ uniformly at random from $\binom{[n] \setminus i}{k-1}$ and let $a_i(t) = [\hat{\pi}_{k,m}^{\text{est}}(s_g(t), s_i(t), s_{\Delta_i}(t))]_l$.
9:    Let $s_g(t+1) \sim P_g(\cdot | s_g(t), a_g(t))$.
10:    Let $s_i(t+1) \sim P_l(\cdot | s_i(t), s_g(t), a_i(t)), \forall i \in [n]$.
11:    $R_{t+1} = R_t + \gamma^t \cdot r(s, a)$

---

is the well-studied Bellman noise. To compare the performance between $\pi^*$ and $\pi_k^{\text{est}}$, we define the value function of a policy $\pi$:

**Definition 4.1.** For a given policy $\pi$, the value function $V^\pi : \mathcal{S} \to \mathbb{R}$ for $\mathcal{S} := \mathcal{S}_g \times \mathcal{S}_l^n$ is given by:

$$V^\pi(s) = \mathop{\mathbb{E}}_{a(t) \sim \pi(\cdot | s(t))} \left[ \sum_{t=0}^\infty \gamma^t r(s(t), a(t)) \,\middle|\, s(0) = s \right]. \tag{14}$$

Intuitively, $V^\pi(s)$ is the expected discounted cumulative reward when starting from state $s$ and applying actions from the policy $\pi$ across an infinite horizon.

With the above preparations, we are primed to present our main result: a high-probability bound on the optimality gap for our learned policy $\pi_{k,m}^{\text{est}}$ that decays with rate $\tilde{O}(1/\sqrt{k})$.

**Theorem 4.2.** *Let $\pi_{k,m}^{\text{est}}$ denote the learned policy deployed in* SUBSAMPLE-MFQ: Execution. *Then, for all $s_0 \in \mathcal{S} := \mathcal{S}_g \times \mathcal{S}_l^n$, we have*

$$V^{\pi^*}(s_0) - V^{\pi_{k,m}^{\text{est}}}(s_0) \leq \frac{\tilde{r}}{(1-\gamma)^2} \sqrt{\frac{n-k+1}{2nk}} \sqrt{\ln \frac{40\tilde{r}|\mathcal{S}_l||\mathcal{A}_l||\mathcal{A}_g|k^{|\mathcal{A}_l|+\frac{1}{2}}}{(1-\gamma)^2}} + \frac{1}{10\sqrt{k}} + 2\epsilon_{k,m}$$

We prove Theorem 4.4 in Appendix G, and provide a proof sketch in Appendix D. We also generalize the result to stochastic rewards in Appendix H.

To control the Bellman noise $\epsilon_{k,m}$, we show that for sufficiently many samples $m^*$, $\epsilon_{k,m^*}$ also decays on the order of $\tilde{O}(1/\sqrt{k})$ with high probability. For this, we introduce Lemma 4.3:

**Lemma 4.3** (Controlling the Bellman Noise.)**.** *For $k \in [n]$, let*

$$m^* = 2|\mathcal{S}_g||\mathcal{A}_g||\mathcal{S}_l||\mathcal{A}_l|k^{3.5+|\mathcal{S}_l||\mathcal{A}_l|} \frac{\log(|\mathcal{S}_g||\mathcal{A}_g||\mathcal{A}_l||\mathcal{S}_l|)}{(1-\gamma)^5} \log \frac{1}{(1-\gamma)^2}$$

*be the number of samples in Equation* (10). *If the number of iterations $T$ satisfies $T \geq \frac{2}{1-\gamma} \log \frac{\tilde{r}\sqrt{k}}{1-\gamma}$ then with probability at least $1 - \frac{1}{100e^k}$, the Bellman error satisfies $\epsilon_{k,m^*} \leq \tilde{O}(\frac{1}{\sqrt{k}})$.*

We defer the proof of Lemma 4.3 to Appendix G.1. To simplify notation, let $\pi_k^{\text{est}} := \pi_{k,m^*}^{\text{est}}$. Then, by combining Lemma 4.3 and Theorem 4.2, we obtain our main result in Theorem 4.4:

**Theorem 4.4.** *Let $\pi_k^{\text{est}}$ denote the learned policy from* SUBSAMPLE-MFQ: Execution *where the number of samples $m$ is determined in Lemma 4.3. Suppose $T \geq \frac{2}{1-\gamma} \log \frac{\tilde{r}\sqrt{k}}{1-\gamma}$. Then, $\forall s_0 \in \mathcal{S} := \mathcal{S}_g \times \mathcal{S}_l^n$, with probability at least $1 - 1/100e^k$, we have [3],*

$$V^{\pi^*}(s_0) - V^{\pi_k^{\text{est}}}(s_0) \leq \frac{\tilde{r}}{(1-\gamma)^2} \sqrt{\frac{n-k+1}{2nk} \ln \frac{40\tilde{r}|\mathcal{S}_l||\mathcal{A}_l||\mathcal{A}_g|k^{|\mathcal{A}_l|+\frac{1}{2}}}{(1-\gamma)^2}} + \frac{4}{\sqrt{k}}.$$

---
[3]The $1/100e^k$ term can be replaced to any arbitrary $\delta > 0$ at the cost of attaching logarithmic dependencies of $\delta$ on the $4/\sqrt{k}$ term in the error bound.

**Remark 4.5.** One could also derive an alternate bound that retains the $T$ factor via a $\gamma^T$ dependence in the final bound (which shows an exponentially decaying error with the time horizon $T$). Moreover, in Lemma G.10, we show that the query complexity is on the order of $O(mT|S_g||S_l|^k|A_g||A_l|^k)$, and we bound $T$ and set $m = \text{poly}(1/(1-\gamma))$ to attain the $1/\sqrt{k}$ rate.

**Remark 4.6.** Additionally, our $\text{poly}(1/(1-\gamma))$-dependence may be loose since we do not use more complicated variance reduction techniques as in Sidford et al. [2018a,b], Wainwright [2019], Jin et al. [2024] to optimize the number of samples $m$ which is used to bound the Bellman error $\epsilon_{k,m}$. Moreover, incorporating variance reduction would significantly complicate the algorithm and intuition.

Our analysis hinges on two non-trivial technical steps. Firstly, for an intermediary step in Theorem E.3 we establish that TV distance, rather than the stronger KL-divergence, is the correct metric to use (as the KL-divergence between $F_{z_\Delta}$ and $F_{z_{[n]}}$ decays too slowly as $k \to n$, as we show in Lemma F.6). This requires exploiting a recent extension of the DKW inequality to sampling without replacement [Anand and Qu, 2024], and showing that the transition dynamics we study saturates the data-processing inequality for TV-distance. Secondly, we adapt the celebrated performance-difference lemma [Kakade and Langford, 2002] to our multi-agent setting, which entails a principled analysis and careful probabilistic argument (to which we refer the reader to Appendix G).

**Remark 4.7.** We also extend the formulation of SUBSAMPLE-MFQ to off-policy $Q$-learning Chen et al. [2021b], Chen and Maguluri [2022a], which replaces the generative oracle assumption with a stochastic approximation scheme that learns an approximately optimal policy using historical data. Appendix I provides theoretical guarantees with a similar decaying optimality gap as in Theorem 4.4.

**Remark 4.8.** The asymptotic sample complexity of Algorithm 1 for learning $\hat{\pi}_k^{\text{est}}$ for a fixed $k$ is $\min\{\tilde{O}(|\mathcal{Z}_l|^k|\mathcal{S}_g||\mathcal{A}_g|), \tilde{O}(k^{|\mathcal{Z}_l|}|\mathcal{Z}_l||\mathcal{S}_g||\mathcal{A}_g|)\}$, which is at least polynomially faster the standard $Q$-learning or mean-field value iteration as discussed in Remark 3.1. By Theorem 4.4, as $k \to n$, the optimality gap decays, revealing a fundamental trade-off in the choice of $k$: increasing $k$ improves the performance of the policy, but increases the size of the $Q$-function. We explore this trade-off further in our experiments. If we set $k = O(\log n)$, this leads to a sample complexity of $\min\{\tilde{O}(n^{\log|\mathcal{Z}_l|}|\mathcal{S}_g||\mathcal{A}_g|), \tilde{O}((\log n)^{|\mathcal{Z}_l|}|\mathcal{S}_g||\mathcal{A}_g|)\}$. This is an exponential speedup on the complexity from mean-field value iteration (from $\text{poly}(n)$ to $\text{poly}(\log n)$), as well as over traditional value-iteration (from $\exp(n)$ to $\text{poly}(n)$), where the optimality gap decays to 0 with rate $O(\frac{1}{\sqrt{\log n}})$.

**Remark 4.9.** If $k = O(\log n)$, SUBSAMPLE-MFQ handles $|\mathcal{E}| \leq O(\log n/\log\log n)$ types of local agents, since the run-time of the learning algorithm becomes $\text{poly}(n)$. This surpasses the previous heterogeneity capacity from Mondal et al. [2022], which only handles constant $|\mathcal{E}| \leq O(1)$. Increasing the agent heterogeneity using this *type formulation* does make the algorithm somewhat more expensive since it factors into the query complexity via the state space of the local agents; however, since the optimality gap of the learned policy is on the order of $\tilde{O}(1/\sqrt{k})$ (modulo very small $\sqrt{\log|\mathcal{S}_l||\mathcal{A}_l|}$ factors), increasing the amount of agent heterogeneity does not degrade the quality of the learned policy as the theoretical bound does not get worse. Moreover, recent methods from the graphon mean-field MARL community might be able to enable stronger heterogeneity in the system [Cui and Koeppl, 2022, Anand and Liaw, 2025, Hu et al., 2023].

In the non-tabular setting with infinite state/action spaces, one could replace the $Q$-learning algorithm with any arbitrary value-based RL method that learns $\hat{Q}_k$ with function approximation [Sutton et al., 1999] such as deep $Q$-networks [Silver et al., 2016]. Doing so raises an additional error that factors into Theorem 4.4. We formalize this below.

**Assumption 4.10** (Linear MDP with infinite state spaces). Suppose $\mathcal{S}_g$ and $\mathcal{S}_l$ are infinite compact sets. Furthermore, suppose there exists a feature map $\phi : \mathcal{S} \times \mathcal{A} \to \mathbb{R}^d$ and $d$ unknown (signed) measures $\mu = (\mu^1, \ldots, \mu^d)$ over $\mathcal{S}$ and a vector $\theta \in \mathbb{R}^d$ such that for any $(s,a) \in \mathcal{S} \times \mathcal{A}$, we have $\mathbb{P}(\cdot|s,a) = \langle\phi(s,a), \mu(\cdot)\rangle$ and $r(s,a) = \langle\phi(s,a), \theta\rangle$.

The existence of $\phi : \mathcal{S} \times \mathcal{A} \to \mathbb{R}^d$ implies one can estimate the $Q$-function of any policy as a linear function. This assumption is commonly used in policy iteration algorithms Lattimore et al. [2020], Wang et al. [2023], and allows one to obtain sample complexity bounds that are independent of $|\mathcal{S}_l|$ and $|\mathcal{A}_l|$. Finally, as is standard in RL, we assume bounded feature-norms [Tkachuk et al., 2023]:

**Assumption 4.11** (Bounded features). We assume that $\|\phi(s,a)\|_2 \leq 1$ for all $(s,a) \in \mathcal{S} \times \mathcal{A}$.

Then, through a reduction from Zhang et al. [2024], Ren et al. [2024] that uses function approximation to learn the spectral features $\phi_k$ for $\hat{Q}_k$, we derive a performance guarantee for the learned policy $\pi_k^{\text{est}}$, where the optimality gap decays with $k$.

**Theorem 4.12.** *When $\pi_k^{\text{est}}$ is derived from the spectral features $\phi_k$ learned in $\hat{Q}_k$, and $M$ is the number of samples used in the function approximation, then*

$$\Pr\left[V^{\pi^*}(s) - V^{\pi_k^{\text{est}}}(s) \le \tilde{O}\left(\frac{1}{\sqrt{k}} + \frac{\|\phi_k\|^5 \log 2k^2}{\sqrt{M}} + \frac{2\gamma\tilde{r} \cdot \|\phi_k\|}{(1-\gamma)\sqrt{k}}\right)\right] \ge 1 + \frac{200}{k} - \frac{201}{200\sqrt{k}}$$

We defer the proof of Theorem 4.12 to Appendix J.

## 5 Conclusion and Future Works

This work develops subsampling for mean field MARL in a cooperative system with a global decision-making agent and $n$ homogeneous local agents. We propose `SUBSAMPLE-MFQ` which learns each agent's best response to the mean effect from a sample of its neighbors, allowing an exponential reduction on the sample complexity of approximating a solution to the MDP. We provide a theoretical analysis on the optimality gap of the learned policy, showing that (with high probability) the learned policy converges to the optimal policy with the number of agents $k$ sampled at the rate $\tilde{O}(1/\sqrt{k})$, and validate our theoretical results through numerical experiments. We show that the decay rate is maintained, on expectation, when the reward functions are stochastic and when agents learn from a single trajectory on historical data through off-policy $Q$-learning. Finally, we extend this result to the non-tabular setting with infinite state and action spaces under assumptions of a linear MDP model.

**Limitations and future work.** Our current work assumes that the global and local agents cooperate to optimize a structured reward under a specific dynamic model. While this model is more general than the federated learning setting, one direction would be to extend our algorithms and analysis to weaker network assumptions. We believe our framework, which can handle dense subgraphs, as well as expander-graph decompositions [Reingold, 2008, Anand and Umans, 2023, Anand, 2025] may be amenable for this. Secondly, our current work incorporates mild heterogeneity among agents and assumes they are cooperative; thus, another avenue would be to consider settings with competitive agents or more complex agent heterogeneity. Finally, it would be exciting to generalize this work to the online no-regret setting.

**Societal Impacts.** This work is theoretical and foundational in nature. As such, while it enables more scalable multi-agent algorithms, it is not tied to any specific applications or deployments.

**Acknowledgements.** This work was supported by NSF Grants 2154171, CAREER Award 2339112, CMU CyLab Seed Funding, and CCF 2338816. We gratefully acknowledge insightful discussions with Siva Theja Maguluri, Yiheng Lin, Jan van den Brand, Adam Wierman, Yunbum Kook, Yi Wu, Sarah Liaw, and Rishi Veerapaneni. Finally, we thank the anonymous NeurIPS reviewers for their helpful feedback.

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

# A  Mathematical Background and Additional Remarks

**Outline of the Appendices**.

- Appendix B presents numerical simulations on the performance of `SUBSAMPLE-MFQ` on the Gaussian squeeze and constrained exploration tasks.

- Appendix C presents notation and basic lemmas involving the learned $\hat{Q}_k$-function, as well as a stable and practical implementation of Algorithm 1.

- Appendix D presents a proof sketch of our main result in Theorem 4.4.

- Appendix E presents the proof of the Lipschitz continuity between $\hat{Q}_k$ and $Q^*$

- Appendix F presents our bound on the TV-distance in Theorem D.2.

- Appendix G proves the bound on the optimality gap between the learned policy $\tilde{\pi}_{k,m}^{\text{est}}$ and the optimal policy $\pi^*$.

- Appendix H presents an extension of the result to stochastic rewards.

- Appendix I presents an extension of the result to off-policy learning that follows by using historical data.

- Appendix J presents an extension of the result to continuous state/action spaces under a linear MDP assumption.

- Appendix K presents some practical derandomized variants with randomness-sharing.

Table 1: Important notations in this paper.

| Notation | Meaning |
|---|---|
| $\|\cdot\|_1$ | $\ell_1$ (Manhattan) norm; |
| $\|\cdot\|_\infty$ | $\ell_\infty$ norm; |
| $\mathbb{Z}_+$ | The set of strictly positive integers; |
| $\mathbb{R}^d$ | The set of $d$-dimensional reals; |
| $[m]$ | The set $\{1,\dots,m\}$, where $m \in \mathbb{Z}_+$; |
| $\binom{[m]}{k}$ | The set of $k$-sized subsets of $\{1,\dots,m\}$; |
| $a_g$ | $a_g \in \mathcal{A}_g$ is the action of the global agent; |
| $s_g$ | $s_g \in \mathcal{S}_g$ is the state of the global agent; |
| $a_1,\dots,a_n$ | $a_1,\dots,a_n \in \mathcal{A}_l^n$ are the actions of the local agents $1,\dots,n$; |
| $s_1,\dots,s_n$ | $s_1,\dots,s_n \in \mathcal{S}_l^n$ are the states of the local agents $1,\dots,n$; |
| $a$ | $a = (a_g, a_1,\dots,a_n) \in \mathcal{A}_g \times \mathcal{A}_l^n$ is the tuple of actions of all agents; |
| $s$ | $s = (s_g, s_1,\dots,s_n) \in \mathcal{S}_g \times \mathcal{S}_l^n$ is the tuple of states of all agents; |
| $z_i$ | $z_i = (s_i, a_i) \in \mathcal{Z}_l$, for $i \in [n]$; |
| $\mu_k(\mathcal{Z}_l)$ | $\mu_k(\mathcal{Z}_l) = \{0, 1/k, 2/k, \dots, 1\}^{|\mathcal{Z}_l|}$; |
| $\mu(\mathcal{Z}_l)$ | $\mu(\mathcal{Z}_l) := \mu_n(\mathcal{Z}_l)\{0, 1/n, 2/n, \dots, 1\}^{|\mathcal{Z}_l|}$; |
| $s_\Delta$ | For $\Delta \subseteq [n]$, and a collection of variables $\{s_1,\dots,s_n\}$, $s_\Delta := \{s_i : i \in \Delta\}$; |
| $\sigma(z_\Delta, z'_\Delta)$ | Product sigma-algebra generated by sequences $z_\Delta$ and $z'_\Delta$; |
| $\pi^*$ | $\pi^*$ is the optimal deterministic policy function such that $a = \pi^*(s)$; |
| $\hat{\pi}_k^*$ | $\hat{\pi}_k^*$ is the optimal deterministic policy function on a constrained system of $|\Delta| = k$ local agents; |
| $\tilde{\pi}_k^{\text{est}}$ | $\tilde{\pi}_k^{\text{est}}$ is the stochastic policy map learned with parameter $k$ such that $a \sim \tilde{\pi}_k^{\text{est}}(s)$; |
| $P_g(\cdot|s_g, a_g)$ | $P_g(\cdot|s_g, a_g)$ is the stochastic transition kernel for the state of the global agent; |
| $P_l(\cdot|a_i, s_i, s_g)$ | $P_l(\cdot|a_i, s_i, s_g)$ is the stochastic transition kernel for the state of local agent $i \in [n]$; |
| $r_g(s_g, a_g)$ | $r_g$ is the global agent's component of the reward; |
| $r_l(s_i, s_g, a_i)$ | $r_l$ is the component of the reward for local agent $i \in [n]$; |
| $r(s, a)$ | $r(s, a) = r_g(s_g, a_g) + \frac{1}{n}\sum_{i\in[n]} r_l(s_i, s_g, a_i)$ is the reward of the system; |
| $r_\Delta(s, a)$ | $r_\Delta(s, a) = r_g(s_g, a_g) + \frac{1}{|\Delta|}\sum_{i\in\Delta} r_l(s_i, s_g, a_i)$ is the constrained system's reward with $|\Delta| = k$ local agents; |
| $\mathcal{T}$ | $\mathcal{T}$ is the centralized Bellman operator; |
| $\hat{\mathcal{T}}_k$ | $\hat{\mathcal{T}}_k$ is the Bellman operator on a constrained system of $|\Delta| = k$ local agents; |
| $\Pi^\Theta(y)$ | $\ell_1$ projection of $y$ onto set $\Theta$; |

**Definition A.1** (Lipschitz continuity). Given metric spaces $(\mathcal{X}, d_{\mathcal{X}})$ and $(\mathcal{Y}, d_{\mathcal{Y}})$ and a constant $L > 0$, a map $f : \mathcal{X} \to \mathcal{Y}$ is $L$-Lipschitz continuous if for all $x, y \in \mathcal{X}$, $d_{\mathcal{Y}}(f(x), f(y)) \leq L \cdot d_{\mathcal{X}}(x, y)$.

**Theorem A.2** (Banach-Caccioppoli fixed point theorem [Banach, 1922]). *Consider the metric space $(\mathcal{X}, d_{\mathcal{X}})$, and $T : \mathcal{X} \to \mathcal{X}$ such that $T$ is a $\gamma$-Lipschitz continuous mapping for $\gamma \in (0, 1)$. Then, by the Banach-Cacciopoli fixed-point theorem, there exists a unique fixed point $x^* \in \mathcal{X}$ for which $T(x^*) = x^*$. Additionally, $x^* = \lim_{s \to \infty} T^s(x_0)$ for any $x_0 \in \mathcal{X}$.*

# B    Numerical Experiments

This section provides numerical simulations for the examples outlined in Section 2. All experiments were run on a 2-core CPU server with 12GB RAM. We chose a parameter complexity for each simulation that was sufficient to emphasize characteristics of the theory, such as the complexity improvement and the decaying optimality gap.[4]

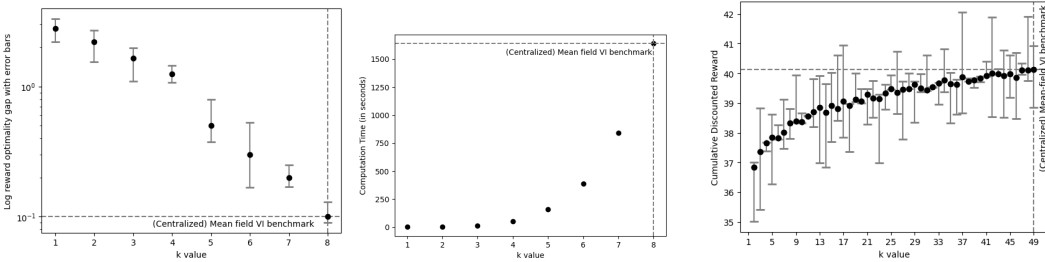

Figure 2: a) Reward optimality gap (log scale) with $\pi_{k,m}^{\text{est}}$ running 300 iterations. b) Computation time (in minutes) against sampling parameter $k$, for $k \leq n = 8$, to learn policy $\hat{\pi}_{k,m}^{\text{est}}$. c) Discounted cumulative rewards for $k \leq n = 50$.

## B.1    Constrained Exploration

Let $\mathcal{S}_g = \mathcal{S}_l = [6]^2$ where for $s_a \in \{\mathcal{S}_g, \mathcal{S}_l\}, s_a^{(i)}$ denotes the projection of $s_a$ onto its $i$'th coordinate. Further, let $\mathcal{A}_l = \mathcal{A}_g = \{\text{"up", "down", "right", "left"}\}$ given formally by $\{\binom{1}{0}, \binom{-1}{0}, \binom{0}{1}, \binom{0}{-1}\}$. Let $\Pi_D(x)$ denote the $\ell_1$-projection of $x$ onto set $D$. Then, let $s_g^{t+1}(s_g^t, a_g^t) = \Pi_{S_g}(s_g^t + a_g^t), s_i^{t+1}(s_i^t, s_g^t, a_i^t) = \Pi_{S_l}(s_i^t + |s_i^t - s_g^t| + a_i^t)$. We let the global agent's reward be $r_g(s_g, a_g) = 2\sum_{i=1}^3 \mathbb{1}\{a_g = \binom{1}{0}, s_g^{(1)} \leq i\} + 2\sum_{i=3}^6 \mathbb{1}\{a_g = \binom{-1}{0}, s_g^{(1)} > i\}$, and the local agent's rewards be $r_l(s_i, s_g, a_i) = \mathbb{1}\{a_i^t \neq 0\} + 6 \cdot \mathbb{1}\{s_i^{(1)} = s_g^{(1)}\} + 2 \cdot \|s_i^t - s_g^t\|_1$.

Intuitively, the reward function is designed to force the global agent to oscillate vertically in the grid, while forcing the local agents to oscillate horizontally around the global agent, thereby simulating a constrained exploration. This model has been studied previously in Lin et al. [2023b].

For this task, we ran a simulation with $n = 8$ agents, with $m = 20$ samples in the empirically adapted Bellman operator. We provide simulation results in Figure 2a. We observe monotonic improvements in the cumulative discounted rewards as $k \to n$. Since $k = n$ recovers value-iteration and mean-field MARL algorithms, the reward at $k = n$ is the baseline we compare our algorithm to. When $k < n$, we observe that the reward accrued by SUBSAMPLE-MFQ is only marginally less than the reward gained by value-iteration.

## B.2    Gaussian Squeeze (GS)

In this task, $n$ homogeneous agents determine their individual action $a_i$ to jointly maximize the objective $r(x) = xe^{-(x-\mu)^2/\sigma^2}$, where $x = \sum_{i=1}^n a_i$, $a_i \in \{0, \ldots, 9\}$, and $\mu$ and $\sigma$ are the predefined mean and variance of the system. We consider a homogeneous system *devoid* of a global

---

[4]We provide supporting code for the algorithm and experiments in `https://github.com/emiletimothy/Mean-Field-Subsample-Q-Learning`

agent to match the mean-field setting in Yang et al. [2018], where the richness of our model setting can still express GS. We use the global agent to model the state of the system, given by the number of vehicles in each controller's lane.

We set $\mathcal{S}_l = [4]$, $\mathcal{A}_l = \{0, \ldots, 4\}$, and $\mathcal{S}_g = \mathcal{S}_l$ such that $s_g = \left\lceil \frac{1}{n} \sum_{i=1}^n s_i \right\rceil$, and $\mathcal{A}_g = \{0\}$. The transition functions are given by $s_i(t+1) = s_i(t) - 1 \cdot \mathbb{1}\{s_i(t) > s_g(t)\} + \mathrm{Ber}(p)$ and $s_g(t+1) = \left\lceil \frac{1}{n} \sum_{i=1}^n s_i(t+1) \right\rceil$, where $\mathrm{Ber}(p)$ is a Bernoulli random variable with parameter $p > 0$. Finally, the global agent's reward function is given by $r_g(s_g, a_g) = -s_g$ and the local agent's reward function is given by $r_l(s_i, s_g, a_i) = s_i \cdot e^{-(s_i - s_g)^2/4} - \max\{s_i, a_i\}$.

For this task, we ran a small-scale simulation with $n = 8$ agents, and a large-scale simulation with $n = 50$ agents, and used $m = 20$ samples in the empirical Bellman operator. We provide simulation results in Figure 2b and Figure 2c, where Figure 2b demonstrates the exponential improvement in computational complexity of SUBSAMPLE-MFQ, and Figure 2c demonstrates a monotonic improvement in the cumulative rewards, consistent with Theorem 4.4. Here, both metrics outperform the mean-field value iteration benchmark.

## C  Notation and Basic Lemmas

For convenience, we restate below the various Bellman operators under consideration.

**Definition C.1** (Bellman Operator $\mathcal{T}$).

$$\mathcal{T}Q^t(s, a) := r(s, a) + \gamma \mathbb{E}_{\substack{s'_g \sim P_g(\cdot|s_g, a_g), \\ s'_i \sim P_l(\cdot|s_i, s_g, a_i), \forall i \in [n]}} \max_{a' \in \mathcal{A}} Q^t(s', a') \tag{15}$$

**Definition C.2** (Adapted Bellman Operator $\hat{\mathcal{T}}_k$). The adapted Bellman operator updates a smaller $Q$ function (which we denote by $\hat{Q}_k$), for a surrogate system with the global agent and $k \in [n]$ local agents denoted by $\Delta$, using mean-field value iteration and $j \in \Delta$ such that:

$$\hat{\mathcal{T}}_k \hat{Q}_k^t(s_g, s_j, F_{z_{\Delta \setminus j}}, a_j, a_g) := r_\Delta(s, a) + \gamma \mathbb{E}_{\substack{s'_g \sim P_g(\cdot|s_g, a_g), \\ s'_i \sim P_l(\cdot|s_i, s_g, a_i), \forall i \in \Delta}} \max_{a' \in \mathcal{A}} \hat{Q}_k^t(s'_g, s'_j, F_{z'_{\Delta \setminus j}}, a'_j, a'_g) \tag{16}$$

**Definition C.3** (Empirical Adapted Bellman Operator $\hat{\mathcal{T}}_{k,m}$). The empirical adapted Bellman operator $\hat{\mathcal{T}}_{k,m}$ empirically estimates the adapted Bellman operator update using mean-field value iteration by drawing $m$ random samples of $s_g \sim P_g(\cdot|s_g, a_g)$ and $s_i \sim P_l(\cdot|s_i, s_g, a_i)$ for $i \in \Delta$, where for $\ell \in [m]$, the $\ell$'th random sample is given by $s_g^\ell$ and $s_\Delta^\ell$, and $j \in \Delta$:

$$\hat{\mathcal{T}}_{k,m} \hat{Q}_{k,m}^t(s_g, s_j, F_{z_{\Delta \setminus j}}, a_j, a_g) := r_\Delta(s, a) + \frac{\gamma}{m} \sum_{\ell \in [m]} \max_{a' \in \mathcal{A}} \hat{Q}_{k,m}^t(s_g^\ell, s_j^\ell, F_{z_{\Delta \setminus j}^\ell}, a_j^\ell, a_g^\ell) \tag{17}$$

**Lemma C.4.** For any $\Delta \subseteq [n]$ such that $|\Delta| = k$, suppose $0 \le r_\Delta(s, a) \le \tilde{r}$. Then, for all $t \in \mathbb{N}$, $\hat{Q}_k^t \le \frac{\tilde{r}}{1-\gamma}$.

*Proof.* The proof follows by induction on $t$. The base case follows from $\hat{Q}_k^0 := 0$. For the induction, note that by the triangle inequality $\|\hat{Q}_k^{t+1}\|_\infty \le \|r_\Delta\|_\infty + \gamma \|\hat{Q}_k^t\|_\infty \le \tilde{r} + \gamma \frac{\tilde{r}}{1-\gamma} = \frac{\tilde{r}}{1-\gamma}$. $\qquad \square$

**Remark C.5.** By the law of large numbers, $\lim_{m \to \infty} \hat{\mathcal{T}}_{k,m} = \hat{\mathcal{T}}_k$, where the error decays in $O(1/\sqrt{m})$ by the Chernoff bound. Also, $\hat{\mathcal{T}}_n := \mathcal{T}$. Further, Lemma C.4 is independent of the choice of $k$. Therefore, for $k = n$, this implies an identical bound on $Q^t$. An identical argument implies the same bound on $\hat{Q}_{k,m}^t$.

$\mathcal{T}$ satisfies a $\gamma$-contractive property under the infinity norm [Watkins and Dayan, 1992]. We similarly show that $\hat{\mathcal{T}}_k$ and $\hat{\mathcal{T}}_{k,m}$ satisfy a $\gamma$-contractive property under infinity norm in Lemmas C.6 and C.7.

**Lemma C.6.** $\hat{\mathcal{T}}_k$ satisfies the $\gamma$-contractive property under infinity norm:

$$\|\hat{\mathcal{T}}_k \hat{Q}'_k - \hat{\mathcal{T}}_k \hat{Q}_k\|_\infty \le \gamma \|\hat{Q}'_k - \hat{Q}_k\|_\infty \tag{18}$$

*Proof.* Suppose we apply $\hat{\mathcal{T}}_k$ to $\hat{Q}_k(s_g, F_{z_\Delta}, a_g)$ and $\hat{Q}'_k(s_g, F_{z_\Delta}, a_g)$ for $|\Delta| = k$. Then:

$$\|\hat{\mathcal{T}}_k \hat{Q}'_k - \hat{\mathcal{T}}_k \hat{Q}_k\|_\infty$$

$$= \gamma \max_{\substack{s_g \in \mathcal{S}_g, \\ a_g \in \mathcal{A}_g, \\ F_{z_\Delta} \in \mu_k(\mathcal{Z}_l)}} \left| \mathbb{E}_{\substack{s'_g \sim P_g(\cdot|s_g, a_g), \\ s'_i \sim P_l(\cdot|s_i, s_g, a_i), \\ \forall i \in \Delta,}} \max_{a' \in \mathcal{A}} \hat{Q}'_k(s'_g, F_{z'_\Delta}, a'_g) - \mathbb{E}_{\substack{s'_g \sim P_g(\cdot|s_g, a_g), \\ s'_i \sim P_l(\cdot|s_i, s_g, a_i), \\ \forall i \in \Delta'}} \max_{a' \in \mathcal{A}} \hat{Q}_k(s'_g, F_{z'_\Delta}, a'_g) \right|$$

$$\leq \gamma \max_{s'_g \in \mathcal{S}_g, F_{z_\Delta} \in \mu_k(\mathcal{Z}_l), a' \in \mathcal{A}} \left| \hat{Q}'_k(s'_g, F_{z'_\Delta}, a'_g) - \hat{Q}_k(s'_g, F_{z'_\Delta}, a'_g) \right| = \gamma \|\hat{Q}'_k - \hat{Q}_k\|_\infty$$

The equality cancels common $r_\Delta(s, a)$ terms in each operator. The second line uses Jensen's inequality, maximizes over actions, and bounds expected values with the maximizers of the random variables. $\square$

**Lemma C.7.** $\hat{\mathcal{T}}_{k,m}$ *satisfies the $\gamma$-contractive property under infinity norm.*

*Proof.* Similarly to Lemma C.6, suppose we apply $\hat{\mathcal{T}}_{k,m}$ to $\hat{Q}_{k,m}(s_g, F_{z_\Delta}, a_g)$ and $\hat{Q}'_{k,m}(s_g, F_{z_\Delta}, a_g)$. Then:

$$\|\hat{\mathcal{T}}_{k,m} \hat{Q}_k - \hat{\mathcal{T}}_{k,m} \hat{Q}'_k\|_\infty = \frac{\gamma}{m} \left\| \sum_{\ell \in [m]} \left( \max_{a' \in \mathcal{A}} \hat{Q}_k(s^\ell_g, F_{z^\ell_\Delta}, a'_g) - \max_{a' \in \mathcal{A}} \hat{Q}'_k(s^\ell_g, F_{z^\ell_\Delta}, a'_g) \right) \right\|_\infty$$

$$\leq \gamma \max_{a'_g \in \mathcal{A}_g, s'_g \in \mathcal{S}_g, z_\Delta \in \mathcal{Z}^k_l} |\hat{Q}_k(s'_g, F_{z'_\Delta}, a'_g) - \hat{Q}'_k(s'_g, F_{z'_\Delta}, a'_g)|$$

$$= \gamma \|\hat{Q}_k - \hat{Q}'_k\|_\infty$$

The first inequality uses the triangle inequality and the general property $|\max_{a \in A} f(a) - \max_{b \in A} f(b)| \leq \max_{c \in A} |f(a) - f(b)|$. The last line recovers the definition of infinity norm. $\square$

**Remark C.8.** The $\gamma$-contractivity of $\hat{\mathcal{T}}_k$ and $\hat{\mathcal{T}}_{k,m}$ attracts the trajectory between two $\hat{Q}_k$ and $\hat{Q}_{k,m}$ functions on the same state-action tuple by $\gamma$ at each step. Repeatedly applying the Bellman operators produces a unique fixed-point from the Banach fixed-point theorem which we introduce in Definitions C.9 and C.10.

**Definition C.9** ($\hat{Q}^*_k$-function)**.** Suppose $\hat{Q}^0_k := 0$ and let $\hat{Q}^{t+1}_k(s_g, F_{z_\Delta}, a_g) = \hat{\mathcal{T}}_k \hat{Q}^t_k(s_g, F_{z_\Delta}, a_g)$ for $t \in \mathbb{N}$. Denote the fixed-point of $\hat{\mathcal{T}}_k$ by $\hat{Q}^*_k$ such that $\hat{\mathcal{T}}_k \hat{Q}^*_k(s_g, F_{z_\Delta}, a_g) = \hat{Q}^*_k(s_g, F_{z_\Delta}, a_g)$.

**Definition C.10** ($\hat{Q}^{\text{est}}_{k,m}$-function)**.** Let $\hat{Q}^0_{k,m} := 0$ and $\hat{Q}^{t+1}_{k,m}(s_g, F_{z_\Delta}, a_g) = \hat{\mathcal{T}}_{k,m} \hat{Q}^t_{k,m}(s_g, F_{z_\Delta}, a_g)$ for $t \in \mathbb{N}$. Then, the Banach-Cacciopoli fixed-point of the adapted Bellman operator $\hat{\mathcal{T}}_{k,m}$ is $\hat{Q}^{\text{est}}_{k,m}$ such that $\hat{\mathcal{T}}_{k,m} \hat{Q}^{\text{est}}_{k,m}(s_g, F_{z_\Delta}, a_g) = \hat{Q}^{\text{est}}_{k,m}(s_g, F_{z_\Delta}, a_g)$.

**Corollary C.11.** Observe that by recursively using the $\gamma$-contractive property for $T$ time steps:

$$\|\hat{Q}^*_k - \hat{Q}^T_k\|_\infty \leq \gamma^T \cdot \|\hat{Q}^*_k - \hat{Q}^0_k\|_\infty \tag{19}$$

$$\|\hat{Q}^{\text{est}}_{k,m} - \hat{Q}^T_{k,m}\|_\infty \leq \gamma^T \cdot \|\hat{Q}^{\text{est}}_{k,m} - \hat{Q}^0_{k,m}\|_\infty \tag{20}$$

*Further, noting that $\hat{Q}^0_k = \hat{Q}^0_{k,m} := 0$, $\|\hat{Q}^*_k\|_\infty \leq \frac{\tilde{r}}{1-\gamma}$, and $\|\hat{Q}^{\text{est}}_{k,m}\|_\infty \leq \frac{\tilde{r}}{1-\gamma}$ from Lemma C.4:*

$$\|\hat{Q}^*_k - \hat{Q}^T_k\|_\infty \leq \gamma^T \frac{\tilde{r}}{1-\gamma}, \tag{21}$$

$$\|\hat{Q}^{\text{est}}_{k,m} - \hat{Q}^T_{k,m}\|_\infty \leq \gamma^T \frac{\tilde{r}}{1-\gamma} \tag{22}$$

**Remark C.12.** Corollary C.11 characterizes the error decay between $\hat{Q}^T_k$ and $\hat{Q}^*_k$ and shows that it decays exponentially in the number of Bellman iterations by a $\gamma^T$ multiplicative factor.

Furthermore, we characterize the maximal policies greedy policies obtained from $Q^*$, $\hat{Q}^*_k$, and $\hat{Q}^{\text{est}}_{k,m}$.

**Definition C.13** (Optimal policy $\pi^*$). The greedy policy derived from $Q^*$ is

$$\pi^*(s) := \arg\max_{a \in \mathcal{A}} Q^*(s, a). \tag{23}$$

**Definition C.14** (Optimal subsampled policy $\hat{\pi}_k^*$). The greedy policy from $\hat{Q}_k^*$ is

$$\hat{\pi}_k^*(s_g, s_i, F_{s_{\Delta \setminus i}}) := \arg\max_{(a_g, a_i, F_{a_{\Delta \setminus i}}) \in \mathcal{A}_g \times \mathcal{A}_l \times \mu_{k-1}(\mathcal{A}_l)} \hat{Q}_k^*(s_g, s_i, F_{z_{\Delta \setminus i}}, a_i, a_g). \tag{24}$$

**Definition C.15** (Optimal empirically subsampled policy $\hat{\pi}_{k,m}^{\text{est}}$). The greedy policy from $\hat{Q}_{k,m}^{\text{est}}$ is given by

$$\hat{\pi}_{k,m}^{\text{est}}(s_g, F_{s_\Delta}) := \arg\max_{(a_g, a_i, F_{a_{\Delta \setminus i}}) \in \mathcal{A}_g \times \mathcal{A}_l \times \mu_{k-1}(\mathcal{A}_l)} \hat{Q}_{k,m}^{\text{est}}(s_g, s_i, F_{z_{\Delta \setminus i}}, a_i, a_g). \tag{25}$$

Figure 4 details the analytic flow on how we use the empirical adapted Bellman operator to perform value iteration on $\hat{Q}_{k,m}$ to get $\hat{Q}_{k,m}^{\text{est}}$ which approximates $Q^*$.

Algorithm 3 gives a stable implementation of Algorithm 1 with learning rates $\{\eta_t\}_{t \in [T]}$. Algorithm 3 is provably numerical stable under fixed-point arithmetic [Anand et al., 2024, 2025]. $\hat{Q}_{k,m}^t$ in Algorithm 3 is $\gamma$-contractive as in Lemma C.6, given an appropriately conditioned sequence of learning rates $\eta_t$:

---

**Algorithm 3** Stable (Practical) Implementation of Algorithm 1: `SUBSAMPLE-Q`: Learning

---

**Require:** A multi-agent system as described in Section 2. Parameter $T$ for the number of iterations in the initial value iteration step. Hyperparameter $k \in [n]$. Discount parameter $\gamma \in (0, 1)$. Oracle $\mathcal{O}$ to sample $s_g' \sim P_g(\cdot | s_g, a_g)$ and $s_i \sim P_l(\cdot | s_i, s_g, a_i)$ for all $i \in [n]$. Learning rate sequence $\{\eta_t\}_{t \in [T]}$ where $\eta_t \in (0, 1]$.

1: Let $\Delta = [k]$.
2: **for** $(s_g, s_1, F_{z_{\Delta \setminus 1}}) \in \mathcal{S}_g \times \mathcal{S}_l \times \mu_{k-1}(\mathcal{Z}_l)$ **do**
3:    **for** $(a_g, a_1) \in \mathcal{A}_g \times \mathcal{A}_l$ **do**
4:       Set $\hat{Q}_{k,m}^0(s_g, s_1, F_{z_{\Delta \setminus 1}}, a_1, a_g) = 0$
5: **for** $t = 1$ to $T$ **do**
6:    **for** $(s_g, s_1, F_{z_{\Delta \setminus 1}}) \in \mathcal{S}_g \times \mathcal{S}_l \times \mu_{k-1}(\mathcal{Z}_l)$ **do**
7:       **for** $(a_g, a_1) \in \mathcal{A}_g \times \mathcal{A}_l$ **do**
8:

$$\hat{Q}_{k,m}^{t+1}(s_g, s_1, F_{z_{\Delta \setminus 1}}, a_g, a_1) = (1 - \eta_t)\hat{Q}_{k,m}^t(s_g, s_1, F_{z_{\Delta \setminus 1}}, a_g, a_1)$$
$$+ \eta_t \hat{\mathcal{T}}_{k,m} \hat{Q}_{k,m}^t(s_g, s_1, F_{z_{\Delta \setminus 1}}, a_g, a_1)$$

9: Let the approximate policy be

$$\hat{\pi}_{k,m}^T(s_g, s_1, F_{s_{\Delta \setminus 1}}) = \arg\max_{(a_g, a_1, a_{\Delta \setminus 1}) \in \mathcal{A}_g \times \mathcal{A}_l \times \mu_{k-1}(\mathcal{A}_l)} \hat{Q}_{k,m}^T(s_g, s_1, F_{z_{\Delta \setminus 1}}, a_g, a_1).$$

---

**Theorem C.16.** As $T \to \infty$, if $\sum_{t=1}^T \eta_t = \infty$, and $\sum_{t=1}^T \eta_t^2 < \infty$, then Q-learning converges to the optimal Q function asymptotically with probability 1.

Furthermore, finite-time guarantees with the learning rate and sample complexity have been shown in Chen and Maguluri [2022b], which when adapted to our $\hat{Q}_{k,m}$ framework in Algorithm 3 yields:

**Theorem C.17** (Chen and Maguluri [2022b]). *For all $t = \{1, \ldots, T\}$ and for any $\epsilon > 0$, if the learning rate sequence $\eta_t$ satisfies $\eta_t = (1 - \gamma)^4 \epsilon^2$ and $T = \frac{|\mathcal{S}_l||\mathcal{A}_l|k^{|\mathcal{S}_l||\mathcal{A}_l|}|\mathcal{S}_g||\mathcal{A}_g|}{(1-\gamma)^5}\epsilon^2$, then*

$$\|\hat{Q}_{k,m}^T - \hat{Q}_{k,m}^{\text{est}}\| \le \epsilon.$$

**Definition C.18** (Star Graph $S_n$). For $n \in \mathbb{N}$, the star graph $S_n$ is the complete bipartite graph $K_{1,n}$.

$S_n$ captures the graph density notion by saturating the set of neighbors of the central node. Such settings find applications beyond RL as well [Chaudhari et al., 2024, Anand and Qu, 2024, Li et al., 2019]. The cardinality of the search space simplex for the optimal policy is exponential in $n$, so it cannot be naively modeled by an MDP: we need to exploit the symmetry of the local agents. This intuition allows our subsampling algorithm to run in polylogarithmic time (in $n$). Some works leverage an exponential decaying property that truncates the search space for policies over immediate neighborhoods of agents; however, this still relies on the assumption that the graph neighborhood for the agent is sparse [Qu et al., 2020a,b]; however, $S_n$ is not locally sparse; hence, previous methods do not apply to this problem instance.

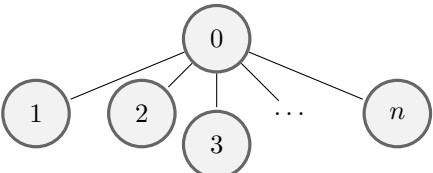

Figure 3: Star graph $S_n$

Finally, we argue why mean-field value iteration faithfully performs the same update as value iteration.

**Lemma C.19** (Equivalence of Mean-Field Value Iteration and Value Iteration). Since the local agents $1, \ldots, n$ are all homogeneous in their state/action spaces, the $\hat{Q}_k$-function only depends on them through their empirical distribution $F_{z_\Delta}$, proving the lemma. Therefore, for the remainder of the paper, we will use $\hat{Q}_k(s_g, F_{z_\Delta}, a_g) := \hat{Q}_k(s_g, s_i, F_{z_{\Delta \setminus i}}, a_g, a_i) := \hat{Q}_k(s_g, s_\Delta, a_g, a_\Delta)$ interchangeably, unless making a remark about the computational complexity of learning each function.

# D   Proof Sketch

This section details an outline for the proof of Theorem 4.4, as well as some key ideas. At a high level, our SUBSAMPLE-MFQ framework recovers exact mean-field $Q$ learning and traditional value iteration when $k = n$ and as $m \to \infty$. Further, as $k \to n$, $\hat{Q}_k^*$ should intuitively get closer to $Q^*$ from which the optimal policy is derived. Thus, the proof is divided into three major steps: firstly, we prove a Lipschitz continuity bound between $\hat{Q}_k^*$ and $\hat{Q}_n^*$ in terms of the total variation (TV) distance between $F_{z_\Delta}$ and $F_{z_{[n]}}$. Next, we bound the TV distance between $F_{z_\Delta}$ and $F_{z_{[n]}}$. Finally, we bound the value differences between $\pi_k^{\text{est}}$ and $\pi^*$ by bounding $Q^*(s, \pi^*(s)) - Q^*(s, \pi_k^{\text{est}}(s))$ and then using the performance difference lemma from Kakade and Langford [2002].

**Step 1: Lipschitz Continuity Bound.** To compare $\hat{Q}_k^*(s_g, F_{s_\Delta}, a_g)$ with $Q^*(s, a_g)$, we prove a Lipschitz continuity bound between $\hat{Q}_k^*(s_g, F_{s_\Delta}, a_g)$ and $\hat{Q}_{k'}^*(s_g, F_{s_{\Delta'}}, a_g)$ with respect to the TV distance measure between $s_\Delta \in \binom{s_{[n]}}{k}$ and $s_{\Delta'} \in \binom{s_{[n]}}{k'}$:

**Theorem D.1** (Lipschitz continuity in $\hat{Q}_k^*$). *For all $(s, a) \in \mathcal{S} \times \mathcal{A}$, $\Delta \in \binom{[n]}{k}$ and $\Delta' \in \binom{[n]}{k'}$,*

$$|\hat{Q}_k^*(s_g, F_{z_\Delta}, a_g) - \hat{Q}_{k'}^*(s_g, F_{z_{\Delta'}}, a_g)| \le \frac{2}{1 - \gamma} \|r_l(\cdot, \cdot)\|_\infty \cdot \text{TV}\left(F_{z_\Delta}, F_{z_{\Delta'}}\right)$$

We defer the proof of Theorem D.1 to Appendix E. See Figure 4 for a comparison between the $\hat{Q}_k^*$ learning and estimation process, and the exact $Q$-learning framework.

**Step 2: Bounding Total Variation (TV) Distance.** We bound the TV distance between $F_{z_\Delta}$ and $F_{z_{[n]}}$, where $\Delta \in \mathcal{U}\binom{[n]}{k}$. This task is equivalent to bounding the discrepancy between the empirical distribution and the distribution of the underlying finite population. When each $i \in \Delta$ is uniformly sampled *without* replacement, we use Lemma F.3 from Anand and Qu [2024] which generalizes the Dvoretzky-Kiefer-Wolfowitz (DKW) concentration inequality for empirical distribution functions.

Using this, we show:

$$\hat{Q}^0_{k,m}(s_g, F_{z_\Delta}, a_g) \qquad\qquad\qquad\qquad Q^*(s_g, s_{[n]}, a_g, a_{[n]})$$

$$(1) \Big\downarrow \qquad\qquad\qquad\qquad\qquad\qquad\qquad \Big\uparrow \overset{(4)}{=}$$

$$\hat{Q}^{\text{est}}_{k,m}(s_g, F_{z_\Delta}, a_g) \xrightarrow{(2)} \hat{Q}^*_k(s_g, F_{z_\Delta}, a_g) \xrightarrow[\approx]{(3)} \hat{Q}^*_n(s_g, F_{z_{[n]}}, a_g)$$

Figure 4: Flow of the algorithm and relevant analyses in learning $Q^*$. Here, (1) follows by performing Algorithm 1 (SUBSAMPLE-MFQ: Learning) on $\hat{Q}^0_{k,m}$. (2) follows from Lemma 4.3. (3) follows from the Lipschitz continuity and total variation distance bounds in Theorems D.1 and D.2. Finally, (4) follows from noting that $\hat{Q}^*_n = Q^*$.

**Theorem D.2.** *Given a finite population $\mathcal{Z} = (z_1, \ldots, z_n)$ for $\mathcal{Z} \in \mathcal{Z}^n_l$, let $\Delta \subseteq [n]$ be a uniformly random sample from $\mathcal{Z}$ of size $k$ chosen without replacement. Fix $\epsilon > 0$. Then, for all $x \in \mathcal{Z}_l$:*

$$\Pr\left[ \sup_{x \in \mathcal{Z}_l} \left| \frac{1}{|\Delta|} \sum_{i \in \Delta} \mathbb{1}\{z_i = x\} - \frac{1}{n} \sum_{i \in [n]} \mathbb{1}\{z_i = x\} \right| \le \epsilon \right] \ge 1 - 2|\mathcal{Z}_l| e^{-\frac{2kn\epsilon^2}{n-k+1}}.$$

Then, by Theorem D.2 and the definition of total variation distance from Section 2, we have that for $\delta \in (0, 1]$, with probability at least $1 - \delta$,

$$\text{TV}(F_{s_\Delta}, F_{s_{[n]}}) \le \sqrt{\frac{n-k+1}{8nk} \ln \frac{2|\mathcal{Z}_l|}{\delta}} \tag{26}$$

We then apply this result to our MARL setting by studying the rate of decay of the objective function between the learned policy $\pi^{\text{est}}_k$ and the optimal policy $\pi^*$ (Theorem 4.4).

**Step 3: Performance Difference Lemma to Complete the Proof.** As a consequence of the prior two steps and Lemma 4.3, $Q^*(s, a')$ and $\hat{Q}^{\text{est}}_{k,m}(s_g, F_{z_\Delta}, a'_g)$ become similar as $k \to n$. We further prove that the value generated by the policies $\pi^*$ and $\pi^{\text{est}}_k$ must also be very close (where the residue shrinks as $k \to n$). We then use the well-known performance difference lemma [Kakade and Langford, 2002] which we restate in Appendix G.1. A crucial theorem needed to use the performance difference lemma is a bound on $Q^*(s', \pi^*(s')) - Q^*(s', \hat{\pi}^{\text{est}}_k(s'_g, F_{s'_\Delta}))$.

Therefore, we formulate and prove Theorem D.3 which yields a probabilistic bound on this difference, where the randomness is over the choice of $\Delta \in \binom{[n]}{k}$:

**Theorem D.3.** *For a fixed $s' \in \mathcal{S} := \mathcal{S}_g \times \mathcal{S}^n_l$ and for $\delta \in (0, 1]$, with probability at least $1 - 2|\mathcal{A}_g| k^{|\mathcal{A}_l|} \delta$:*

$$Q^*(s', \pi^*(s')) - Q^*(s', \hat{\pi}^{\text{est}}_{k,m}(s'_g, F_{s'_\Delta})) \le \frac{2\|r_l(\cdot, \cdot)\|_\infty}{1-\gamma} \sqrt{\frac{n-k+1}{2nk} \ln\left(\frac{2|\mathcal{Z}_l|}{\delta}\right)} + 2\epsilon_{k,m}.$$

We defer the proof of Theorem D.3 and finding optimal value of $\delta$ to Lemma G.8 in the Appendix. Using Theorem D.3 and the performance difference lemma leads to Theorem 4.4.

# E Proof of Lipschitz-continuity Bound

This section proves a Lipschitz-continuity bound between $\hat{Q}^*_k$ and $Q^*$ and includes a framework to compare $\frac{1}{\binom{n}{k}} \sum_{\Delta \in \binom{[n]}{k}} \hat{Q}^*_k(s_g, s_\Delta, a_g)$ and $Q^*(s, a_g)$ in Lemma E.12.

Let $z_i := (s_i, a_i) \in \mathcal{Z}_l := \mathcal{S}_l \times \mathcal{A}_l$ and $z_\Delta = \{z_i : i \in \Delta\} \in \mathcal{Z}^k_l$. For $z_i = (s_i, a_i)$, let $z_i(s) = s_i$, and $z_i(a) = a_i$. With abuse of notation, note that $\hat{Q}^T_k(s_g, a_g, s_\Delta, a_\Delta)$ is equal to $\hat{Q}^T_k(s_g, a_g, z_\Delta)$.

The following definitions will be relevant to the proof of Theorem E.3.

**Definition E.1** (Empirical Distribution Function). *For all $z \in \mathcal{Z}^{|\Delta|}_l$ and $(s', a') \in \mathcal{S}_l \times \mathcal{A}_l$, where $\Delta \subseteq [n]$,*

$$F_{z_\Delta}(s', a') = \frac{1}{|\Delta|} \sum_{i \in \Delta} \mathbb{1}\{z_i(s) = s', z_i(a) = a'\}$$

**Definition E.2** (Total Variation Distance). Let $P$ and $Q$ be discrete probability distribution over some domain $\Omega$. Then,

$$\mathrm{TV}(P, Q) = \frac{1}{2} \|P - Q\|_1 = \sup_{E \subseteq \Omega} \left| \Pr_P(E) - \Pr_Q(E) \right|$$

**Theorem E.3** ($\hat{Q}_k^T$ is $\frac{2}{1-\gamma}\|r_l(\cdot, \cdot)\|_\infty$-Lipschitz continuous with respect to $F_{z_\Delta}$ in total variation distance). *Suppose $\Delta, \Delta' \subseteq [n]$ such that $|\Delta| = k$ and $|\Delta'| = k'$. Then:*

$$\left| \hat{Q}_k^T(s_g, a_g, F_{z_\Delta}) - \hat{Q}_{k'}^T(s_g, a_g, F_{z_{\Delta'}}) \right| \leq \left( \sum_{t=0}^{T-1} 2\gamma^t \right) \|r_l(\cdot, \cdot, \cdot)\|_\infty \cdot \mathrm{TV}\left( F_{z_\Delta}, F_{z_{\Delta'}} \right)$$

*Proof.* We prove this inductively. First, note that $\hat{Q}_k^0(\cdot, \cdot, \cdot) = \hat{Q}_{k'}^0(\cdot, \cdot, \cdot) = 0$ from the initialization step, which proves the lemma for $T = 0$, since $\mathrm{TV}(\cdot, \cdot) \geq 0$. At $T = 1$:

$$|\hat{Q}_k^1(s_g, a_g, F_{z_\Delta}) - \hat{Q}_{k'}^1(s_g, a_g, F_{z_{\Delta'}})| = \left| \hat{\mathcal{T}}_k \hat{Q}_k^0(s_g, a_g, F_{z_\Delta}) - \hat{\mathcal{T}}_{k'} \hat{Q}_{k'}^0(s_g, a_g, F_{z_{\Delta'}}) \right|$$

$$= \left| r(s_g, F_{s_\Delta}, a_g) + \gamma \mathbb{E}_{s_g', s_\Delta'} \max_{a_g' \in \mathcal{A}_g} \hat{Q}_k^0(s_g', F_{s_\Delta'}, a_g') \right.$$

$$\left. - r(s_g, F_{s_{\Delta'}}, a_g) - \gamma \mathbb{E}_{s_g', s_{\Delta'}'} \max_{a_g' \in \mathcal{A}_g} \hat{Q}_{k'}^0(s_g', F_{s_{\Delta'}'}, a_g') \right|$$

$$= |r(s_g, a_g, F_{z_\Delta}) - r(s_g, a_g, F_{z_{\Delta'}})|$$

$$= \left| \frac{1}{k} \sum_{i \in \Delta} r_l(s_g, z_i) - \frac{1}{k'} \sum_{i \in \Delta'} r_l(s_g, z_i) \right|$$

$$= \left| \mathbb{E}_{z_l \sim F_{z_\Delta}} r_l(s_g, z_l) - \mathbb{E}_{z_l' \sim F_{z_{\Delta'}}} r_l(s_g, z_l') \right|$$

In the first and second equalities, we use the time evolution property of $\hat{Q}_k^1$ and $\hat{Q}_{k'}^1$ by applying the adapted Bellman operators $\hat{\mathcal{T}}_k$ and $\hat{\mathcal{T}}_{k'}$ to $\hat{Q}_k^0$ and $\hat{Q}_{k'}^0$, respectively, and expanding. In the third and fourth equalities, we note that $\hat{Q}_k^0(\cdot, \cdot, \cdot) = \hat{Q}_{k'}^0(\cdot, \cdot, \cdot) = 0$, and subtract the common 'global component' of the reward function.

Then, noting the general property that for any function $f : \mathcal{X} \to \mathcal{Y}$ for $|\mathcal{X}| < \infty$ we can write $f(x) = \sum_{y \in \mathcal{X}} f(y) \mathbb{1}\{y = x\}$, we have:

$$|\hat{Q}_k^1(s_g, a_g, F_{z_\Delta}) - \hat{Q}_{k'}^1(s_g, a_g, F_{z_{\Delta'}})|$$

$$= \left| \mathbb{E}_{z_l \sim F_{z_\Delta}} \left[ \sum_{z \in \mathcal{Z}} r_l(s_g, z) \mathbb{1}\{z_l = z\} \right] - \mathbb{E}_{z_l' \sim F_{z_{\Delta'}}} \left[ \sum_{z \in \mathcal{Z}} r_l(s_g, z) \mathbb{1}\{z_l' = z\} \right] \right|$$

$$= \left| \sum_{z \in \mathcal{Z}} r_l(s_g, z) \cdot (\mathbb{E}_{z_l \sim F_{z_\Delta}} \mathbb{1}\{z_l = z\} - \mathbb{E}_{z_l' \sim F_{z_{\Delta'}}} \mathbb{1}\{z_l' = z\}) \right|$$

$$= \left| \sum_{z \in \mathcal{Z}} r_l(s_g, z) \cdot (F_{z_\Delta}(z) - F_{z_{\Delta'}}(z)) \right|$$

$$\leq \left| \max_{z \in \mathcal{Z}} r_l(s_g, z) \right| \cdot \sum_{z \in \mathcal{Z}} |F_{z_\Delta}(z) - F_{z_{\Delta'}}(z)|$$

$$\leq 2\|r_l(\cdot, \cdot)\|_\infty \cdot \mathrm{TV}(F_{z_\Delta}, F_{z_{\Delta'}})$$

The second equality follows from the linearity of expectations, and the third equality follows by noting that for any random variable $X \sim \mathcal{X}$, $\mathbb{E}_X \mathbb{1}[X = x] = \Pr[X = x]$. The first inequality follows from an application of the triangle inequality and the Cauchy-Schwarz inequality, and the second inequality uses the definition of TV distance. Thus, at $T = 1$, $\hat{Q}$ is $(2\|r_l(\cdot, \cdot)\|_\infty)$-Lipschitz continuous in TV distance, proving the base case.

Assume that for $T \leq t' \in \mathbb{N}$:

$$\left| \hat{Q}_k^T(s_g, a_g, F_{z_\Delta}) - \hat{Q}_{k'}^T(s_g, a_g, F_{z_{\Delta'}}) \right| \leq \left( \sum_{t=0}^{T-1} 2\gamma^t \right) \|r_l(\cdot, \cdot)\|_\infty \cdot \mathrm{TV}\left( F_{z_\Delta}, F_{z_{\Delta'}} \right)$$

Then, inductively:

$$|\hat{Q}_k^{T+1}(s_g, a_g, F_{z_\Delta}) - \hat{Q}_{k'}^{T+1}(s_g, a_g, F_{z_{\Delta'}})|$$

$$\leq \left| \frac{1}{|\Delta|} \sum_{i \in \Delta} r_l(s_g, z_i) - \frac{1}{|\Delta'|} \sum_{i \in \Delta'} r_l(s_g, z_i) \right|$$

$$+ \gamma \left| \mathbb{E}_{s'_g, s'_\Delta} \max_{\substack{a'_g \in \mathcal{A}_g, \\ a'_\Delta \in \mathcal{A}_l^k}} \hat{Q}_k^T(s'_g, a'_g, F_{z'_\Delta}) - \mathbb{E}_{s'_g, s'_{\Delta'}} \max_{\substack{a'_g \in \mathcal{A}_g, \\ a'_{\Delta'} \in \mathcal{A}_l^{k'}}} \hat{Q}_{k'}^T(s'_g, a'_g, F_{z'_{\Delta'}}) \right|$$

$$\leq 2\|r_l(\cdot, \cdot)\|_\infty \cdot \mathrm{TV}(F_{z_\Delta}, F_{z_{\Delta'}})$$

$$+ \gamma \left| \mathbb{E}_{(s'_g, s'_\Delta) \sim \mathcal{J}_k} \max_{\substack{a'_g \in \mathcal{A}_g, \\ a'_\Delta \in \mathcal{A}_l^k}} \hat{Q}_k^T(s'_g, a'_g, F_{z'_\Delta}) - \mathbb{E}_{(s'_g, s'_{\Delta'}) \sim \mathcal{J}_{k'}} \max_{\substack{a'_g \in \mathcal{A}_g, \\ a'_{\Delta'} \in \mathcal{A}_l^{k'}}} \hat{Q}_{k'}^T(s'_g, a'_g, F_{z'_{\Delta'}}) \right|$$

$$\leq 2\|r_l(\cdot, \cdot)\|_\infty \cdot \mathrm{TV}(F_{z_\Delta}, F_{z_{\Delta'}}) + \gamma \left( \sum_{\tau=0}^{T-1} 2\gamma^\tau \right) \|r_l(\cdot, \cdot)\|_\infty \cdot \mathrm{TV}(F_{z_\Delta}, F_{z_{\Delta'}})$$

$$= \left( \sum_{\tau=0}^{T} 2\gamma^\tau \right) \|r_l(\cdot, \cdot)\|_\infty \cdot \mathrm{TV}(F_{z_\Delta}, F_{z_{\Delta'}})$$

In the first inequality, we rewrite the expectations over the states as the expectation over the joint transition probabilities. The second inequality then follows from Lemma E.14. To apply it to Lemma E.14, we conflate the joint expectation over $(s_g, s_{\Delta \cup \Delta'})$ and reduce it back to the original form of its expectation. Finally, the third inequality follows from Lemma E.5.

By the inductive hypothesis, the claim is proven. $\qquad \square$

**Definition E.4.** [Joint Stochastic Kernels] The joint stochastic kernel on $(s_g, s_\Delta)$ for $\Delta \subseteq [n]$ where $|\Delta| = k$ is defined as $\mathcal{J}_k : \mathcal{S}_g \times \mathcal{S}_l^k \times \mathcal{S}_g \times \mathcal{A}_g \times \mathcal{S}_l^k \times \mathcal{A}_l^k \to [0, 1]$, where

$$\mathcal{J}_k(s'_g, s'_\Delta | s_g, a_g, s_\Delta, a_\Delta) := \Pr[(s'_g, s'_\Delta) | s_g, a_g, s_\Delta, a_\Delta] \tag{27}$$

**Lemma E.5.** *For all $T \in \mathbb{N}$, for any $a_g \in \mathcal{A}_g, s_g \in \mathcal{S}_g, s_\Delta \in \mathcal{S}_l^k, a_\Delta \in \mathcal{A}_l^k, a'_\Delta \in \mathcal{A}_l^k$, and for all joint stochastic kernels $\mathcal{J}_k$ as defined in Definition E.4:*

$$\left| \mathbb{E}_{(s'_g, s'_\Delta) \sim \mathcal{J}_k(\cdot, \cdot | s_g, a_g, s_\Delta, a_\Delta)} \max_{a'_g, a'_\Delta} \hat{Q}_k^T(s'_g, a'_g, F_{z'_\Delta}) \right.$$

$$\left. - \mathbb{E}_{(s'_g, s'_{\Delta'}) \sim \mathcal{J}_{k'}(\cdot, \cdot | s_g, a_g, s_{\Delta'}, a_{\Delta'})} \max_{a'_g, a'_{\Delta'}} \hat{Q}_{k'}^T(s'_g, a'_g, F_{z'_{\Delta'}}) \right| \leq \left( \sum_{\tau=0}^{T-1} 2\gamma^\tau \right) \|r_l(\cdot, \cdot)\|_\infty \mathrm{TV}\left( F_{z_\Delta}, F_{z_{\Delta'}} \right)$$

*Proof.* We prove this inductively. At $T = 0$, the statement is true since $\hat{Q}_k^0(\cdot, \cdot, \cdot) = \hat{Q}_{k'}^0(\cdot, \cdot, \cdot) = 0$ and $\mathrm{TV}(\cdot, \cdot) \geq 0$.

At $T = 1$,

$$\left| \mathbb{E}_{(s'_g, s'_\Delta) \sim \mathcal{J}_k(\cdot, \cdot | s_g, a_g, s_\Delta, a_\Delta)} \max_{a'_g, a'_\Delta} \hat{Q}_k^1(s'_g, a'_g, s'_\Delta, a'_\Delta) \right.$$

$$\left. - \mathbb{E}_{(s'_g, s'_{\Delta'}) \sim \mathcal{J}_{k'}(\cdot, \cdot | s_g, a_g, s_{\Delta'}, a_{\Delta'})} \max_{a'_g, a'_{\Delta'}} \hat{Q}_{k'}^1(s'_g, a'_g, s'_{\Delta'}, a'_{\Delta'}) \right|$$

$$= \left| \mathbb{E}_{(s'_g, s'_\Delta) \sim \mathcal{J}_k(\cdot, \cdot | s_g, a_g, s_\Delta, a_\Delta)} \max_{a'_g, a'_\Delta} \left[ r_g(s'_g, a'_g) + \frac{\sum_{i \in \Delta} r_l(s'_i, a'_i, s'_g)}{k} \right] \right.$$

$$\left. - \mathbb{E}_{(s'_g, s'_{\Delta'}) \sim \mathcal{J}_{k'}(\cdot, \cdot | s_g, a_g, s_{\Delta'}, a_{\Delta'})} \max_{a'_g, a'_{\Delta'}} \left[ r_g(s'_g, a'_g) + \frac{\sum_{i \in \Delta'} r_l(s'_i, a'_i, s'_g)}{k'} \right] \right|$$

$$= \left| \mathbb{E}_{(s'_g, s'_\Delta) \sim \mathcal{J}_k(\cdot, \cdot | s_g, a_g, s_\Delta, a_\Delta)} \max_{a'_\Delta} \frac{\sum_{i \in \Delta} r_l(s'_i, a'_i, s'_g)}{k} \right.$$

$$\left. - \mathbb{E}_{(s'_g, s'_{\Delta'}) \sim \mathcal{J}_{k'}(\cdot, \cdot | s_g, a_g, s_{\Delta'}, a_{\Delta'})} \max_{a'_{\Delta'}} \frac{\sum_{i \in \Delta'} r_l(s'_i, a'_i, s'_g)}{k'} \right|$$

$$= \left| \mathbb{E}_{(s'_g, s'_\Delta) \sim \mathcal{J}_k(\cdot, \cdot | s_g, a_g, s_\Delta, a_\Delta)} \frac{1}{k} \sum_{i \in \Delta} \tilde{r}_l(s'_i, s'_g) - \mathbb{E}_{(s'_g, s'_{\Delta'}) \sim \mathcal{J}_{k'}(\cdot, \cdot | s_g, a_g, s_{\Delta'}, a_{\Delta'})} \frac{1}{k'} \sum_{i \in \Delta'} \tilde{r}_l(s'_i, s'_g) \right|$$

In the last equality, we note that

$$\max_{a'_\Delta} \sum_{i \in \Delta} r_l(s'_i, a'_i, s'_g) = \sum_{i \in \Delta} \max_{a'_\Delta} r_l(s'_i, a'_i, s'_g) = \sum_{i \in \Delta} \max_{a'_i} r_l(s'_i, a'_i, s'_g) = \sum_{i \in \Delta} \tilde{r}_l(s'_i, s'_g),$$

where $\tilde{r}_l(s'_i, s'_g) := \max_{a'_i} r_l(s'_i, a'_i, s'_g)$.

Then, we have:

$$\left| \mathbb{E}_{(s'_g, s'_\Delta) \sim \mathcal{J}_k(\cdot, \cdot | s_g, a_g, s_\Delta, a_\Delta)} \frac{1}{k} \sum_{i \in \Delta} \tilde{r}_l(s'_i, s'_g) - \mathbb{E}_{(s'_g, s'_{\Delta'}) \sim \mathcal{J}_{k'}(\cdot, \cdot | s_g, a_g, s_{\Delta'}, a_{\Delta'})} \frac{1}{k'} \sum_{i \in \Delta'} \tilde{r}_l(s'_i, s'_g) \right|$$

$$= \left| \mathbb{E}_{(s'_g, s'_\Delta) \sim \mathcal{J}_k(\cdot, \cdot | s_g, a_g, s_\Delta, a_\Delta)} \frac{1}{k} \sum_{x \in \mathcal{S}_l} \tilde{r}_l(x, s'_g) F_{s'_\Delta}(x) \right.$$

$$\left. - \mathbb{E}_{(s'_g, s'_{\Delta'}) \sim \mathcal{J}_{k'}(\cdot, \cdot | s_g, a_g, s_{\Delta'}, a_{\Delta'})} \frac{1}{k'} \sum_{x \in \mathcal{S}_l} \tilde{r}_l(x, s'_g) F_{s'_{\Delta'}}(x) \right|$$

$$= \left| \mathbb{E}_{s'_g \sim \sum_{s'_{\Delta \cup \Delta'} \in \mathcal{S}_l^{|\Delta \cup \Delta'|}} \mathcal{J}_{|\Delta \cup \Delta'|}(\cdot, s'_{\Delta \cup \Delta'} | s_g, a_g, s_{\Delta \cup \Delta'}, a_{\Delta \cup \Delta'})} \right.$$

$$\left. \sum_{x \in \mathcal{S}_l} \tilde{r}_l(x, s'_g) \mathbb{E}_{s'_{\Delta \cup \Delta'} \sim \mathcal{J}_{|\Delta \cup \Delta'|}(\cdot | s'_g, s_g, a_g, s_{\Delta \cup \Delta'}, a_{\Delta \cup \Delta'})} [F_{s'_\Delta}(x) - F_{s'_{\Delta'}}(x)] \right|$$

$$\leq \|\tilde{r}_l(\cdot, \cdot)\|_\infty \cdot \mathbb{E}_{s'_g \sim \sum_{s'_{\Delta \cup \Delta'} \in \mathcal{S}_l^{|\Delta \cup \Delta'|}} \mathcal{J}_{|\Delta \cup \Delta'|}(\cdot, s'_{\Delta \cup \Delta'} | s_g, a_g, s_{\Delta \cup \Delta'}, a_{\Delta \cup \Delta'})}$$

$$\sum_{x \in \mathcal{S}_l} |\mathbb{E}_{s'_{\Delta \cup \Delta'} | s'_g} F_{s'_\Delta}(x) - \mathbb{E}_{s'_{\Delta \cup \Delta'} | s'_g} F_{s'_{\Delta'}}(x)|$$

$$\leq \|r_l(\cdot, \cdot)\|_\infty \cdot \mathbb{E}_{s'_g \sim \sum_{s'_{\Delta \cup \Delta'} \in \mathcal{S}_l^{|\Delta \cup \Delta'|}} \mathcal{J}_{|\Delta \cup \Delta'|}(\cdot, s'_{\Delta \cup \Delta'} | s_g, a_g, s_{\Delta \cup \Delta'}, a_{\Delta \cup \Delta'})}$$

$$\sum_{x \in \mathcal{S}_l} |\mathbb{E}_{s'_{\Delta \cup \Delta'} | s'_g} F_{s'_\Delta}(x) - \mathbb{E}_{s'_{\Delta \cup \Delta'} | s'_g} F_{s'_{\Delta'}}(x)|$$

$$\leq 2\|r_l(\cdot, \cdot)\|_\infty \cdot \mathbb{E}_{s'_g \sim \sum_{s'_{\Delta \cup \Delta'} \in \mathcal{S}_l^{|\Delta \cup \Delta'|}} \mathcal{J}_{|\Delta \cup \Delta'|}(\cdot, s'_{\Delta \cup \Delta'} | s_g, a_g, s_{\Delta \cup \Delta'}, a_{\Delta \cup \Delta'})}$$

$$\text{TV}(\mathbb{E}_{s'_{\Delta \cup \Delta'} | s'_g} F_{s'_\Delta}, \mathbb{E}_{s'_{\Delta \cup \Delta'} | s'_g} F_{s'_{\Delta'}})$$

$$\leq 2\|r_l(\cdot, \cdot)\|_\infty \cdot \text{TV}(F_{z_\Delta}, F_{z'_\Delta})$$

The first equality follows from noting that $f(x) = \sum_{x' \in \mathcal{X}} f(x') \mathbb{1}\{x = x'\}$ and from Fubini-Tonelli's inequality which allows us to swap the order of summations as the summand is finite. The second equality uses the law of total expectation. The first inequality uses Jensen's inequality and the triangle inequality. The second inequality uses $\|\tilde{r}_l(\cdot, \cdot)\|_\infty \leq \|r_l(\cdot, \cdot)\|_\infty$ which holds as $\|r_l\|_\infty$ is the infinite-norm of the local reward functions and is therefore atleast as large any other element in the image of $r_l$. The third inequality follows from the definition of total variation distance, and the final inequality follows from Lemma E.7. This proves the base case.

Then, assume that for $T \leq t' \in \mathbb{N}$, for all joint stochastic kernels $\mathcal{J}_k$ and $\mathcal{J}_{k'}$, and for all $a'_g \in \mathcal{A}_g, a'_\Delta \in \mathcal{A}_l^k$:

$$\left| \mathbb{E}_{(s'_g, s'_\Delta) \sim \mathcal{J}_k(\cdot, \cdot | s_g, a_g, s_\Delta, a_\Delta)} \max_{a'_g, a'_\Delta} \hat{Q}_k^T(s'_g, a'_g, s'_\Delta, a'_\Delta) - \right.$$

$$\left. \mathbb{E}_{(s'_g, s'_{\Delta'}) \sim \mathcal{J}_{k'}(\cdot, \cdot | s_g, a_g, s_{\Delta'}, a_{\Delta'})} \max_{a'_g, a'_{\Delta'}} \hat{Q}_{k'}^T(s'_g, a'_g, s'_{\Delta'}, a'_{\Delta'}) \right| \leq 2 \left( \sum_{t=0}^{T-1} \gamma^t \right) \|r_l(\cdot, \cdot)\|_\infty \text{TV}(F_{z_\Delta}, F_{z_{\Delta'}})$$

For the remainder of the proof, we adopt the shorthand $\mathbb{E}_{(s'_g, s'_\Delta) \sim \mathcal{J}}$ to denote $\mathbb{E}_{(s'_g, s'_\Delta) \sim \mathcal{J}_{|\Delta|}(\cdot, \cdot | s_g, a_g, s_\Delta, a_\Delta)}$, and $\mathbb{E}_{(s''_g, s''_\Delta) \sim \mathcal{J}}$ to denote $\mathbb{E}_{(s''_g, s''_\Delta) \sim \mathcal{J}_{|\Delta|}(\cdot, \cdot | s'_g, a'_g, s'_\Delta, a'_\Delta)}$.

Then, inductively, we have:

$$
\left| \mathbb{E}_{(s'_g, s'_\Delta) \sim \mathcal{J}_k(\cdot, \cdot | s_g, a_g, s_\Delta, a_\Delta)} \max_{a'_g, a'_\Delta} \hat{Q}_k^{T+1}(s'_g, a'_g, s'_\Delta, a'_\Delta) \right.
$$

$$
\left. - \mathbb{E}_{(s'_g, s'_{\Delta'}) \sim \mathcal{J}_{k'}(\cdot, \cdot | s_g, a_g, s_{\Delta'}, a_{\Delta'})} \max_{a'_g, a'_{\Delta'}} \hat{Q}_{k'}^{T+1}(s'_g, a'_g, s'_{\Delta'}, a'_{\Delta'}) \right| = \text{(I) - (II)},
$$

where

$$
\text{(I)} = \left| \mathbb{E}_{(s'_g, s'_\Delta) \sim \mathcal{J}_k(\cdot, \cdot | s_g, a_g, s_\Delta, a_\Delta)} \max_{a'_g, a'_\Delta} \left[ r_\Delta(s'_g, a'_g, s'_\Delta, a'_\Delta) \right. \right.
$$

$$
\left. \left. + \gamma \mathbb{E}_{(s''_g, s''_\Delta) \sim \mathcal{J}_k(\cdot, \cdot | s'_g, a'_g, s'_\Delta, a'_\Delta)} \max_{a''_g, a''_\Delta} \hat{Q}_k^T(s''_g, s''_\Delta, a''_g, a''_\Delta) \right] \right|
$$

and

$$
\text{(II)} = \left| \mathbb{E}_{(s'_g, s'_{\Delta'}) \sim \mathcal{J}_{k'}(\cdot, \cdot | s_g, a_g, s_{\Delta'}, a_{\Delta'})} \max_{a'_g, a'_{\Delta'}} \left[ r_{\Delta'}(s'_g, a'_g, s'_{\Delta'}, a'_{\Delta'}) \right. \right.
$$

$$
\left. \left. + \gamma \mathbb{E}_{(s''_g, s''_{\Delta'}) \sim \mathcal{J}_{k'}(\cdot, \cdot | s'_g, a'_g, s'_{\Delta'}, a'_{\Delta'})} \max_{a''_g, a''_{\Delta'}} \hat{Q}_{k'}^T(s''_g, s''_{\Delta'}, a''_g, a''_{\Delta'}) \right] \right|
$$

Let

$$
\tilde{a}_g, \tilde{a}_\Delta
$$

$$
= \underset{a'_g \in \mathcal{A}_g, a'_\Delta \in \mathcal{A}_l^k}{\arg \max} \left[ r_\Delta(s'_g, a'_g, s'_\Delta, a'_\Delta) + \gamma \mathbb{E}_{(s''_g, s''_\Delta) \sim \mathcal{J}_k(\cdot, \cdot | s'_g, a'_g, s'_\Delta, a'_\Delta)} \max_{a''_g, a''_\Delta} \hat{Q}_k^T(s''_g, s''_\Delta, a''_g, a''_\Delta) \right],
$$

and

$$
\hat{a}_g, \hat{a}_{\Delta'}
$$

$$
= \underset{a'_g \in \mathcal{A}_g, a'_{\Delta'} \in \mathcal{A}_l^k}{\arg \max} \left[ r_{\Delta'}(s'_g, a'_g, s'_{\Delta'}, a'_{\Delta'}) + \gamma \underset{(s''_g, s''_{\Delta'}) \sim \mathcal{J}_k(\cdot, \cdot | s'_g, a'_g, s'_{\Delta'}, a'_{\Delta'})}{\mathbb{E}} \max_{a''_g, a''_{\Delta'}} \hat{Q}_k^T(s''_g, s''_{\Delta'}, a''_g, a''_{\Delta'}) \right]
$$

Then, define $\tilde{a}_{\Delta \cup \Delta'} \in \mathcal{A}_l^{|\Delta \cup \Delta'|}$ by

$$
\tilde{a}_i = \begin{cases} \tilde{a}_i, & \text{if } i \in \Delta, \\ \hat{a}_i, & \text{if } i \in \Delta' \setminus \Delta \end{cases}
$$

Similarly, define $\hat{a}_{\Delta \cup \Delta'} \in \mathcal{A}_l^{|\Delta \cup \Delta'|}$ by

$$
\hat{a}_i = \begin{cases} \hat{a}_i, & i \in \Delta' \\ \tilde{a}_i, & i \in \Delta \setminus \Delta' \end{cases}
$$

Suppose (I) $\geq$ (II). Then,

$$
\left| \mathbb{E}_{(s'_g, s'_\Delta) \sim \mathcal{J}_k(\cdot, \cdot | s_g, a_g, s_\Delta, a_\Delta)} \max_{a'_g, a'_\Delta} \hat{Q}_k^{T+1}(s'_g, a'_g, s'_\Delta, a'_\Delta) \right.
$$

$$
\left. - \mathbb{E}_{(s'_g, s'_{\Delta'}) \sim \mathcal{J}_{k'}(\cdot, \cdot | s_g, a_g, s_{\Delta'}, a_{\Delta'})} \max_{a'_g, a'_{\Delta'}} \hat{Q}_{k'}^{T+1}(s'_g, a'_g, s'_{\Delta'}, a'_{\Delta'}) \right|
$$

$$= \left| \mathbb{E}_{(s'_g, s'_\Delta) \sim \mathcal{J}_k(\cdot, \cdot | s_g, a_g, s_\Delta, a_\Delta)} \hat{Q}_k^{T+1}(s'_g, \tilde{a}_g, s'_\Delta, \tilde{a}_\Delta) \right.$$

$$\left. - \mathbb{E}_{(s'_g, s'_{\Delta'}) \sim \mathcal{J}_{k'}(\cdot, \cdot | s_g, a_g, s_{\Delta'}, a_{\Delta'})} \hat{Q}_{k'}^{T+1}(s'_g, \hat{a}_g, s'_{\Delta'}, \hat{a}_{\Delta'}) \right|$$

$$\leq \left| \mathbb{E}_{(s'_g, s'_\Delta) \sim \mathcal{J}_k(\cdot, \cdot | s_g, a_g, s_\Delta, a_\Delta)} \hat{Q}_k^{T+1}(s'_g, \tilde{a}_g, s'_\Delta, \tilde{a}_\Delta) \right.$$

$$\left. - \mathbb{E}_{(s'_g, s'_{\Delta'}) \sim \mathcal{J}_{k'}(\cdot, \cdot | s_g, a_g, s_{\Delta'}, a_{\Delta'})} \hat{Q}_{k'}^{T+1}(s'_g, \tilde{a}_g, s'_{\Delta'}, \tilde{a}_{\Delta'}) \right|$$

$$= \left| \mathbb{E}_{(s'_g, s'_\Delta) \sim \mathcal{J}_k(\cdot, \cdot | s_g, a_g, s_\Delta, a_\Delta)} \left[ r_\Delta(s'_g, s'_\Delta, \tilde{a}_g, \tilde{a}_\Delta) \right. \right.$$

$$\left. + \gamma \mathbb{E}_{(s''_g, s''_\Delta) \sim \mathcal{J}_k(\cdot, \cdot | s'_g, \tilde{a}_g, s'_\Delta, \tilde{a}_\Delta)} \max_{a''_g, a''_\Delta} \hat{Q}_k^T(s''_g, a''_g, s''_\Delta, a''_\Delta) \right]$$

$$- \mathbb{E}_{(s'_g, s'_{\Delta'}) \sim \mathcal{J}_{k'}(\cdot, \cdot | s_g, a_g, s_{\Delta'}, a_{\Delta'})} \left[ r_{\Delta'}(s'_g, s'_{\Delta'}, \tilde{a}_g, \tilde{a}_{\Delta'}) \right.$$

$$\left. \left. + \gamma \mathbb{E}_{(s''_g, s''_{\Delta'}) \sim \mathcal{J}_{k'}(\cdot, \cdot | s'_g, \tilde{a}_g, s'_{\Delta'}, \tilde{a}_{\Delta'})} \max_{a''_g, a''_{\Delta'}} \hat{Q}_{k'}^T(s''_g, a''_g, s''_{\Delta'}, sa''_{\Delta'}) \right] \right|$$

$$\leq \left| \mathbb{E}_{(s'_g, s'_\Delta) \sim \mathcal{J}_k(\cdot, \cdot | s_g, a_g, s_\Delta, a_\Delta)} r_\Delta(s'_g, s'_\Delta, \tilde{a}_g, \tilde{a}_\Delta) \right.$$

$$\left. - \mathbb{E}_{(s'_g, s'_{\Delta'}) \sim \mathcal{J}_{k'}(\cdot, \cdot | s_g, a_g, s_{\Delta'}, a_{\Delta'})} r_{\Delta'}(s'_g, s'_{\Delta'}, \tilde{a}_g, \tilde{a}_{\Delta'}) \right|$$

$$+ \gamma \left| \mathbb{E}_{(s'_g, s'_\Delta) \sim \mathcal{J}_k(\cdot, \cdot | s_g, a_g, s_\Delta, a_\Delta)} \mathbb{E}_{(s''_g, s''_\Delta) \sim \mathcal{J}_k(\cdot, \cdot | s'_g, \tilde{a}_g, s'_\Delta, \tilde{a}_\Delta)} \max_{a''_g, a''_\Delta} \hat{Q}_k^T(s''_g, a''_g, s''_\Delta, a''_\Delta) \right.$$

$$\left. - \mathbb{E}_{(s'_g, s'_{\Delta'}) \sim \mathcal{J}_{k'}(\cdot, \cdot | s_g, a_g, s_{\Delta'}, a_{\Delta'})} \mathbb{E}_{(s''_g, s''_{\Delta'}) \sim \mathcal{J}_{k'}(\cdot, \cdot | s'_g, \tilde{a}_g, s'_{\Delta'}, \tilde{a}_{\Delta'})} \max_{a''_g, a''_{\Delta'}} \hat{Q}_{k'}^T(s''_g, a''_g, s''_{\Delta'}, a''_{\Delta'}) \right|$$

$$= \left| \mathbb{E}_{(s'_g, s'_\Delta) \sim \mathcal{J}_k(\cdot, \cdot | s_g, a_g, s_\Delta, a_\Delta)} \frac{1}{k} \sum_{i \in \Delta} r_l(s'_i, s'_g, \tilde{a}_i) \right.$$

$$\left. - \mathbb{E}_{(s'_g, s'_{\Delta'}) \sim \mathcal{J}_{k'}(\cdot, \cdot | s_g, a_g, s_{\Delta'}, a_{\Delta'})} \frac{1}{k'} \sum_{i \in \Delta'} r_l(s'_i, s'_g, \tilde{a}_i) \right|$$

$$+ \gamma \left| \mathbb{E}_{(s''_g, s''_\Delta) \sim \tilde{\mathcal{J}}_k(\cdot, \cdot | s_g, a_g, s_\Delta, a_\Delta)} \max_{a''_g, a''_\Delta} \hat{Q}_k^T(s''_g, a''_g, s''_\Delta, a''_\Delta) \right.$$

$$\left. - \mathbb{E}_{(s''_g, s''_{\Delta''}) \sim \tilde{\mathcal{J}}_{k'}(\cdot, \cdot | s_g, a_g, s_{\Delta'}, a_{\Delta'})} \max_{a''_g, a''_{\Delta'}} \hat{Q}_{k'}^T(s''_g, a''_g, s''_{\Delta'}, a''_{\Delta'}) \right|$$

$$\leq 2 \|r_l(\cdot, \cdot)\|_\infty \cdot \text{TV}(F_{z_\Delta}, F_{z_{\Delta'}}) + \gamma \left( \sum_{t=0}^T 2\gamma^t \right) \|r_l(\cdot, \cdot)\|_\infty \cdot \text{TV}(F_{z_\Delta}, F_{z_{\Delta'}})$$

$$= \left( \sum_{t=0}^{T+1} 2\gamma^t \right) \|r_l(\cdot, \cdot)\|_\infty \cdot \text{TV}(F_{z_\Delta}, F_{z_{\Delta'}})$$

The first equality rewrites the equations with their respective maximizing actions. The first inequality upper-bounds this difference by allowing all terms to share the common action $\tilde{a}$. Using the Bellman equation, the second equality expands $\hat{Q}_k^{T+1}$ and $\hat{Q}_{k'}^{T+1}$. The second inequality follows from the triangle inequality. The third equality follows by subtracting the common $r_g(s'_g, \tilde{a}_g)$ terms from the reward and noting that the two expectation terms $\mathbb{E}_{s'_g, s'_\Delta} \sim \mathcal{J}_k(\cdot, \cdot | s_g, a_g, s_\Delta, a_\Delta) \mathbb{E}_{s''_g, s_{\Delta''} \sim \mathcal{J}_k(\cdot, \cdot | s_g, \tilde{a}_g, s_\Delta, \tilde{a}_\Delta)}$ can be combined into a single expectation $\mathbb{E}_{s''_g, s''_\Delta \sim \tilde{\mathcal{J}}_k(\cdot, \cdot | s_g, a_g, s_\Delta, a_\Delta)}$.

In the special case where $a_\Delta = \tilde{a}_\Delta$, we derive a closed form expression for $\tilde{\mathcal{J}}_k$ in Lemma E.11. To justify the third inequality, the second term follows from the induction hypothesis and the first term

follows from the following derivation:

$$\left| \mathbb{E}_{(s'_g, s'_\Delta) \sim \mathcal{J}_k(\cdot, \cdot | s_g, a_g, s_\Delta, a_\Delta)} \frac{\sum_{i \in \Delta} r_l(s'_i, s'_g, \tilde{a}_i)}{k} - \mathbb{E}_{(s'_g, s'_{\Delta'}) \sim \mathcal{J}_{k'}(\cdot, \cdot | s_g, a_g, s_{\Delta'}, a_{\Delta'})} \frac{\sum_{i \in \Delta'} r_l(s'_i, s'_g, \tilde{a}_i)}{k'} \right|$$

$$= \left| \mathbb{E}_{(s'_g, s'_\Delta) \sim \mathcal{J}_k(\cdot, \cdot | s_g, a_g, s_\Delta, a_\Delta)} \frac{\sum_{i \in \Delta} r_l^{\tilde{a}}(s'_i, s'_g)}{k} - \mathbb{E}_{(s'_g, s'_{\Delta'}) \sim \mathcal{J}_{k'}(\cdot, \cdot | s_g, a_g, s_{\Delta'}, a_{\Delta'})} \frac{\sum_{i \in \Delta'} r_l^{\tilde{a}}(s'_i, s'_g)}{k'} \right|$$

$$\leq \|r_l^{\tilde{a}}\|_\infty \mathbb{E}_{s'_g \sim \sum_{s'_{\Delta \cup \Delta'} \in \mathcal{S}_l^{|\Delta \cup \Delta'|}} \mathcal{J}_{|\Delta \cup \Delta'|}(\cdot, s'_{\Delta \cup \Delta'} | s_g, a_g, s_{\Delta \cup \Delta'}, a_{\Delta \cup \Delta'})} \mathrm{TV}(\mathbb{E}_{s'_{\Delta \cup \Delta'} | s'_g} F_{s_\Delta}, \mathbb{E}_{s'_{\Delta \cup \Delta'} | s'_g} F_{s_{\Delta'}})$$

$$\leq 2\|r_l(\cdot, \cdot)\|_\infty \cdot \mathrm{TV}(F_{z_\Delta}, F_{z_{\Delta'}})$$

The above derivation follows from the same argument as in the base case where $r_l^{\tilde{a}}(s'_i, s'_g) := r_l(s'_i, s'_g, \tilde{a}_i)$ for any $i \in \Delta \cup \Delta'$. Similarly, if (I) < (II), an analogous argument that replaces $\tilde{a}_\Delta$ with $\hat{a}_\Delta$ yields the same result.

Therefore, by the induction hypothesis, the claim is proven. $\square$

**Remark E.6.** Given a joint transition probability function $\mathcal{J}_{|\Delta \cup \Delta'|}$ as defined in Definition E.4, we can recover the transition function for a single agent $i \in \Delta \cup \Delta'$ given by $\mathcal{J}_1$ using the law of total probability and the conditional independence between $s_i$ and $s_g \cup s_{[n] \setminus i}$ in Equation (28). This characterization is crucial in Lemma E.7 and Lemma E.5:

$$\mathcal{J}_1(\cdot | s'_g, s_g, a_g, s_i, a_i) = \sum_{s'_{\Delta \cup \Delta' \setminus i} \sim \mathcal{S}_l^{|\Delta \cup \Delta'| - 1}} \mathcal{J}_{|\Delta \cup \Delta'|}(s'_{\Delta \cup \Delta' \setminus i} \circ s'_i | s'_g, s_g, a_g, s_{\Delta \cup \Delta'}, a_{\Delta \cup \Delta'}) \quad (28)$$

Here, we use a conditional independence property proven in Lemma E.10.

**Lemma E.7.** The TV-distance between the next-step expected empirical distribution functions is bounded by the TV-distance between the existing empirical distribution functions.

$$\mathrm{TV}\left( \mathbb{E}_{s'_{\Delta \cup \Delta'} | s'_g \sim \mathcal{J}_{|\Delta \cup \Delta'|}(\cdot | s'_g, s_g, a_g, s_{\Delta \cup \Delta'}, a_{\Delta \cup \Delta'})} F_{s'_\Delta}, \mathbb{E}_{s'_{\Delta \cup \Delta'} | s'_g \sim \mathcal{J}_{|\Delta \cup \Delta'|}(\cdot | s'_g, s_g, a_g, s_{\Delta \cup \Delta'}, a_{\Delta \cup \Delta'})} F_{s'_{\Delta'}} \right)$$
$$\leq \mathrm{TV}(F_{z_\Delta}, F_{z_{\Delta'}})$$

*Proof.* From the definition of total variation distance, we have:

$$\mathrm{TV}\left( \mathbb{E}_{s'_{\Delta \cup \Delta'} | s'_g \sim \mathcal{J}_{|\Delta \cup \Delta'|}(\cdot | s'_g, s_g, a_g, s_{\Delta \cup \Delta'}, a_{\Delta \cup \Delta'})} F_{s'_\Delta}, \mathbb{E}_{s'_{\Delta \cup \Delta'} | s'_g \sim \mathcal{J}_{|\Delta \cup \Delta'|}(\cdot | s'_g, s_g, a_g, s_{\Delta \cup \Delta'}, a_{\Delta \cup \Delta'})} F_{s'_{\Delta'}} \right)$$

$$= \frac{1}{2} \sum_{x \in \mathcal{S}_l} \left| \mathbb{E}_{s'_{\Delta \cup \Delta'} | s'_g \sim \mathcal{J}_{|\Delta \cup \Delta'|}(\cdot | s'_g, s_g, a_g, s_{\Delta \cup \Delta'}, a_{\Delta \cup \Delta'})} F_{s'_\Delta}(x) \right.$$

$$\left. - \mathbb{E}_{s'_{\Delta \cup \Delta'} | s'_g \sim \mathcal{J}_{|\Delta \cup \Delta'|}(\cdot | s'_g, s_g, a_g, s_{\Delta \cup \Delta'}, a_{\Delta \cup \Delta'})} F_{s'_{\Delta'}}(x) \right|$$

$$= \frac{1}{2} \sum_{x \in \mathcal{S}_l} \left| \frac{1}{k} \sum_{i \in \Delta} \mathcal{J}_1(x | s'_g, s_g, a_g, s_i, a_i) - \frac{1}{k'} \sum_{i \in \Delta'} \mathcal{J}_1(x | s'_g, s_g, a_g, s_i, a_i) \right|$$

$$= \frac{1}{2} \sum_{x \in \mathcal{S}_l} \left| \frac{1}{k} \sum_{a_l \in \mathcal{A}_l} \sum_{s_l \in \mathcal{S}_l} \sum_{i \in \Delta} \mathcal{J}_1(x | s'_g, s_g, a_g, s_i, a_i) \mathbb{1} \left\{ {a_i = a_l \atop s_i = s_l} \right\} \right.$$

$$\left. - \frac{1}{k'} \sum_{a_l \in \mathcal{A}_l} \sum_{s_l \in \mathcal{S}_l} \sum_{i \in \Delta'} \mathcal{J}_1(x | s'_g, s_g, a_g, s_i, a_i) \mathbb{1} \left\{ {a_i = a_l \atop s_i = s_l} \right\} \right|$$

$$= \frac{1}{2} \sum_{x \in \mathcal{S}_l} \left| \frac{1}{k} \sum_{a_l \in \mathcal{A}_l} \sum_{s_l \in \mathcal{S}_l} \sum_{i \in \Delta} \mathcal{J}_1(x | s'_g, s_g, a_g, \cdot, \cdot) \vec{\mathbb{1}}_{\left\{ {a_i = a_l \atop s_i = s_l} \right\}} \right.$$

$$\left. - \frac{1}{k'} \sum_{a_l \in \mathcal{A}_l} \sum_{s_l \in \mathcal{S}_l} \sum_{i \in \Delta'} \mathcal{J}_1(x | s'_g, s_g, a_g, \cdot, \cdot) \vec{\mathbb{1}}_{\left\{ {a_i = a_l \atop s_i = s_l} \right\}} \right|$$

The second equality uses Lemma E.9. The third equality uses the property that each local agent can only have one state/action pair, and the fourth equality vectorizes the indicator variables. Then,

$$
\mathrm{TV}\Big(\mathbb{E}_{s'_{\Delta \cup \Delta'}|s'_g \sim \mathcal{J}_{|\Delta \cup \Delta'|}(\cdot|s'_g,s_g,a_g,s_{\Delta \cup \Delta'},a_{\Delta \cup \Delta'})}F_{s'_{\Delta}}, \mathbb{E}_{s'_{\Delta \cup \Delta'}|s'_g \sim \mathcal{J}_{|\Delta \cup \Delta'|}(\cdot|s'_g,s_g,a_g,s_{\Delta \cup \Delta'},a_{\Delta \cup \Delta'})}F_{s'_{\Delta'}}\Big)
$$

$$
\leq \frac{1}{2}\sum_{x \in \mathcal{S}_l}\sum_{a_l \in \mathcal{A}_l}\sum_{s_l \in \mathcal{S}_l}\left|\frac{1}{k}\sum_{i\in\Delta}\mathcal{J}_1(x|s'_g,s_g,a_g,\cdot,\cdot)\vec{\mathbb{I}}_{\{a_i=a_l \atop s_i=s_l\}} - \frac{1}{k'}\sum_{i\in\Delta'}\mathcal{J}_1(x|s'_g,s_g,a_g,\cdot,\cdot)\vec{\mathbb{I}}_{\{a_i=a_l \atop s_i=s_l\}}\right|
$$

$$
= \frac{1}{2}\sum_{a_l \in \mathcal{A}_l}\sum_{s_l \in \mathcal{S}_l}\left\|\frac{1}{k}\sum_{i\in\Delta}\mathcal{J}_1(\cdot|s'_g,s_g,a_g,\cdot,\cdot)\vec{\mathbb{I}}_{\{a_i=a_l \atop s_i=s_l\}} - \frac{1}{k'}\sum_{i\in\Delta'}\mathcal{J}_1(\cdot|s'_g,s_g,a_g,\cdot,\cdot)\vec{\mathbb{I}}_{\{a_i=a_l \atop s_i=s_l\}}\right\|_1
$$

$$
\leq \frac{1}{2}\sum_{a_l \in \mathcal{A}_l}\sum_{s_l \in \mathcal{S}_l}\left\|\frac{1}{k}\sum_{i\in\Delta}\vec{\mathbb{I}}_{\{a_i=a_l \atop s_i=s_l\}} - \frac{1}{k'}\sum_{i\in\Delta'}\vec{\mathbb{I}}_{\{a_i=a_l \atop s_i=s_l\}}\right\|_1 \cdot \|\mathcal{J}_1(\cdot|s'_g,s_g,a_g,\cdot,\cdot)\|_1
$$

$$
= \frac{1}{2}\sum_{a_l \in \mathcal{A}_l}\sum_{s_l \in \mathcal{S}_l}\left\|\frac{1}{k}\sum_{i\in\Delta}\mathbb{1}\{{}^{a_i=a_l}_{s_i=s_l}\} - \frac{1}{k'}\sum_{i\in\Delta'}\mathbb{1}\{{}^{a_i=a_l}_{s_i=s_l}\}\right\|_1
$$

$$
= \frac{1}{2}\sum_{(s_l,a_l)\in\mathcal{S}_l\times\mathcal{A}_l}\left|\frac{1}{k}\sum_{i\in\Delta}\mathbb{1}\{{}^{a_i=a_l}_{s_i=s_l}\} - \frac{1}{k'}\sum_{i\in\Delta'}\mathbb{1}\{{}^{a_i=a_l}_{s_i=s_l}\}\right|
$$

$$
= \frac{1}{2}\sum_{z_l \in \mathcal{S}_l\times\mathcal{A}_l}\left|F_{z_\Delta}(z_l)-F_{z_{\Delta'}}(z_l)\right|
$$

$$
:= \mathrm{TV}(F_{z_\Delta}, F_{z_{\Delta'}})
$$

The first inequality uses the triangle inequality. The second inequality and fifth equality follow from Hölder's inequality and the sum of the probabilities from the stochastic transition function $\mathcal{J}_1$ being equal to 1. The sixth equality uses Fubini-Tonelli's theorem which applies as the total variation distance measure is bounded from above by 1. The final equality recovers the total variation distance for the variable $z = (s,a) \in \mathcal{S}_l \times \mathcal{A}_l$ across agents $\Delta$ and $\Delta'$, which proves the claim. $\square$

Next, recall the data processing inequality.

**Lemma E.8** (Data Processing Inequality.). *Let $A$ and $B$ be random variables over some domain $S$. Let $f$ be some function (not necessarily deterministically) mapping from $S$ to any codomain $T$. Then, every $f$-divergence $\chi$ satisfies*

$$
\chi(f(A)\|f(B)) \leq \chi(A\|B)
$$

**Remark:** We show an analog of the data processing inequality. Under this lens, Lemma E.7 saturates the data-processing relation for the TV distance in our multi-agent setting.

**Lemma E.9.**

$$
\mathbb{E}_{s'_{\Delta \cup \Delta'}|s'_g \sim \mathcal{J}_{|\Delta \cup \Delta'|}(\cdot|s_g,a_g,s_{\Delta \cup \Delta'},a_{\Delta \cup \Delta'})}F_{s'_{\Delta}}(x) = \frac{1}{k}\sum_{i\in\Delta}\mathcal{J}_1(x|s'_g,s_g,a_g,s_i,a_i)
$$

*Proof.* By expanding on the definition of the empirical distribution function $F_{s_{\Delta'}}(x) = \frac{1}{|\Delta|}\sum_{i\in\Delta}\mathbb{1}\{s_i = x\}$, we have:

$$
\mathbb{E}_{s_{\Delta \cup \Delta'}|s'_g \sim \mathcal{J}_{|\Delta \cup \Delta'|}(\cdot|s'_g,s_g,a_g,s_{\Delta \cup \Delta'},a_{\Delta \cup \Delta'})}F_{s'_{\Delta}}(x)
$$

$$
= \frac{1}{k}\sum_{i\in\Delta}\mathbb{E}_{s_{\Delta \cup \Delta'}|s'_g \sim \mathcal{J}_{|\Delta \cup \Delta'|}(\cdot|s'_g,s_g,a_g,s_{\Delta \cup \Delta'},a_{\Delta \cup \Delta'})}\mathbb{1}\{s'_i = x\}
$$

$$
= \frac{1}{k}\sum_{i\in\Delta}\mathbb{E}_{s_{\Delta \cup \Delta'}|s'_g \sim \mathcal{J}_1(\cdot|s'_g,s_g,a_g,s_i,a_i)}\mathbb{1}\{s'_i = x\}
$$

$$
= \frac{1}{k}\sum_{i\in\Delta}\mathcal{J}_1(x|s'_g,s_g,a_g,s_i,a_i)
$$

The second equality follows from the conditional independence of $s_i'$ from $s_{\Delta \cup \Delta' \setminus i}$ from Lemma E.10, and the final equality uses the fact that the expectation of an indicator random variable is the probability distribution function of the random variable. $\qquad \square$

**Lemma E.10.** *The distribution $s_{\Delta \cup \Delta' \setminus i}' | s_g', s_g, a_g, s_{\Delta \cup \Delta'}, a_{\Delta \cup \Delta'}$ is conditionally independent to the distribution $s_i' | s_g', s_g, a_g, s_i, a_i$ for any $i \in \Delta \cup \Delta'$.*

*Proof.* We direct the interested reader to the Bayes-Ball theorem in Shachter [2013] for proving conditional independence. For ease of exposition, we restate the two rules in Shachter [2013] that introduce the notion of $d$-separations which implies conditional independence. Suppose we have a causal graph $G = (V, E)$ where the vertex set $V = [p]$ for $p \in \mathbb{N}$ is a set of variables and the edge set $E \subseteq 2^V$ denotes dependence through connectivity. Then, the two rules that establish $d$-separations are as follows:

1. For $x, y \in V$, we say that $x, y$ are $d$-connected if there exists a path $(x, \dots, y)$ that can be traced without traversing a pair of arrows that point at the same vertex.

2. We say that $x, y \in V$ are $d$-connected, conditioned on a set of variables $Z \subseteq V$, if there is a path $(x, \dots, y)$ that does not contain an event $z \in Z$ that can be traced without traversing a pair of arrows that point at the same vertex.

If $x, y \in V$ is not $d$-connected through any such path, then $x, y$ is $d$-separated, which implies conditional independence.

Let $Z = \{s_g', s_g, a_g, s_i, a_i\}$ be the set of variables we condition on. Then, the below figure demonstrates the causal graph for the events of interest.

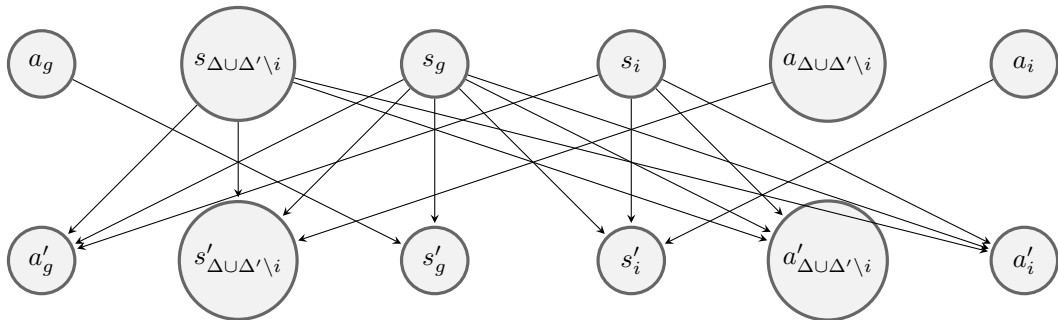

Figure 5: Causal graph to demonstrate the dependencies between variables.

1. Observe that all paths (through undirected edges) stemming from $s_{\Delta \cup \Delta' \setminus i}' \to s_{\Delta \cup \Delta' \setminus i}$ pass through $s_i \in Z$ which is blocked.

2. All other paths from $s_{\Delta \cup \Delta' \setminus i}'$ to $s_i'$ pass through $s_g \cup s_g' \in Z$.

Therefore, $s_{\Delta \cup \Delta' \setminus i}'$ and $s_i'$ are $d$-separated by $Z$. Hence, by Shachter [2013], $s_{\Delta \cup \Delta' \setminus i}'$ and $s_i'$ are conditionally independent. $\qquad \square$

**Lemma E.11.** *For any joint transition probability function $\mathcal{J}_k$ on $s_g \in \mathcal{S}_g, s_\Delta \in \mathcal{S}_l^k$, $a_\Delta \in \mathcal{A}_l^k$ where $|\Delta| = k$, given by $\mathcal{J}_k : \mathcal{S}_g \times \mathcal{S}_l^{|\Delta|} \times \mathcal{S}_g \times \mathcal{A}_g \times \mathcal{S}_l^{|\Delta|} \times \mathcal{A}_l^{|\Delta|} \to [0, 1]$, we have:*

$$\mathbb{E}_{(s_g', s_\Delta') \sim \mathcal{J}_k(\cdot, \cdot | s_g, a_g, s_\Delta, a_\Delta)} \left[ \mathbb{E}_{(s_g'', s_\Delta'') \sim \mathcal{J}_k(\cdot, \cdot | s_g', a_g, s_\Delta', a_\Delta)} \max_{a_g'' \in \mathcal{A}_g, a_\Delta'' \in \mathcal{A}_l^k} \hat{Q}_k^T(s_g'', s_\Delta'', a_g'', a_\Delta'') \right]$$

$$= \mathbb{E}_{(s_g'', s_\Delta'') \sim \mathcal{J}_k^2(\cdot, \cdot | s_g, a_g, s_\Delta, a_\Delta)} \max_{a_g'' \in \mathcal{A}_g} \hat{Q}_k^T(s_g'', s_\Delta'', a_g'', a_\Delta'')$$

*Proof.* By expanding the expectations:

$$\mathbb{E}_{(s_g', s_\Delta') \sim \mathcal{J}_k(\cdot, \cdot | s_g, a_g, s_\Delta, a_\Delta)} \left[ \mathbb{E}_{(s_g'', s_\Delta'') \sim \mathcal{J}_k(\cdot, \cdot | s_g', a_g, s_\Delta', a_\Delta)} \max_{a_g'' \in \mathcal{A}_g, a_\Delta'' \in \mathcal{A}_l^k} \hat{Q}_k^T(s_g'', s_\Delta'', a_g'', a_\Delta'') \right]$$

$$= \sum_{(s_g', s_\Delta') \in \mathcal{S}_g \times \mathcal{S}_l^{|\Delta|}} \sum_{(s_g'', s_\Delta'') \in \mathcal{S}_g \times \mathcal{S}_l^{|\Delta|}} \mathcal{J}_k[s_g', s_\Delta', s_g, a_g, s_\Delta, a_\Delta] \mathcal{J}_k[s_g'', s_\Delta'', s_g', a_g, s_\Delta', a_\Delta]$$

$$\max_{a_g'' \in \mathcal{A}_g, a_\Delta'' \in \mathcal{A}_l^k} \hat{Q}_k^T(s_g'', s_\Delta'', a_g'', a_\Delta'')$$

$$= \sum_{(s_g'', s_\Delta'') \in \mathcal{S}_g \times \mathcal{S}_l^{|\Delta|}} \mathcal{J}_k^2[s_g'', s_\Delta'', s_g, a_g, s_\Delta, a_\Delta] \max_{a_g'' \in \mathcal{A}_g, a_\Delta'' \in \mathcal{A}_l^k} \hat{Q}_k^T(s_g'', s_\Delta'', a_g'', a_\Delta'')$$

$$= \mathbb{E}_{(s_g'', s_\Delta'') \sim \mathcal{J}_k^2(\cdot, \cdot | s_g, a_g, s_\Delta, a_\Delta)} \max_{a_g'' \in \mathcal{A}_g, a_\Delta'' \in \mathcal{A}_l^k} \hat{Q}_k^T(s_g'', s_\Delta'', a_g'', a_\Delta'')$$

In the second equality, the right-stochasticity of $\mathcal{J}_k$ implies the right-stochasticity of $\mathcal{J}_k^2$.

Further, observe that $\mathcal{J}_k[s_g', s_\Delta', s_g, a_g, s_\Delta, a_\Delta] \mathcal{J}_k[s_g'', s_\Delta'', s_g', a_g, s_\Delta', a_\Delta]$ denotes the probability of the transitions $(s_g, s_\Delta) \to (s_g', s_\Delta') \to (s_g'', s_\Delta'')$ with actions $a_g, a_\Delta$ at each step, where the joint state evolution is governed by $\mathcal{J}_k$. Thus,

$$\sum_{(s_g', s_\Delta') \in \mathcal{S}_g \times \mathcal{S}_l^{|\Delta|}} \mathcal{J}_k[s_g', s_\Delta', s_g, a_g, s_\Delta, a_\Delta] \mathcal{J}_k[s_g'', s_\Delta'', s_g', a_g, s_g', a_\Delta] = \mathcal{J}_k^2[s_g'', s_\Delta'', s_g, a_g, s_\Delta, a_\Delta]$$

since $\sum_{(s_g', s_\Delta') \in \mathcal{S}_g \times \mathcal{S}_l^{|\Delta|}} \mathcal{J}_k[s_g', s_\Delta', s_g, a_g, s_\Delta, a_\Delta] \mathcal{J}_k[s_g'', s_\Delta'', s_g', a_g, s_g', a_\Delta]$ is the stochastic probability function corresponding to the two-step evolution of the joint states from $(s_g, s_\Delta)$ to $(s_g'', s_\Delta'')$ under actions $a_g, a_\Delta$. This can be thought of as an analogously to the fact that a 1 on the $(i,j)$'th entry on the square of a $0/1$ adjacency matrix of a graph represents the fact that there is a path of distance 2 between vertices $i$ and $j$.

Finally, the third equality recovers the definition of the expectation, with respect to the joint probability function $\mathcal{J}_k^2$. $\qquad\square$

We next show $\lim_{T \to \infty} \mathbb{E}_{\Delta \in \binom{[n]}{k}} \hat{Q}_k^T(s_g, s_\Delta, a_g, a_\Delta) = \mathbb{E}_{\Delta \in \binom{[n]}{k}} \hat{Q}_k^*(s_g, s_\Delta, a_g, a_\Delta) = Q^*(s, a)$.

**Lemma E.12.** *The $Q^*$ function is the average value of each of the $\binom{n}{k}$ sub-sampled $\hat{Q}_k^*$-functions.*

$$Q^*(s, a) - \frac{1}{\binom{n}{k}} \sum_{\Delta \in \binom{[n]}{k}} \hat{Q}_k^T(s_g, s_\Delta, a_g, a_\Delta) \leq \gamma^T \cdot \frac{\tilde{r}}{1 - \gamma}$$

*Proof.* We bound the differences between $\hat{Q}_k^T$ at each Bellman iteration of our approximation to $Q^*$. Note that:

$$Q^*(s, a) - \frac{1}{\binom{n}{k}} \sum_{\Delta \in \binom{[n]}{k}} \hat{Q}_k^T(s_g, s_\Delta, a_g, a_\Delta)$$

$$= \mathcal{T} Q^*(s, a) - \frac{1}{\binom{n}{k}} \sum_{\Delta \in \binom{[n]}{k}} \hat{\mathcal{T}}_k \hat{Q}_k^{T-1}(s_g, s_\Delta, a_g, a_\Delta)$$

$$= r_{[n]}(s_g, s_{[n]}, a_g) + \gamma \mathbb{E}_{\substack{s_g' \sim P_g(\cdot | s_g, a_g), \\ s_i' \sim P_l(\cdot | s_i, a_i, s_g), \forall i \in [n]}} \max_{a_g' \in \mathcal{A}_g, a_{[n]}' \in \mathcal{A}_l^n} Q^*(s', a')$$

$$- \frac{1}{\binom{n}{k}} \sum_{\Delta \in \binom{[n]}{k}} \left[ r_\Delta(s_g, s_\Delta, a_g, a_\Delta) + \gamma \mathbb{E}_{\substack{s_g' \sim P_g(\cdot | s_g, a_g) \\ s_i' \sim P_l(\cdot | s_i, a_i s_g), \forall i \in \Delta}} \max_{a_g' \in \mathcal{A}_g, a_\Delta' \in \mathcal{A}_l^k} Q^T(s_g', s_\Delta', a_g', a_\Delta') \right]$$

Next, observe that $r_{[n]}(s_g, s_{[n]}, a_g, a_{[n]}) = \frac{1}{\binom{n}{k}} \sum_{\Delta \in \binom{[n]}{k}} r_{[\Delta]}(s_g, s_\Delta, a_g, a_\Delta)$.

To prove this, we write:

$$\frac{1}{\binom{n}{k}} \sum_{\Delta \in \binom{[n]}{k}} r_{[\Delta]}(s_g, s_\Delta, a_g, a_\Delta) = \frac{1}{\binom{n}{k}} \sum_{\Delta \in \binom{[n]}{k}} \left( r_g(s_g, a_g) + \frac{1}{k} \sum_{i \in \Delta} r_l(s_i, a_i, s_g) \right)$$

$$= r_g(s_g, a_g) + \frac{\binom{n-1}{k-1}}{k\binom{n}{k}} \sum_{i \in [n]} r_l(s_i, a_i, s_g)$$

$$= r_g(s_g, a_g) + \frac{1}{n} \sum_{i \in [n]} r_l(s_i, a_i, s_g) := r_{[n]}(s_g, s_{[n]}, a_g, a_{[n]})$$

In the second equality, we reparameterized the sum to count the number of times each $r_l(s_i, s_g)$ was added for each $i \in \Delta$. To do this, we count the number of $(k-1)$ other agents that could form a $k$-tuple with agent $i$, and there are $(n-1))$ candidates from which we chooses the $(k-1)$ agents. In the last equality, we expanded and simplified the binomial coefficients.

So, we have that:

$$\sup_{(s,a) \in \mathcal{S} \times \mathcal{A}} \left[ Q^*(s,a) - \frac{1}{\binom{n}{k}} \sum_{\Delta \in \binom{[n]}{k}} \hat{Q}_k^T(s_g, s_{[n]}, a_g, a_\Delta) \right]$$

$$= \sup_{(s,a) \in \mathcal{S} \times \mathcal{A}} \left[ \mathcal{T} Q^*(s,a) - \frac{1}{\binom{n}{k}} \sum_{\Delta \in \binom{[n]}{k}} \hat{\mathcal{T}}_k \hat{Q}_k^{T-1}(s_g, s_\Delta, a_g, a_\Delta) \right]$$

$$= \gamma \sup_{(s,a) \in \mathcal{S} \times \mathcal{A}} \left[ \mathbb{E}_{\substack{s_g' \sim P(\cdot | s_g, a_g), \\ s_i' \sim P_l(\cdot | s_i, a_i, s_g), \forall i \in [n]}} \max_{a' \in \mathcal{A}} Q^*(s', a') \right.$$

$$\left. - \frac{1}{\binom{n}{k}} \sum_{\Delta \in \binom{[n]}{k}} \mathbb{E}_{\substack{s_g' \sim P_g(\cdot | s_g, a_g), \\ s_i' \sim P_l(\cdot | s_i, a_i, s_g), \forall i \in \Delta}} \max_{\substack{a_g' \in \mathcal{A}_g, \\ a_\Delta' \in \mathcal{A}_l^k}} \hat{Q}_k^{T-1}(s_g', s_\Delta', a_g', a_\Delta') \right]$$

$$= \gamma \sup_{(s,a) \in \mathcal{S} \times \mathcal{A}} \mathbb{E}_{\substack{s_g' \sim P_g(\cdot | s_g, a_g), \\ s_i' \sim P_l(\cdot | s_i, a_i, s_g), \forall i \in [n]}} \left[ \max_{a' \in \mathcal{A}} Q^*(s', a') \right.$$

$$\left. - \frac{1}{\binom{n}{k}} \sum_{\Delta \in \binom{[n]}{k}} \max_{a_g' \in \mathcal{A}_g, a_\Delta' \in \mathcal{A}_l^k} \hat{Q}_k^{T-1}(s_g', s_\Delta', a_g', a_\Delta') \right]$$

$$\leq \gamma \sup_{(s,a) \in \mathcal{S} \times \mathcal{A}} \mathbb{E}_{\substack{s_g' \sim P_g(\cdot | s_g, a_g), \\ s_i' \sim P_l(\cdot | s_i, a_i, s_g), \forall i \in [n]}}$$

$$\max_{a_g' \in \mathcal{A}_g, a_{[n]}' \in \mathcal{A}_l^n} \left[ Q^*(s', a') - \frac{1}{\binom{n}{k}} \sum_{\Delta \in \binom{[n]}{k}} \hat{Q}_k^{T-1}(s_g', s_\Delta', a_g', a_\Delta') \right]$$

$$\leq \gamma \sup_{(s',a') \in \mathcal{S} \times \mathcal{A}} \left[ Q^*(s', a') - \frac{1}{\binom{n}{k}} \sum_{\Delta \in \binom{[n]}{k}} \hat{Q}_k^{T-1}(s_g', s_\Delta', a_g', a_\Delta') \right]$$

We justify the first inequality by noting the general property that for positive vectors $v, v'$ for which $v \succeq v'$ which follows from the triangle inequality:

$$\left\| v - \frac{1}{\binom{n}{k}} \sum_{\Delta \in \binom{[n]}{k}} v' \right\|_\infty \geq \left| \|v\|_\infty - \left\| \frac{1}{\binom{n}{k}} \sum_{\Delta \in \binom{[n]}{k}} v' \right\|_\infty \right|$$

$$= \|v\|_\infty - \left\| \frac{1}{\binom{n}{k}} \sum_{\Delta \in \binom{[n]}{k}} v' \right\|_\infty$$

$$\geq \|v\|_\infty - \frac{1}{\binom{n}{k}} \sum_{\Delta \in \binom{[n]}{k}} \|v'\|_\infty$$

Thus, applying this bound recursively, we get:

$$Q^*(s,a) - \frac{1}{\binom{n}{k}} \sum_{\Delta \in \binom{[n]}{k}} \hat{Q}_k^T(s_g, s_\Delta, a_g, a_\Delta)$$

$$\leq \gamma^T \sup_{(s',a') \in \mathcal{S} \times \mathcal{A}} \left[ Q^*(s',a') - \frac{1}{\binom{n}{k}} \sum_{\Delta \in \binom{[n]}{k}} \hat{Q}_k^0(s_g', s_\Delta', a_g', a_\Delta') \right]$$

$$= \gamma^T \sup_{(s',a') \in \mathcal{S} \times \mathcal{A}} Q^*(s',a')$$

$$= \gamma^T \cdot \frac{\tilde{r}}{1-\gamma}$$

The first inequality follows from the $\gamma$-contraction property of the update procedure, and the ensuing equality follows from our bound on the maximum possible value of $Q$ from Lemma C.4 and noting that $\hat{Q}_k^0 := 0$.

Therefore, as $T \to \infty$,

$$Q^*(s, a_g) - \frac{1}{\binom{n}{k}} \sum_{\Delta \in \binom{[n]}{k}} \hat{Q}_k^T(s_g, s_\Delta, a_g) \to 0,$$

which proves the lemma. $\qquad\square$

**Corollary E.13.** *Since* $\frac{1}{\binom{n}{k}} \sum_{\Delta \in \binom{[n]}{k}} \hat{Q}_k^T(s_g, s_\Delta, a_g, a_\Delta) := \mathbb{E}_{\Delta \in \binom{[n]}{k}} \hat{Q}_k^T(s_g, s_\Delta, a_g, a_\Delta)$, *we therefore get:*

$$Q^*(s,a) - \lim_{T \to \infty} \mathbb{E}_{\Delta \in \binom{[n]}{k}} \hat{Q}_k^T(s_g, s_\Delta, a_g, a_\Delta) \leq \lim_{T \to \infty} \gamma^T \cdot \frac{\tilde{r}}{1-\gamma}$$

*Consequently,*

$$Q^*(s,a) - \mathbb{E}_{\Delta \in \binom{[n]}{k}} \hat{Q}_k^*(s_g, s_\Delta, a_g, a_\Delta) \leq 0$$

*Further, we have that*

$$Q^*(s,a) - \mathbb{E}_{\Delta \in \binom{[n]}{k}} \hat{Q}_k^*(s_g, s_\Delta, a_g, a_\Delta) = 0,$$

*since* $\hat{Q}_k^*(s_g, s_\Delta, a_g, a_\Delta) \leq Q^*(s,a)$.

**Lemma E.14.** *The absolute difference between the expected maximums between $\hat{Q}_k$ and $\hat{Q}_{k'}$ is atmost the maximum of the absolute difference between $\hat{Q}_k$ and $\hat{Q}_{k'}$, where the expectations are taken over any joint distributions of states $\mathcal{J}$, and the maximums are taken over the actions.*

$$\left| \mathbb{E}_{(s_g', s_{\Delta \cup \Delta'}') \sim \mathcal{J}_{|\Delta \cup \Delta'|}(\cdot, \cdot | s_g, a_g, s_{\Delta \cup \Delta'}, a_{\Delta \cup \Delta'})} \left[ \max_{a_g' \in \mathcal{A}_g, a_\Delta' \in \mathcal{A}_l^k} \hat{Q}_k^T(s_g', s_\Delta', a_g', a_\Delta') \right. \right.$$

$$\left. \left. - \max_{a_g' \in \mathcal{A}_g, a_{\Delta'}' \in \mathcal{A}_l^{k'}} \hat{Q}_{k'}^T(s_g', s_{\Delta'}', a_g', a_{\Delta'}') \right] \right|$$

$$\leq \max_{a_g' \in \mathcal{A}_g, a_{\Delta \cup \Delta'}' \in \mathcal{A}_l^{|\Delta \cup \Delta'|}} \left| \mathbb{E}_{(s_g', s_{\Delta \cup \Delta'}') \sim \mathcal{J}_{|\Delta \cup \Delta'|}(\cdot, \cdot | s_g, a_g, s_{\Delta \cup \Delta'}, a_{\Delta \cup \Delta'})} \left[ \hat{Q}_k^T(s_g', s_\Delta', a_g', a_\Delta') \right. \right.$$

$$\left. \left. - \hat{Q}_{k'}^T(s_g', s_{\Delta'}', a_g', a_{\Delta'}') \right] \right|$$

*Proof.* Denote:

$$a_g^*, a_\Delta^* := \arg\max_{\substack{a_g' \in \mathcal{A}_g, \\ a_\Delta' \in \mathcal{A}_l^k}} \hat{Q}_k^T(s_g', F_{s_\Delta'}, a_g', a_\Delta')$$

$$\tilde{a}_g^*, \tilde{a}_{\Delta'}^* := \arg\max_{\substack{a_g' \in \mathcal{A}_g, \\ a_{\Delta'}' \in \mathcal{A}_l^{k'}}} \hat{Q}_{k'}^T(s_g', F_{s_{\Delta'}'}, a_g', a_{\Delta'}')$$

We extend $a^*_\Delta$ to $a^*_{\Delta'}$ by letting $a^*_{\Delta'\setminus\Delta}$ be the corresponding $\Delta\setminus\Delta'$ variables from $\tilde{a}^*_{\Delta'}$ in $\mathcal{A}_l^{|\Delta'\setminus\Delta|}$. For the remainder of this proof, we adopt the shorthand $\mathbb{E}_{s'_g,s'_{\Delta\cup\Delta'}}$ to refer to $\mathbb{E}_{(s'_g,s'_{\Delta\cup\Delta'})\sim\mathcal{J}_{|\Delta\cup\Delta'|}(\cdot,\cdot|s_g,a_g,s_{\Delta\cup\Delta'},a_{\Delta\cup\Delta'})}$. Then, if $\mathbb{E}_{s'_g,s'_{\Delta\cup\Delta'}}\max_{a'_g\in\mathcal{A}_g,a'_\Delta\in\mathcal{A}_l^k}\hat{Q}_k^T(s'_g,F_{z'_\Delta},a'_g)-\mathbb{E}_{s'_g,s'_{\Delta\cup\Delta'}}\max_{a'_g\in\mathcal{A}_g,a'_{\Delta'}\in\mathcal{A}_l^{k'}}\hat{Q}_{k'}^T(s'_g,F_{z'_{\Delta'}},a'_g)>0$, we have:

$$\left|\mathbb{E}_{s'_g,s'_{\Delta\cup\Delta'}}\max_{a'_g\in\mathcal{A}_g,a'_\Delta\in\mathcal{A}_l^k}\hat{Q}_k^T(s'_g,F_{z'_\Delta},a'_g)-\mathbb{E}_{s'_g,s'_{\Delta\cup\Delta'}}\max_{a'_g\in\mathcal{A}_g,a'_{\Delta'}\in\mathcal{A}_l^{k'}}\hat{Q}_{k'}^T(s'_g,F_{z'_{\Delta'}},a'_g)\right|$$

$$=\mathbb{E}_{s'_g,s'_{\Delta\cup\Delta'}}\hat{Q}_k^T(s'_g,s'_\Delta,a^*_g,a^*_\Delta)-\mathbb{E}_{s'_g,s'_{\Delta\cup\Delta'}}\hat{Q}_{k'}^T(s'_g,s'_{\Delta'},\tilde{a}^*_g,\tilde{a}^*_{\Delta'})$$

$$\leq\mathbb{E}_{s'_g,s'_{\Delta\cup\Delta'}}\hat{Q}_k^T(s'_g,s'_\Delta,a^*_g,a^*_\Delta)-\mathbb{E}_{s'_g,s'_{\Delta\cup\Delta'}}\hat{Q}_{k'}^T(s'_g,s'_{\Delta'},a^*_g,a^*_{\Delta'})$$

$$\leq\max_{\substack{a'_g\in\mathcal{A}_g,\\a_{\Delta\cup\Delta'}\in\mathcal{A}_l^{|\Delta\cup\Delta'|}}}\left|\mathbb{E}_{s'_g,s'_{\Delta\cup\Delta'}}\hat{Q}_k^T(s'_g,s'_\Delta,a'_g,a'_\Delta)-\mathbb{E}_{s'_g,s'_{\Delta\cup\Delta'}}\hat{Q}_{k'}^T(s'_g,s'_{\Delta'},a'_g,a'_{\Delta'})\right|$$

We observe that if the opposite inequality holds (i.e., $\mathbb{E}_{s'_g,s'_{\Delta\cup\Delta'}}\max_{a'_g\in\mathcal{A}_g,a'_\Delta\in\mathcal{A}_l^k}\hat{Q}_k^T(s'_g,s'_\Delta,a'_g,a'_\Delta)-\mathbb{E}_{s'_g,s'_{\Delta\cup\Delta'}}\max_{a'_g\in\mathcal{A}_g,a'_{\Delta'}\in\mathcal{A}_l^{k'}}\hat{Q}_{k'}^T(s'_g,s'_{\Delta'},a'_g,a'_{\Delta'})<0$), an analogous argument by replacing $a^*_g$ with $\tilde{a}^*_g$ and $a^*_\Delta$ with $\tilde{a}^*_\Delta$ yields an identical bound. $\qquad\square$

**Lemma E.15.** *Suppose $z,z'\geq 1$. Consider functions $\Gamma:\Theta_1\times\Theta_2\times\cdots\times\Theta_z\times\Theta^*\to\mathbb{R}$ and $\Gamma':\Theta'_1\times\Theta'_2\times\cdots\times\Theta'_{z'}\times\Theta^*\to\mathbb{R}$, where $\Theta_1,\ldots,\Theta_z$ and $\Theta'_1,\ldots,\Theta'_{z'}$ are finite sets. Consider a probability distribution function $\mu_{\Theta_i}$ for $i\in[z]$ and $\mu'_{\Theta_i}$ for $i\in[z']$. Then:*

$$\left|\mathbb{E}_{\substack{\theta_1\sim\mu_{\Theta_1}\\\cdots\\\theta_z\sim\mu_{\Theta_z}}}\max_{\theta^*\in\Theta^*}\Gamma(\theta_1,\ldots,\theta_z,\theta^*)-\mathbb{E}_{\substack{\theta_1\sim\mu'_{\Theta_1}\\\cdots\\\theta_{z'}\sim\mu'_{\Theta_{z'}}}}\max_{\theta^*\in\Theta^*}\Gamma'(\theta_1,\ldots,\theta_{z'},\theta^*)\right|$$

$$\leq\max_{\theta^*\in\Theta^*}\left|\mathbb{E}_{\substack{\theta_1\sim\mu_{\Theta_1}\\\cdots\\\theta_z\sim\mu_{\Theta_z}}}\Gamma(\theta_1,\ldots,\theta_z,\theta^*)-\mathbb{E}_{\substack{\theta_1\sim\mu'_{\Theta_1}\\\cdots\\\theta_{z'}\sim\mu'_{\Theta_{z'}}}}\Gamma'(\theta_1,\ldots,\theta_{z'},\theta^*)\right|$$

*Proof.* Let $\hat{\theta}^*:=\arg\max_{\theta^*\in\Theta^*}\Gamma(\theta_1,\ldots,\theta_z,\theta^*)$ and $\tilde{\theta}^*:=\arg\max_{\theta^*\in\Theta^*}\Gamma'(\theta_1,\ldots,\theta_{z'},\theta^*)$.

If $\mathbb{E}_{\theta_1\sim\mu_{\Theta_1},\ldots,\theta_z\sim\mu_{\Theta_z}}\max_{\theta^*\in\Theta^*}\Gamma(\theta_1,\ldots,\theta_z,\theta^*)-\mathbb{E}_{\theta_1\sim\mu'_{\Theta'_1},\ldots,\theta_{z'}\sim\mu'_{\Theta'_{z'}}}\max_{\theta^*\in\Theta^*}\Gamma'(\theta_1,\ldots,\theta_{z'},\theta^*)>0$, then:

$$\left|\mathbb{E}_{\substack{\theta_1\sim\mu_{\Theta_1}\\\cdots\\\theta_z\sim\mu_{\Theta_z}}}\max_{\theta^*\in\Theta^*}\Gamma(\theta_1,\ldots,\theta_z,\theta^*)-\mathbb{E}_{\substack{\theta_1\sim\mu'_{\Theta'_1}\\\cdots\\\theta_{z'}\sim\mu'_{\Theta'_{z'}}}}\max_{\theta^*\in\Theta^*}\Gamma'(\theta_1,\ldots,\theta_{z'},\theta^*)\right|$$

$$=\mathbb{E}_{\substack{\theta_1\sim\mu_{\Theta_1}\\\cdots\\\theta_z\sim\mu_{\Theta_z}}}\Gamma(\theta_1,\ldots,\theta_z,\hat{\theta}^*)-\mathbb{E}_{\substack{\theta_1\sim\mu'_{\Theta'_1}\\\cdots\\\theta_{z'}\sim\mu'_{\Theta'_{z'}}}}\Gamma'(\theta_1,\ldots,\theta_{z'},\tilde{\theta}^*)$$

$$\leq\mathbb{E}_{\substack{\theta_1\sim\mu_{\Theta_1}\\\cdots\\\theta_z\sim\mu_{\Theta_z}}}\Gamma(\theta_1,\ldots,\theta_z,\hat{\theta}^*)-\mathbb{E}_{\substack{\theta_1\sim\mu'_{\Theta'_1}\\\cdots\\\theta_{z'}\sim\mu'_{\Theta'_{z'}}}}\Gamma'(\theta_1,\ldots,\theta_{z'},\hat{\theta}^*)$$

$$\leq\max_{\theta^*\in\Theta^*}\left|\mathbb{E}_{\substack{\theta_1\sim\mu_{\Theta_1}\\\cdots\\\theta_z\sim\mu_{\Theta_z}}}\Gamma(\theta_1,\ldots,\theta_z,\theta^*)-\mathbb{E}_{\substack{\theta_1\sim\mu'_{\Theta'_1}\\\cdots\\\theta_{z'}\sim\mu'_{\Theta'_{z'}}}}\Gamma'(\theta_1,\ldots,\theta_{z'},\theta^*)\right|$$

Here, we replace each $\theta^*$ with the maximizers of their corresponding terms, and upper bound them by the maximizer of the larger term. Next, we replace $\hat{\theta}^*$ in both expressions with the maximizer choice $\theta^*$ from $\Theta^*$, and further bound the expression by its absolute value.

If $\mathbb{E}_{\theta_1\sim\mu_{\Theta_1},\ldots,\theta_z\sim\mu_{\Theta_z}}\max_{\theta^*\in\Theta^*}\Gamma(\theta_1,\ldots,\theta_z,\theta^*)-\mathbb{E}_{\theta_1\sim\mu'_{\Theta'_1},\ldots,\theta_{z'}\sim\mu'_{\Theta'_{z'}}}\max_{\theta^*\in\Theta^*}\Gamma'(\theta_1,\ldots,\theta_{z'},\theta^*)$ is negative, then an analogous argument that replaces $\hat{\theta}^*$ with $\tilde{\theta}^*$ yields the same result. $\qquad\square$

## F  Bounding Total Variation Distance

As $|\Delta| \to n$, we prove that the total variation (TV) distance between the empirical distribution of $z_\Delta$ and $z_{[n]}$ goes to 0. Here, recall that $z_i \in \mathcal{Z} = \mathcal{S}_l \times \mathcal{A}_l$, and $z_\Delta = \{z_i : i \in \Delta\}$ for $\Delta \in \binom{[n]}{k}$. Before bounding the total variation distance between $F_{z_\Delta}$ and $F_{z_{\Delta'}}$, we first introduce Lemma C.5 of Anand and Qu [2024] which can be viewed as a generalization of the Dvoretzky-Kiefer-Wolfowitz concentration inequality, for sampling without replacement. We first make an important remark.

**Remark F.1.** First, observe that if $\Delta$ is an independent random variable uniformly supported on $\binom{[n]}{k}$, then $s_\Delta$ and $a_\Delta$ are also independent random variables uniformly supported on the global state $\binom{s_{[n]}}{k}$ and the global action $\binom{a_{[n]}}{k}$. To see this, let $\psi_1 : [n] \to \mathcal{S}_l$ where $\psi_1(i) = s_i$ and $\xi_1 : [n] \to \mathcal{A}_l$ where $\xi_1(i) = a_i$. This naturally extends to $\psi_k : [n]^k \to \mathcal{S}_l^k$, where $\psi_k(i_1, \ldots, i_k) = (s_{i_1}, \ldots, s_{i_k})$ and $\xi_k : [n]^k \to \mathcal{A}_l^k$, where $\xi_k(i_1, \ldots, i_k) = (a_{i_1}, \ldots, a_{i_k})$ for all $k \in [n]$. Then, the independence of $\Delta$ implies the independence of the generated $\sigma$-algebra. Further, $\psi_k$ and $\xi_k$ (which are a Lebesgue measurable function of a $\sigma$-algebra) are sub-algebras, implying that $s_\Delta$ and $a_\Delta$ must also be independent random variables.

For reference, we present the multidimensional Dvoretzky-Kiefer-Wolfowitz (DKW) inequality (Dvoretzky et al. [1956], Massart [1990], Naaman [2021]) which bounds the difference between an empirical distribution function for a set $B_\Delta$ and $B_{[n]}$ when each element of $\Delta$ for $|\Delta| = k$ is sampled uniformly at random from $[n]$ *with* replacement.

**Theorem F.2** (Multi-dimensional Dvoretzky-Kiefer-Wolfowitz (DFW) inequality [Dvoretzky et al., 1956, Naaman, 2021]). *Suppose $B \subset \mathbb{R}^d$ and $\epsilon > 0$. If $\Delta \subseteq [n]$ is sampled uniformly with replacement, then*

$$\Pr\left[\sup_{x \in B} \left| \frac{1}{|\Delta|} \sum_{i \in \Delta} \mathbb{1}\{B_i = x\} - \frac{1}{n} \sum_{i=1}^n \mathbb{1}\{B_i = x\} \right| < \epsilon \right] \geq 1 - d(n+1)e^{-2|\Delta|\epsilon^2}.$$

Lemma C.5 of Anand and Qu [2024] generalizes the DKW inequality for sampling without replacement:

**Lemma F.3** (Sampling without replacement analogue of the DKW inequality, Lemma C.5 in Anand and Qu [2024]). *Consider a finite population $\mathcal{X} = (x_1, \ldots, x_n) \in \mathcal{B}_l^n$ where $\mathcal{B}_l$ is a finite set. Let $\Delta \subseteq [n]$ be a random sample of size $k$ chosen uniformly and without replacement. Then, for all $x \in \mathcal{B}_l$:*

$$\Pr\left[\sup_{x \in \mathcal{B}_l} \left| \frac{1}{|\Delta|} \sum_{i \in \Delta} \mathbb{1}\{x_i = x\} - \frac{1}{n} \sum_{i \in [n]} \mathbb{1}\{x_i = x\} \right| < \epsilon \right] \geq 1 - 2|\mathcal{B}_l|e^{-\frac{2|\Delta|n\epsilon^2}{n - |\Delta| + 1}}$$

**Lemma F.4.** *With probability atleast $1 - 2|\mathcal{S}_l||\mathcal{A}_l|e^{-\frac{8kn\epsilon^2}{n-k+1}}$,*

$$\mathrm{TV}(F_{z_\Delta}, F_{z_{[n]}}) \leq \epsilon$$

*Proof.* Recall that $z := (s_l, a_l) \in \mathcal{S}_l \times \mathcal{A}_l$. From Lemma F.3, substituting $\mathcal{B}_l = \mathcal{Z}_l$ yields:

$$\Pr\left[\sup_{z_l \in \mathcal{Z}_l} \left| F_{z_\Delta}(z_l) - F_{z_{[n]}}(z_l) \right| \leq 2\epsilon \right] \geq 1 - 2|\mathcal{Z}_l|e^{-\frac{8kn\epsilon^2}{n-k+1}}$$

$$= 1 - 2|\mathcal{S}_l||\mathcal{A}_l|e^{-\frac{8kn\epsilon^2}{n-k+1}},$$

which yields the proof. $\qquad\square$

We now present an alternate bound for the total variation distance, where the distance *actually goes to 0* as $|\Delta| \to n$. For this, we use the fact that the total variation distance between two product distributions is subadditive.

**Lemma F.5** (Lemma B.8.1 of Ghosal and van der Vaart [2017]. Subadditivity of TV distance for Product Distributions). *Let $P$ and $Q$ be product distributions over some domain $S$. Let $\alpha_1, \ldots, \alpha_d$ be the marginal distributions of $P$ and $\beta_1, \ldots, \beta_q$ be the marginal distributions of $Q$. Then,*

$$\|P - Q\|_1 \leq \sum_{i=1}^d \|\alpha_i - \beta_i\|_1$$

**Lemma F.6** (KL-divergence decays too slowly).

$$\mathrm{TV}(F_{z_\Delta}, F_{z_{[n]}}) \le \sqrt{1 - \frac{|\Delta|}{n}}$$

*Proof.* By the symmetry of the total variation distance metric, we have that

$$\mathrm{TV}(F_{z_{[n]}}, F_{z_\Delta}) = \mathrm{TV}(F_{z_\Delta}, F_{z_{[n]}}).$$

From the Bretagnolle-Huber inequality Tsybakov [2008] we have that

$$\mathrm{TV}(f, g) = \sqrt{1 - e^{-D_{\mathrm{KL}}(f\|g)}}.$$

Here, $D_{\mathrm{KL}}(f\|g)$ is the Kullback-Leibler (KL) divergence metric between probability distributions $f$ and $g$ over the sample space, which we denote by $\mathcal{X}$ and is given by

$$D_{\mathrm{KL}}(f\|g) := \sum_{x \in \mathcal{X}} f(x) \ln \frac{f(x)}{g(x)} \tag{29}$$

Thus, from Equation (29):

$$
\begin{aligned}
D_{\mathrm{KL}}(F_{z_\Delta}\|F_{z_{[n]}}) &= \sum_{z \in \mathcal{Z}_l} \left( \frac{1}{|\Delta|} \sum_{i \in \Delta} \mathbb{1}\{z_i = z\} \right) \ln \frac{n \sum_{i \in \Delta} \mathbb{1}\{z_i = z\}}{|\Delta| \sum_{i \in [n]} \mathbb{1}\{z_i = z\}} \\
&= \frac{1}{|\Delta|} \sum_{z \in \mathcal{Z}_l} \left( \sum_{i \in \Delta} \mathbb{1}\{z_i = z\} \right) \ln \frac{n}{|\Delta|} \\
&\quad + \frac{1}{|\Delta|} \sum_{z \in \mathcal{Z}_l} \left( \sum_{i \in \Delta} \mathbb{1}\{z_i = z\} \right) \ln \frac{\sum_{i \in \Delta} \mathbb{1}\{z_i = z\}}{\sum_{i \in [n]} \mathbb{1}\{z_i = z\}} \\
&= \ln \frac{n}{|\Delta|} + \frac{1}{|\Delta|} \sum_{z \in \mathcal{Z}_l} \left( \sum_{i \in \Delta} \mathbb{1}\{z_i = z\} \right) \ln \frac{\sum_{i \in \Delta} \mathbb{1}\{z_i = z\}}{\sum_{i \in [n]} \mathbb{1}\{z_i = z\}} \\
&\le \ln \left( \frac{n}{|\Delta|} \right)
\end{aligned}
$$

In the third line, we note that $\sum_{z \in \mathcal{Z}_l} \sum_{i \in \Delta} \mathbb{1}\{z_i = z\} = |\Delta|$ since each local agent contained in $\Delta$ must have some state/action pair contained in $\mathcal{Z}_l$. In the last line, we note that $\sum_{i \in \Delta} \mathbb{1}\{z_i = z\} \le \sum_{i \in [n]} \mathbb{1}\{z_i = z\}$, For all $z \in \mathcal{Z}_l$, and thus the summation of logarithmic terms in the third line is negative.

Finally, using this bound in the Bretagnolle-Huber inequality yields the lemma. $\square$

**Corollary F.7.** From Lemma F.6, setting $\Delta = [n]$ also recovers $\mathrm{TV}(F_{z_\Delta}, F_{z_{[n]}}) = 0$.

**Theorem F.8.** *With probability atleast $1 - \delta$ for $\delta \in (0, 1)^2$:*

$$\left| \hat{Q}_k^*(s_g, F_{s_\Delta}, a_g, F_{a_\Delta}) - \hat{Q}_n^*(s_g, F_{s_{[n]}}, a_g, F_{a_{[n]}}) \right| \le \frac{\ln \frac{2|\mathcal{S}_l||\mathcal{A}_l|}{\delta}}{1 - \gamma} \sqrt{\frac{n - k + 1}{8kn}} \cdot \|r_l(\cdot, \cdot)\|_\infty$$

*Proof.* From combining the total variation distance bound in Lemma F.4 and the Lipschitz continuity bound in Theorem E.3 with $\sum_{t=0}^T \gamma^T \le \frac{1}{1-\gamma}$ for $\gamma \in (0, 1)$, we have:

$$\Pr\left[ |\hat{Q}_k^*(s_g, F_{s_\Delta}, a_g, F_{a_\Delta}) - \hat{Q}_n^*(s_g, F_{s_{[n]}}, a_g, F_{a_{[n]}})| \le \frac{2\epsilon}{1 - \gamma} \cdot \|r_l(\cdot, \cdot)\|_\infty \right] \ge 1 - 2|\mathcal{S}_l||\mathcal{A}_l| e^{-\frac{8kn\epsilon^2}{n-k+1}}$$

Then, reparameterizing $1 - 2|\mathcal{S}_l||\mathcal{A}_l| e^{-\frac{8kn\epsilon^2}{n-k+1}}$ into $1 - \delta$ to get $\epsilon = \sqrt{\frac{n-k+1}{8kn} \ln \left( \frac{2|\mathcal{S}_l||\mathcal{A}_l|}{\delta} \right)}$ gives that with probability at least $1 - \delta$,

$$\left| \hat{Q}_k^*(s_g, F_{s_\Delta}, a_g, F_{a_\Delta}) - \hat{Q}_n^*(s_g, F_{s_{[n]}}, a_g, F_{a_{[n]}}) \right| \le \frac{\ln \frac{2|\mathcal{S}_l||\mathcal{A}_l|}{\delta}}{1 - \gamma} \sqrt{\frac{n - k + 1}{8kn}} \cdot \|r_l(\cdot, \cdot)\|_\infty,$$

proving the claim. $\square$

# G    Using the Performance Difference Lemma

In general, convergence analysis requires the guarantee that a stationary optimal policy exists. Fortunately, when working with the empirical distribution function, the existence of a stationary optimal policy is guaranteed when the state/action spaces are finite or countably infinite. However, lifting the knowledge of states onto the continuous empirical distribution function space, and designing a policy on the lifted space is still analytically challenging. To circumvent this, Gu et al. [2021] creates lifted $\epsilon$-nets and does kernel regression to obtain convergence guarantees. Moreover, our result has a similar flavor to MDPs with dynamic exogenous inputs from learning theory, [Dietterich et al., 2018, Foster et al., 2022, Anand and Qu, 2024], wherein our subsampling algorithm treats each sampled state as an endogenous state.

Here, our analytic approach bears a stark difference, wherein we analyze the *sampled* structure of the mean-field empirical distribution function, rather than studying the structure of the lifted space. For this, we leverage the classical Performance Difference Lemma, which we restate below for completeness.

**Lemma G.1** (Performance Difference Lemma, Kakade and Langford [2002]). *Given policies $\pi_1, \pi_2$, with corresponding value functions $V^{\pi_1}, V^{\pi_2}$:*

$$V^{\pi_1}(s) - V^{\pi_2}(s) = \frac{1}{1-\gamma} \mathbb{E}_{\substack{s' \sim d_s^{\pi_1} \\ a' \sim \pi_1(\cdot|s')}} [A^{\pi_2}(s', a')]$$

Here, $A^{\pi_2}(s', a') := Q^{\pi_2}(s', a') - V^{\pi_2}(s')$ and $d_s^{\pi_1}(s') = (1-\gamma) \sum_{h=0}^{\infty} \gamma^h \Pr_h^{\pi_1}[s', s]$ where $\Pr_h^{\pi_1}[s', s]$ is the probability of $\pi_1$ reaching state $s'$ at time step $h$ starting from state $s$.

We denote our learned policy $\pi_{k,m}^{\text{est}}$ where:

$$\pi_{k,m}^{\text{est}}(s_g, s_{[n]}) = (\pi_{k,m}^{g,\text{est}}(s_g, s_{[n]}), \pi_{k,m}^{1,\text{est}}(s_g, s_{[n]}), \ldots, \pi_{k,m}^{n,\text{est}}(s_g, s_{[n]})) \in \mathcal{P}(\mathcal{A}_g) \times \mathcal{P}(\mathcal{A}_l)^n,$$

where $\pi_{k,m}^{g,\text{est}}(s_g, s_{[n]}) = \hat{\pi}_{k,m}^{g,\text{est}}(s_g, s_u, F_{s_{\Delta \setminus u}})$ is the global agent's action and $\pi_{k,m}^{i,\text{est}}(s_g, s_{[n]}) := \hat{\pi}_{k,m}^{\text{est}}(s_g, s_i, F_{s_{\Delta_i}})$ is the action of the $i$'th local agent. Here, $\Delta_i$ is a random variable supported on $\binom{[n] \setminus i}{k-1}$, $u$ is a random variable uniformly distributed on $[n]$, and $\Delta$ is a random variable uniformly distributed on $[n] \setminus u$. Then, denote the optimal policy $\pi^*$ given by

$$\pi^*(s) = (\pi_g^*(s_g, s_{[n]}), \pi_1^*(s_g, s_{[n]}), \ldots, \pi_n^*(s_g, s_{[n]})) \in \mathcal{P}(\mathcal{A}_g) \times \mathcal{P}(\mathcal{A}_l)^n,$$

where $\pi_g^*(s_g, s_{[n]})$ is the global agent's action, and $\pi_i^*(s_g, s_{[n]}) := \pi^*(s_g, s_i, F_{s_{\Delta_i}})$ is the action of the $i$'th local agent. Next, in order to compare the difference in the performance of $\pi^*(s)$ and $\pi_{k,m}^{\text{est}}(s_g, s_{[n]})$, we define the value function of a policy $\pi$ to be the infinite-horizon $\gamma$-discounted rewards, (denoted by $V^\pi$) as follows:

**Definition G.2.** The value function $V^\pi : \mathcal{S} \to \mathbb{R}$ of a given policy $\pi$, for $\mathcal{S} := \mathcal{S}_g \times \mathcal{S}_l^n$ is:

$$V^\pi(s) = \mathbb{E}_{a(t) \sim \pi(\cdot|s(t))} \left[ \sum_{t=0}^{\infty} \gamma^t r(s(t), a(t)) | s(0) = s \right]. \tag{30}$$

**Theorem G.3.** *For the optimal policy $\pi^*$ and the learned policy $\pi_{k,m}^{\text{est}}$, for any state $s_0 \in \mathcal{S}$, we have:*

$$V^{\pi^*}(s_0) - V^{\pi_{k,m}^{\text{est}}}(s_0) \le \frac{\tilde{r}}{(1-\gamma)^2} \sqrt{\frac{n-k+1}{2nk}} \sqrt{\ln \frac{2|\mathcal{S}_l||\mathcal{A}_l|}{\delta}} + 2\epsilon_{k,m} + \frac{\tilde{2r}}{(1-\gamma)^2} |\mathcal{A}_g| k^{|\mathcal{A}_l|} \delta$$

*Proof.* Applying the performance difference lemma to the policies gives us:

$$V^{\pi^*}(s_0) - V^{\tilde{\pi}_{k,m}^{\text{est}}}(s_0) = \frac{1}{1-\gamma} \mathbb{E}_{s \sim d_{s_0}^{\pi_{k,m}^{\text{est}}}} \mathbb{E}_{a \sim \pi_{k,m}^{\text{est}}(\cdot|s)} [V^{\pi^*}(s) - Q^{\pi^*}(s, a)]$$

$$= \frac{1}{1-\gamma} \mathbb{E}_{s \sim d_{s_0}^{\pi_{k,m}^{\text{est}}}} \left[ \mathbb{E}_{a' \sim \pi^*(\cdot|s)} Q^{\pi^*}(s, a') - \mathbb{E}_{a \sim \pi_{k,m}^{\text{est}}(\cdot|s)} Q^{\pi^*}(s, a) \right]$$

$$= \frac{1}{1-\gamma} \mathbb{E}_{s \sim d_{s_0}^{\tilde{\pi}_{k,m}^{\text{est}}}} \left[ Q^{\pi^*}(s, \pi^*(\cdot|s)) - \mathbb{E}_{a \sim \pi_{k,m}^{\text{est}}(\cdot|s)} Q^{\pi^*}(s, a) \right]$$

Next, by the law of total expectation,

$$\mathbb{E}_{a\sim\pi_{k,m}^{\text{est}}(\cdot|s)}\left[Q^*(s,a)\right]$$

$$= \sum_{\Delta\in\binom{[n]}{k}} \sum_{\substack{\Delta^1,\dots,\Delta^n:\\\Delta^i\in\binom{[n]\setminus i}{k-1}}} \frac{1}{\binom{n}{k}} \frac{1}{\binom{n-1}{k-1}^n} Q^*(s,\hat{\pi}_{k,m}^{\text{est}}(s_g,F_{s_\Delta})_g,\hat{\pi}_{k,m}^{\text{est}}(s_g,s_1,F_{s_{\Delta^1}}),\dots,\hat{\pi}_{k,m}^{\text{est}}(s_g,s_n,F_{s_{\Delta^n}}))$$

Therefore,

$$Q^{\pi^*}(s,\pi^*(\cdot|s)) - \mathbb{E}_{a\sim\pi_{k,m}^{\text{est}}(\cdot|s)}Q^*(s,a)$$

$$= \sum_{\Delta\in\binom{[n]}{k}} \sum_{\Delta^1\in\binom{[n]\setminus 1}{k-1}} \sum_{\Delta^2\in\binom{[n]\setminus 2}{k-1}} \cdots \sum_{\Delta^n\in\binom{[n]\setminus n}{k-1}} \frac{1}{\binom{n}{k}} \frac{1}{\binom{n-1}{k-1}^n} \bigg( Q^*(s,\pi^*(\cdot|s))$$

$$- Q^*(s,\hat{\pi}_{k,m}^{\text{est}}(s_g,F_{s_\Delta})_g,\hat{\pi}_{k,m}^{\text{est}}(s_g,s_1,F_{s_{\Delta^1}}),\dots,\hat{\pi}_{k,m}^{\text{est}}(s_g,s_n,F_{s_{\Delta^n}})) \bigg)$$

Therefore, by grouping the equations above, we have:

$$V^{\pi^*}(s_0) - V^{\tilde{\pi}_{k,m}^{\text{est}}}(s_0)$$

$$\leq \frac{1}{1-\gamma}\mathbb{E}_{s\sim d_{s_0}^{\tilde{\pi}_{k,m}^{\text{est}}}} \mathbb{E}_{a\sim\tilde{\pi}_{k,m}^{\text{est}}(\cdot|s)} \sum_{\Delta\in\binom{[n]}{k}} \sum_{\substack{\Delta^i\in\binom{[n]\setminus i}{k-1},\\\forall i\in[n]}} \frac{1}{n\binom{n}{k}\binom{n-1}{k-1}^n} \bigg( \sum_{i\in[n]} \bigg| Q^*(s,\pi^*(s))$$

$$- \hat{Q}_{k,m}^{\text{est}}(s_g,s_i,F_{s_{\Delta^i}},\pi^*(s)_g,\{\pi^*(s)_j\}_{j\in\{i,\Delta^i\}}) \bigg|$$

$$+ \frac{\frac{1}{n}}{\binom{n}{k}\binom{n-1}{k-1}^n} \sum_{i\in[n]} \bigg| \hat{Q}_{k,m}^{\text{est}}(s_g,s_i,F_{s_{\Delta^i}},\hat{\pi}_{k,m}^{\text{est}}(s_g,F_{s_\Delta})_g,\{\hat{\pi}_{k,m}^{\text{est}}(s_g,s_{j,\Delta^j})\}_{j\in\{i,\Delta^i\}})$$

$$- Q^*(s,\hat{\pi}_{k,m}^{\text{est}}(s_g,s_\Delta)_g,\{\hat{\pi}_{k,m}^{\text{est}}(s_g,s_j,F_{s_{\Delta^j}})\}_{j\in[n]}) \bigg| \bigg)$$

Lemma G.4 shows a uniform bound on

$$Q^*(s,\pi^*(\cdot|s)) - Q^*(s,\hat{\pi}_{k,m}^{\text{est}}(s_g,F_{s_\Delta})_g,\hat{\pi}_{k,m}^{\text{est}}(s_g,s_1,F_{s_{\Delta^1}}),\dots,\hat{\pi}_{k,m}^{\text{est}}(s_g,s_n,F_{s_{\Delta^n}}))$$

(independent of $\Delta^i, \forall i \in [n]$), allowing the sums and the counts in the denominator will cancel out. Observe that $\hat{\pi}_{k,m}^* : \mathcal{S} \to \mathcal{A}$ and $\pi^* : \mathcal{S} \to \mathcal{A}$ are deterministic functions. Therefore, denote $a = \pi^*(s)$. Then, from Lemma G.6,

$$\frac{1}{1-\gamma}\mathbb{E}_{s\sim d_{s_0}^{\pi_{k,m}^{\text{est}}}} \mathbb{E}_{a\sim\pi_{k,m}^{\text{est}}(\cdot|s)} \sum_{\Delta\in\binom{[n]}{k}} \sum_{\Delta^1\in\binom{[n]\setminus 1}{k-1}} \cdots \sum_{\Delta^n\in\binom{[n]\setminus n}{k-1}} \frac{1}{n\binom{n}{k}\binom{n-1}{k-1}^n} \sum_{i\in[n]} \bigg| Q^*(s,\pi^*(s))$$

$$- \hat{Q}_{k,m}^{\text{est}}(s_g,s_i,F_{s_{\Delta^i}},\pi^*(s)_g,\{\pi^*(s)_j\}_{j\in\{i,\Delta^i\}}) \bigg|$$

$$= \frac{1}{1-\gamma}\mathbb{E}_{s\sim d_{s_0}^{\pi_{k,m}^{\text{est}}}} \sum_{\Delta\in\binom{[n]}{k}} \sum_{\substack{\Delta^1,\dots,\Delta^n:\\\Delta^i\in\binom{[n]\setminus i}{k-1}}} \frac{\frac{1}{n\binom{n}{k}}}{\binom{n-1}{k-1}^n} \sum_{i\in[n]} \bigg| \hat{Q}_n^*(s_g,F_{s_{[n]}},\hat{\pi}_n^*(s_g,F_{s_{[n]}})_g,\hat{\pi}_n^*(s_g,F_{s_{[n]}})_{1:n})$$

$$- \hat{Q}_{k,m}^{\text{est}}(s_g,s_i,F_{s_{\Delta^i}},\pi^*(s)_g,\{\pi^*(s)_j\}_{j\in\{i,\Delta^i\}}) \bigg|$$

$$\leq \frac{\tilde{r}}{(1-\gamma)^2}\sqrt{\frac{n-k+1}{8nk}}\sqrt{\ln\frac{2|\mathcal{S}_l||\mathcal{A}_l|}{\delta}} + \epsilon_{k,m} + \frac{\tilde{r}}{(1-\gamma)^2}|\mathcal{A}_g|k^{|\mathcal{A}_l|}\delta$$

Similarly, from Corollary G.7, we have that

$$\frac{1}{1-\gamma}\mathbb{E}_{s\sim d_{s_0}^{\pi_{k,m}^{\text{est}}}} \sum_{\Delta\in\binom{[n]}{k}} \sum_{\Delta^1\in\binom{[n]\setminus 1}{k-1}} \cdots \sum_{\Delta^n\in\binom{[n]\setminus n}{k-1}} \frac{\frac{1}{n\binom{n}{k}}}{\binom{n-1}{k-1}^n}\Bigg|\hat{Q}_{k,m}^{\text{est}}(s_g,s_i,F_{s_{\Delta^i}},$$

$$\hat{\pi}_{k,m}^{\text{est}}(s_g,s_\Delta)_g,\{\hat{\pi}_{k,m}^{\text{est}}(s_g,s_j,F_{s_{\Delta^j}})\}_{j\in\{i,\Delta^i\}}) - Q^*(s,\hat{\pi}_{k,m}^{\text{est}}(s_g,s_\Delta)_g,\{\hat{\pi}_{k,m}^{\text{est}}(s_g,s_j,F_{s_{\Delta^j}})\}_{j\in[n]})\Bigg|$$

$$\leq \frac{\tilde{r}}{(1-\gamma)^2}\sqrt{\frac{n-k+1}{8nk}}\sqrt{\ln\frac{2|\mathcal{S}_l||\mathcal{A}_l|}{\delta}} + \epsilon_{k,m} + \frac{\tilde{r}}{(1-\gamma)^2}|\mathcal{A}_g|k^{|\mathcal{A}_l|}\delta$$

Hence, combining the above inequalities, we get:

$$V^{\pi^*}(s_0) - V^{\pi_{k,m}^{\text{est}}}(s_0) \leq \frac{2\tilde{r}}{(1-\gamma)^2}\sqrt{\frac{n-k+1}{8nk}}\sqrt{\ln\frac{2|\mathcal{S}_l||\mathcal{A}_l|}{\delta}} + 2\epsilon_{k,m} + \frac{2\tilde{r}}{(1-\gamma)^2}|\mathcal{A}_g|k^{|\mathcal{A}_l|}\delta$$

which yields the claim. We defer parameter optimization to Lemma G.8. $\qquad\square$

**Lemma G.4** (Uniform Bound on $Q^*$ with different actions). *For all $s\in\mathcal{S}$, $\Delta\in\binom{[n]}{k}$ and $\Delta^i\in\binom{[n]\setminus i}{k-1}$ for $i\in[n]$, we have:*

$$Q^*(s,\pi^*(\cdot|s)) - Q^*(s,\hat{\pi}_{k,m}^{\text{est}}(s_g,F_{s_\Delta})_g,\{\hat{\pi}_{k,m}^{\text{est}}(s_g,s_i,F_{s_{\Delta^i}})\}_{i\in[n]})$$

$$\leq \frac{1}{n}\sum_{i\in[n]}\Bigg|Q^*(s,\pi^*(s)) - \hat{Q}_{k,m}^{\text{est}}(s_g,s_i,F_{s_{\Delta^i}},\pi^*(s)_g,\{\pi^*(s)_j\}_{j\in\{i,\Delta^i\}})\Bigg|$$

$$+ \Bigg|\hat{Q}_{k,m}^{\text{est}}(s_g,s_i,F_{s_{\Delta^i}},\hat{\pi}_{k,m}^{\text{est}}(s_g,F_{s_\Delta})_g,\{\hat{\pi}_{k,m}^{\text{est}}(s_g,s_{j,\Delta^j})\}_{j\in\{i,\Delta^i\}})$$

$$- Q^*(s,\hat{\pi}_{k,m}^{\text{est}}(s_g,s_\Delta)_g,\{\hat{\pi}_{k,m}^{\text{est}}(s_g,s_j,F_{s_{\Delta^j}})\}_{j\in[n]})\Bigg|$$

*Proof.* Observe that

$$Q^*(s,\pi^*(\cdot|s)) - Q^*(s,\hat{\pi}_{k,m}^{\text{est}}(s_g,F_{s_\Delta})_g,\{\hat{\pi}_{k,m}^{\text{est}}(s_g,s_i,F_{s_{\Delta^i}})\}_{i\in[n]})$$

$$\leq \frac{1}{n}\sum_{i\in[n]}\hat{Q}_{k,m}^{\text{est}}(s_g,s_i,F_{s_{\Delta^i}},\hat{\pi}_{k,m}^{\text{est}}(s_g,F_{s_\Delta})_g,\hat{\pi}_{k,m}^{\text{est}}(s_g,s_i,F_{s_{\Delta^i}}),\{\hat{\pi}_{k,m}^{\text{est}}(s_g,s_j,F_{s_{\Delta^j}})\}_{j\in\Delta^i})$$

$$- \frac{1}{n}\sum_{i\in[n]}\hat{Q}_{k,m}^{\text{est}}(s_g,s_i,F_{s_{\Delta^i}},\hat{\pi}_{k,m}^{\text{est}}(s_g,F_{s_\Delta})_g,\hat{\pi}_{k,m}^{\text{est}}(s_g,s_i,F_{s_{\Delta^i}}),\{\hat{\pi}_{k,m}^{\text{est}}(s_g,s_j,F_{s_{\Delta^j}})\}_{j\in\Delta^i})$$

$$+ \frac{1}{n}\sum_{i\in[n]}\hat{Q}_{k,m}^{\text{est}}(s_g,s_i,F_{s_{\Delta^i}},\pi^*(s)_g,\pi^*(s)_i,\{\pi^*(s)_j\}_{j\in\Delta^k})$$

$$- \frac{1}{n}\sum_{i\in[n]}\hat{Q}_{k,m}^{\text{est}}(s_g,s_i,F_{s_{\Delta^i}},\pi^*(s)_g,\pi^*(s)_i,\{\pi^*(s)_j\}_{j\in\Delta^k})$$

$$\leq \Bigg|Q^*(s,\pi^*(s)) - \frac{1}{n}\sum_{i\in[n]}\hat{Q}_{k,m}^{\text{est}}(s_g,s_i,F_{s_{\Delta^i}},\pi^*(s)_g,\pi^*(s)_i,\{\pi^*(s)_j\}_{j\in\Delta^i})\Bigg|$$

$$+ \Bigg|\frac{1}{n}\sum_{i\in[n]}\hat{Q}_{k,m}^{\text{est}}(s_g,s_i,F_{s_{\Delta^i}},\hat{\pi}_{k,m}^{\text{est}}(s_g,F_{s_\Delta})_g,\{\hat{\pi}_{k,m}^{\text{est}}(s_g,s_j,F_{s_{\Delta^j}})\}_{j\in\{i,\Delta^i\}})$$

$$- Q^*(s,\hat{\pi}_{k,m}^{\text{est}}(s_g,F_{s_\Delta})_g,\{\hat{\pi}_{k,m}^{\text{est}}(s_g,s_j,F_{s_{\Delta^j}})\}_{j\in[n]})\Bigg|$$

$$\leq \frac{1}{n}\sum_{i\in[n]}\Bigg|Q^*(s,\pi^*(s)) - \hat{Q}_{k,m}^{\text{est}}(s_g,s_i,F_{s_{\Delta^i}},\pi^*(s)_g,\{\pi^*(s)_j\}_{j\in\{i,\Delta^i\}})\Bigg|$$

$$+ \frac{1}{n} \sum_{i \in [n]} \left| \hat{Q}^{\text{est}}_{k,m}(s_g, s_i, F_{s_{\Delta i}}, \hat{\pi}^{\text{est}}_{k,m}(s_g, F_{s_\Delta})_g, \{\hat{\pi}^{\text{est}}_{k,m}(s_g, s_{j,\Delta^j})\}_{j \in \{i, \Delta^i\}}) \right.$$

$$\left. - Q^*(s, \hat{\pi}^{\text{est}}_{k,m}(s_g, s_\Delta)_g, \{\hat{\pi}^{\text{est}}_{k,m}(s_g, s_j, F_{s_{\Delta^j}})\}_{j \in [n]}) \right|,$$

which proves the claim. $\qquad \square$

**Lemma G.5.** *Fix $s \in \mathcal{S} := \mathcal{S}_g \times \mathcal{S}_l^n$. For each $j \in [n]$, suppose we are given $T$-length sequences of random variables $\{\Delta^j_1, \ldots, \Delta^j_T\}_{j \in [n]}$, distributed uniformly over the support $\binom{[n] \setminus j}{k-1}$. Further, suppose we are given a fixed sequence $\delta_1, \ldots, \delta_T$, where each $\delta_t \in (0, 1]$ for $t \in [T]$. Let*

$$\Phi_{k,t} = \frac{1}{1 - \gamma} \sqrt{\frac{n - k + 1}{8kn} \ln \frac{2|\mathcal{S}_l||\mathcal{A}_l|}{\delta_t}}.$$

*Then, for each action $a = (a_g, a_{[n]}) = \pi^*(s)$, for $t \in [T]$ and $j \in [n]$, define* deviation events $B_t^{a_g, a_{j, \Delta^j_t}}$ *such that:*

$$B_t^{a_g, a_{j, \Delta^j_t}} := \left\{ \left| Q^*(s_g, s_j, F_{z_{[n] \setminus j}}, a_j, a_g) - \hat{Q}^{\text{est}}_{k,m}(s_g, s_j, F_{z_{\Delta^j_t}}, a_j, a_g) \right| > \Phi_{k,t} \cdot \|r_l(\cdot, \cdot)\|_\infty + \epsilon_{k,m} \right\}$$
(31)

*For $i \in [T]$, we define bad-events $B_t$ (which is a union over each deviation event):*

$$B_t = \bigcup_{j \in [n]} \bigcup_{a_g \in \mathcal{A}_g} \bigcup_{a_{j, \Delta^j_t} \in \mathcal{A}_l^k} B_t^{a_g, a_{j, \Delta^j_t}}$$

*Next, denote $B = \bigcup_{i=1}^T B_i$. Then, the probability that no bad event $B_t$ occurs is:*

$$\Pr[\bar{B}] := 1 - \Pr[B] \geq 1 - |\mathcal{A}_g| k^{|\mathcal{A}_l|} \sum_{i=1}^T \delta_i$$

*Proof.*

$$\left| Q^*(s_g, s_j, F_{z_{[n] \setminus j}}, a_g, a_j) - \hat{Q}^{\text{est}}_{k,m}(s_g, s_j, F_{z_{\Delta^j_t}}, a_g, a_j) \right|$$

$$= \left| Q^*(s_g, s_j, F_{z_{[n] \setminus j}}, a_g, a_j) - \hat{Q}^*_k(s_g, s_j, F_{z_{\Delta^j_t}}, a_g, a_j) \right.$$

$$\left. + \hat{Q}^*_k(s_g, s_j, F_{z_{\Delta^j_t}}, a_g, a_j) - \hat{Q}^{\text{est}}_{k,m}(s_g, s_j, F_{z_{\Delta^j_t}}, a_g, a_j) \right|$$

$$\leq \left| Q^*(s_g, s_j, F_{z_{[n] \setminus j}}, a_g, a_j) - \hat{Q}^*_k(s_g, s_j, F_{z_{\Delta^j_t}}, a_g, a_j) \right|$$

$$+ \left| \hat{Q}^*_k(s_g, s_j, F_{z_{\Delta^j_t}}, a_g, a_j) - \hat{Q}^{\text{est}}_{k,m}(s_g, s_j, F_{z_{\Delta^j_t}}, a_g, a_j) \right|$$

$$\leq \left| Q^*(s_g, s_j, F_{z_{[n] \setminus j}}, a_g, a_j) - \hat{Q}^*_k(s_g, s_j, F_{z_{\Delta^j_t}}, a_g, a_j) \right| + \epsilon_{k,m}$$

The first inequality above follows from the triangle inequality, and the second inequality uses

$$\left| \hat{Q}^*_k(s_g, s_j, F_{z_{\Delta^j_t}}, a_g, a_j) - \hat{Q}^{\text{est}}_{k,m}(s_g, s_j, F_{z_{\Delta^j_t}}, a_g, a_j) \right|$$

$$\leq \left\| \hat{Q}^*_k(s_g, s_j, F_{z_{\Delta^j_t}}, a_g, a_j) - \hat{Q}^{\text{est}}_{k,m}(s_g, s_j, F_{z_{\Delta^j_t}}, a_g, a_j) \right\|_\infty$$

$$\leq \epsilon_{k,m},$$

where the $\epsilon_{k,m}$ follows from Lemma 4.3. From Theorem F.8, we have that with probability at least $1 - \delta_t$,

$$\left| Q^*(s_g, s_j, F_{z_{[n] \setminus j}}, a_g, a_j) - \hat{Q}^*_k(s_g, s_j, F_{z_{\Delta^j_t}}, a_g, a_j) \right| \leq \Phi_{k,t} \cdot \|r_l(\cdot, \cdot)\|_\infty$$

$$= \frac{1}{1 - \gamma} \sqrt{\ln \frac{2|\mathcal{S}_l||\mathcal{A}_l|}{\delta_t}} \sqrt{\frac{n - k + 1}{8kn}} \cdot \|r_l(\cdot, \cdot)\|_\infty$$

So, event $B_t^{a_g,a_{j,\Delta_t^j}}$ occurs with probability atmost $\delta_t$.

Here, observe that if we union bound across all the events parameterized by the empirical distributions of $a_g, a_{j,\Delta_t^j} \in \mathcal{A}_g \times \mathcal{A}_l^k$ given by $F_{a_{\Delta_t^j}}$, this also forms a covering of the choice of variables $F_{s_{\Delta_t^j}}$ by agents $j \in [m]$, and therefore across all choices of $\Delta_t^1, \ldots, \Delta_t^n$ (subject to the permutation invariance of local agents) for a fixed $t$.

Thus, from the union bound, we get:

$$\Pr[B_t] \leq \sum_{a_g \in \mathcal{A}_g} \sum_{F_{a_{\Delta_t}} \in \mu_k(\mathcal{A}_l)} \Pr[B_t^{a_g,a_{1,\Delta_t^1}}] \leq |\mathcal{A}_g| k^{|\mathcal{A}_l|} \Pr[B_t^{a_g,a_{1,\Delta_t^1}}]$$

Applying the union bound again proves the lemma:

$$\Pr[\bar{B}] \geq 1 - \sum_{t=1}^{T} \Pr[B_t] \geq 1 - |\mathcal{A}_g| k^{|\mathcal{A}_l|} \sum_{t=1}^{T} \delta_t,$$

which proves the claim. $\qquad\square$

**Lemma G.6.** *For any arbitrary distribution $\mathcal{D}$ of states $\mathcal{S} := \mathcal{S}_g \times \mathcal{S}_l^n$, for any $\Delta^i \in \binom{[n]\backslash i}{k-1}$ for $i \in [n]$ and for $\delta \in (0,1]$ we have:*

$$\mathbb{E}_{s \sim \mathcal{D}} \Bigg[ \sum_{\Delta \in \binom{[n]}{k}} \sum_{\Delta^1 \in \binom{[n]\backslash 1}{k-1}} \cdots \sum_{\Delta^n \in \binom{[n]\backslash n}{k-1}} \frac{1}{\binom{n}{k}} \frac{1}{\binom{n-1}{k-1}^n} \sum_{i \in [n]} \frac{1}{n} \Bigg| Q^*(s, \pi^*(s))$$
$$- \hat{Q}_{k,m}^{\text{est}}(s_g, s_i, F_{s_{\Delta^i}}, \pi^*(s)_g, \{\pi^*(s)_j\}_{j \in \{i, \Delta^i\}}) \Bigg| \Bigg]$$

$$\leq \frac{\tilde{r}}{1-\gamma} \sqrt{\frac{n-k+1}{8nk}} \sqrt{\ln \frac{2|\mathcal{S}_l||\mathcal{A}_l|}{\delta}} + \epsilon_{k,m} + \frac{\tilde{r}}{1-\gamma} |\mathcal{A}_g| k^{|\mathcal{A}_l|} \delta$$

*Proof.* By the linearity of expectations, observe that:

$$\mathbb{E}_{s \sim \mathcal{D}} \Bigg[ \sum_{\Delta \in \binom{[n]}{k}} \sum_{\Delta^1 \in \binom{[n]\backslash 1}{k-1}} \cdots \sum_{\Delta^n \in \binom{[n]\backslash n}{k-1}} \frac{1}{\binom{n}{k}} \frac{1}{\binom{n-1}{k-1}^n} \sum_{i \in [n]} \frac{1}{n} \Bigg| Q^*(s, \pi^*(s))$$
$$- \hat{Q}_{k,m}^{\text{est}}(s_g, s_i, F_{s_{\Delta^i}}, \pi^*(s)_g, \{\pi^*(s)_j\}_{j \in \{i, \Delta^i\}}) \Bigg| \Bigg]$$

$$= \sum_{\Delta \in \binom{[n]}{k}} \sum_{\Delta^1 \in \binom{[n]\backslash 1}{k-1}} \cdots \sum_{\Delta^n \in \binom{[n]\backslash n}{k-1}} \frac{1}{\binom{n}{k}} \frac{1}{\binom{n-1}{k-1}^n} \sum_{i \in [n]} \frac{1}{n} \mathbb{E}_{s \sim \mathcal{D}} \Bigg| Q^*(s, \pi^*(s))$$
$$- \hat{Q}_{k,m}^{\text{est}}(s_g, s_i, F_{s_{\Delta^i}}, \pi^*(s)_g, \{\pi^*(s)_j\}_{j \in \{i, \Delta^i\}}) \Bigg|$$

Let

$$\Phi_{k,\delta} = \sqrt{\frac{n-k+1}{8nk}} \sqrt{\ln \frac{2|\mathcal{S}_l||\mathcal{A}_l|}{\delta}}.$$

Then, define the indicator function $\mathcal{I} : [n] \times \mathcal{S} \times \mathbb{N} \times (0,1] \to \{0,1\}$ by:

$\mathcal{I}(i, s, k, \delta) :=$
$$\mathbb{1} \left\{ \left| Q^*(s, \pi^*(s)) - \hat{Q}_{k,m}^{\text{est}}(s_g, s_i, F_{s_{\Delta^i}}, \pi^*(s)_g, \{\pi^*(s)_j\}_{j \in \{i, \Delta^i\}}) \right| \leq \frac{\|r_l(\cdot, \cdot)\|_\infty}{1-\gamma} \Phi_{k,\delta} + \epsilon_{k,m} \right\}$$

The expected difference between $Q^*(s', \pi^*(s'))$ and $\hat{Q}_{k,m}^{\text{est}}(s_g, s_i, F_{s_{\Delta^i}}, \pi^*(s)_g, \{\pi^*(s)_j\}_{j \in \{i, \Delta^i\}})$ is bounded as follows:

$$\mathbb{E}_{s \sim \mathcal{D}} \left| Q^*(s', \pi^*(s')) - \hat{Q}_{k,m}^{\text{est}}(s_g, s_i, F_{s_{\Delta^i}}, \pi^*(s)_g, \{\pi^*(s)_j\}_{j \in \{i, \Delta^i\}}) \right|$$

$$= \mathbb{E}_{s \sim \mathcal{D}} \left[ \mathcal{I}(i, s, k, \delta) \left| Q^*(s', \pi^*(s')) - \hat{Q}_{k,m}^{\text{est}}(s_g, s_i, F_{s_{\Delta^i}}, \pi^*(s)_g, \{\pi^*(s)_j\}_{j \in \{i, \Delta^i\}}) \right| \right]$$

$$+ \mathbb{E}_{s \sim \mathcal{D}} \left[ (1 - \mathcal{I}(i, s, k, \delta)) \left| Q^*(s', \pi^*(s')) - \hat{Q}_{k,m}^{\text{est}}(s_g, s_i, F_{s_{\Delta^i}}, \pi^*(s)_g, \{\pi^*(s)_j\}_{j \in \{i, \Delta^i\}}) \right| \right]$$

Here, we have used the general property for a random variable $X$ and constant $c$ that $\mathbb{E}[X] = \mathbb{E}[X \mathbb{1}\{X \leq c\}] + \mathbb{E}[(1 - \mathbb{1}\{X \leq c\})X]$. Then,

$$\mathbb{E}_{s \sim \mathcal{D}} \left| Q^*(s', \pi^*(s')) - \hat{Q}_{k,m}^{\text{est}}(s_g, s_{i,\Delta^i}, \pi^*(s)_g, \pi^*(s)_{i,\Delta^i}) \right|$$

$$\leq \frac{\tilde{r}}{1-\gamma} \sqrt{\frac{n-k+1}{8nk}} \sqrt{\ln \frac{2|\mathcal{S}_l||\mathcal{A}_l|}{\delta}} + \epsilon_{k,m} + \frac{\tilde{r}}{1-\gamma} (1 - \mathbb{E}_{s' \sim \mathcal{D}} \mathcal{I}(s', k, \delta)))$$

$$\leq \frac{\tilde{r}}{1-\gamma} \sqrt{\frac{n-k+1}{8nk}} \sqrt{\ln \frac{2|\mathcal{S}_l||\mathcal{A}_l|}{\delta}} + \epsilon_{k,m} + \frac{\tilde{r}}{1-\gamma} |\mathcal{A}_g| k^{|\mathcal{A}_l|} \delta$$

$$= \frac{\tilde{r}}{1-\gamma} \sqrt{\frac{n-k+1}{8nk}} \sqrt{\ln \frac{2|\mathcal{S}_l||\mathcal{A}_l|}{\delta}} + \epsilon_{k,m} + \frac{\tilde{r}}{1-\gamma} |\mathcal{A}_g| k^{|\mathcal{A}_l|} \delta$$

For the first term in the first inequality, we use $\mathbb{E}[X \mathbb{1}\{X \leq c\}] \leq c$. For the second term, we trivially bound $Q^*(s', \pi^*(s')) - \hat{Q}_k^*(s_g, s_{i,\Delta^i}, \pi^*(s)_g, \pi^*(s)_{i,\Delta^i})$ by the maximum value $Q^*$ can take, which is $\frac{\tilde{r}}{1-\gamma}$ by Lemma C.4. In the second inequality, we use the fact that the expectation of an indicator function is the conditional probability of the underlying event. The second inequality follows from Lemma G.5.

Since this is a uniform bound that is independent of $\Delta^j$ for $j \in [n]$ and $i \in [n]$, we have:

$$\sum_{\Delta \in \binom{[n]}{k}} \sum_{\substack{\Delta^1, \ldots, \Delta^n: \\ \Delta^i \in \binom{[n]\backslash i}{k-1}}} \frac{1/n}{\binom{n}{k}\binom{n-1}{k-1}^n} \sum_{i \in [n]} \mathbb{E}_{s \sim \mathcal{D}} \left| Q^*(s, \pi^*(s)) - \hat{Q}_{k,m}^{\text{est}}(s_g, s_i, F_{s_{\Delta^i}}, \pi^*(s)_g, \{\pi^*(s)_j\}_{j \in \{i,\Delta^i\}}) \right|$$

$$\leq \frac{\tilde{r}}{1-\gamma} \sqrt{\frac{n-k+1}{8nk}} \sqrt{\ln \frac{2|\mathcal{S}_l||\mathcal{A}_l|}{\delta}} + \epsilon_{k,m} + \frac{\tilde{r}}{1-\gamma} |\mathcal{A}_g| k^{|\mathcal{A}_l|} \delta$$

This proves the claim. □

**Corollary G.7.** *Changing the notation of the policy to $\pi_{k,m}^{\text{est}}$ in the proofs of Lemmas G.5 and G.6, such that $a \sim \tilde{\pi}_{k,m}^{\text{est}}$ then yields that:*

$$\mathbb{E}_{s \sim \mathcal{D}} \left[ \sum_{\Delta \in \binom{[n]}{k}} \sum_{\Delta^1, \ldots, \Delta^n: \Delta^i \in \binom{[n]\backslash i}{k-1}} \frac{1}{\binom{n}{k}} \frac{1}{\binom{n-1}{k-1}^n} \sum_{i \in [n]} \frac{1}{n} \left| Q^*(s, a) - \hat{Q}_{k,m}^*(s_g, s_i, F_{s_{\Delta^i}}, a_g, a_{i,\Delta^i}\}) \right| \right]$$

$$\leq \frac{\tilde{r}}{1-\gamma} \sqrt{\frac{n-k+1}{8nk}} \sqrt{\ln \frac{2|\mathcal{S}_l||\mathcal{A}_l|}{\delta}} + \epsilon_{k,m} + \frac{\tilde{r}}{1-\gamma} |\mathcal{A}_g| k^{|\mathcal{A}_l|} \delta$$

**Lemma G.8** (Optimizing Parameters). *With probability at least $1 - \frac{1}{100e^k}$,*

$$V^{\pi^*}(s_0) - V^{\pi_{k,m}}(s_0) \leq \tilde{O}\left(\frac{1}{\sqrt{k}}\right)$$

*Proof.* Setting $\delta = \frac{(1-\gamma)^2}{2\tilde{r}\varepsilon|\mathcal{A}_g|k^{|\mathcal{A}_l|}}$ in the bound for the optimality gap in Theorem G.3 gives:

$$V^{\pi^*}(s_0) - V^{\tilde{\pi}_{k,m}^{\text{est}}}(s_0) \leq \frac{\tilde{r}}{(1-\gamma)^2} \sqrt{\frac{n-k+1}{2nk}} \sqrt{\ln \frac{2|\mathcal{S}_l||\mathcal{A}_l|}{\delta}} + 2\epsilon_{k,m} + \frac{2\tilde{r}}{(1-\gamma)^2} |\mathcal{A}_g| k^{|\mathcal{A}_l|} \delta$$

$$= \frac{\tilde{r}}{(1-\gamma)^2} \sqrt{\frac{n-k+1}{2nk}} \sqrt{\ln \frac{4\tilde{r}\varepsilon|\mathcal{S}_l||\mathcal{A}_l||\mathcal{A}_g|k^{|\mathcal{A}_l|}}{(1-\gamma)^2}} + \frac{1}{\varepsilon} + 2\epsilon_{k,m}$$

Setting $\varepsilon = 10\sqrt{k}$ for any $c > 0$ recovers a decaying optimality gap of the order

$$V^{\pi^*}(s_0) - V^{\tilde{\pi}_{k,m}^{\text{est}}}(s_0) \leq \frac{\tilde{r}}{(1-\gamma)^2} \sqrt{\frac{n-k+1}{2nk}} \sqrt{\ln \frac{40\tilde{r}|\mathcal{S}_l||\mathcal{A}_l||\mathcal{A}_g|k^{|\mathcal{A}_l|+\frac{1}{2}}}{(1-\gamma)^2}} + \frac{1}{10\sqrt{k}} + 2\epsilon_{k,m}$$

Finally, using the probabilistic bound of $\epsilon_{k,m} \leq \tilde{O}(\frac{1}{\sqrt{k}})$ from Lemma G.10 yields that with probability at least $1 - \frac{1}{100e^k}$,

$$V^{\pi^*}(s_0) - V^{\tilde{\pi}_{k,m}}(s_0) \leq \tilde{O}\left(\frac{1}{\sqrt{k}}\right),$$

which proves the claim. $\qquad \square$

### G.1   Bounding the Bellman Error

This section is devoted to the proof of Lemma 4.3.

**Theorem G.9** (Theorem 2 of Li et al. [2022]). *If $m \in \mathbb{N}$ is the number of samples in the Bellman update, there exists a universal constant $0 < c_0 < 2$ and a Bellman noise $0 < \epsilon_{k,m} \leq \frac{1}{1-\gamma}$ such that*

$$\|\hat{\mathcal{T}}_{k,m}\hat{Q}_{k,m}^{\text{est}} - \hat{\mathcal{T}}_k\hat{Q}_k^*\|_\infty = \|\hat{Q}_{k,m}^{\text{est}} - \hat{Q}_k^*\|_\infty \leq \epsilon_{k,m},$$

*where $\epsilon_{k,m}$ satisfies*

$$m = \frac{c_0 \cdot \tilde{r} \cdot t_{cover}}{(1-\gamma)^5 \epsilon_{k,m}^2} \log^2\left(\frac{|\mathcal{S}||\mathcal{A}|}{\rho}\right) \log\left(\frac{1}{(1-\gamma)^2}\right), \tag{32}$$

*with probability at least $1 - \rho$, for any $\rho \in (0,1)$. Here, $t_{cover}$ stands for the cover time, which is the time taken for the trajectory to visit all state-action pairs at least once. Formally,*

$$t_{cover} := \min\left\{ t \,\middle|\, \min_{(s_0,a_0)\in\mathcal{S}\times\mathcal{A}} \mathbb{P}(\mathcal{B}_t|s_0,a_0) \geq \frac{1}{2} \right\}, \tag{33}$$

*where $\mathcal{B}_t$ denotes the event that all $(s,a) \in \mathcal{S} \times \mathcal{A}$ have been visited at least once between time 0 and time $t$, and $\mathbb{P}(\mathcal{B}_t|s_0,a_0)$ denotes the probability of $\mathcal{B}_t$ conditioned on the initial state $(s_0, a_0)$.*

**Lemma G.10.** *If $T = \frac{2}{1-\gamma} \log \frac{\tilde{r}\sqrt{k}}{1-\gamma}$, SUB-SAMPLE-MFQ: Learning runs in time $\tilde{O}(T|\mathcal{S}_g|^2|\mathcal{A}_g|^2|\mathcal{A}_l|^2|\mathcal{S}_l|^2 k^{3.5+2|\mathcal{S}_l||\mathcal{A}_l|}\tilde{r})$, while accruing a Bellman noise $\epsilon_{k,m} \leq \tilde{O}(1/\sqrt{k})$ with probability at least $1 - \frac{1}{100e^k}$, .*

*Proof.* We first prove that $\|\hat{Q}_k^T - \hat{Q}_k^*\|_\infty \leq \frac{1}{\sqrt{k}}$.

For this, it suffices to show $\gamma^T \frac{\tilde{r}}{1-\gamma} \leq \frac{1}{\sqrt{k}} \implies \gamma^T \leq \frac{1-\gamma}{\tilde{r}\sqrt{k}}$. Then, using $\gamma = 1 - (1-\gamma) \leq e^{-(1-\gamma)}$, it again suffices to show $e^{-(1-\gamma)T} \leq \frac{1-\gamma}{\tilde{r}\sqrt{k}}$. Taking logarithms, we have

$$\exp(-T(1-\gamma)) \leq \frac{1-\gamma}{\tilde{r}\sqrt{k}}$$

$$-T(1-\gamma) \leq \log\frac{1-\gamma}{\tilde{r}\sqrt{k}}$$

$$T \geq \frac{1}{1-\gamma} \log\frac{\tilde{r}\sqrt{k}}{1-\gamma}$$

Since $T = \frac{2}{1-\gamma} \log \frac{\tilde{r}\sqrt{k}}{1-\gamma} > \frac{1}{1-\gamma} \log \frac{\tilde{r}\sqrt{k}}{1-\gamma}$, the condition holds and $\|\hat{Q}_k^T - \hat{Q}_k^*\|_\infty \leq \frac{1}{\sqrt{k}}$. Then, rearranging Equation (32) and incorporating the convergence error of the $\hat{Q}_k$-function, one has that with probability at least $1 - \rho$,

$$\epsilon_{k,m} \leq \frac{1}{\sqrt{k}} + \frac{k\sqrt{2\tilde{r}t_{\text{cover}}}}{(1-\gamma)^{2.5}\sqrt{m}} \log\left(\frac{|\mathcal{S}_g||\mathcal{A}_g||\mathcal{A}_l||\mathcal{S}_l|}{\rho}\right) \log\left(\frac{1}{(1-\gamma)^2}\right) \tag{34}$$

Then, using the naïve bound $t_{\text{cover}}$ (since we are doing offline learning), we have

$$t_{\text{cover}} \leq |\mathcal{S}_g||\mathcal{A}_g||\mathcal{S}_l||\mathcal{A}_l|k^{|\mathcal{S}_l||\mathcal{A}_l|}.$$

Substituting this in Equation (34) yields that with probability $1 - \rho$,

$$\epsilon_{k,m} \leq \frac{1}{\sqrt{k}} + \frac{k\sqrt{2\tilde{r}|\mathcal{S}_g||\mathcal{A}_g||\mathcal{S}_l||\mathcal{A}_l|k^{|\mathcal{S}_l||\mathcal{A}_l|}}}{(1-\gamma)^{2.5}\sqrt{m}} \log\left(\frac{|\mathcal{S}_g||\mathcal{A}_g||\mathcal{A}_l||\mathcal{S}_l|}{\rho}\right) \log\left(\frac{1}{(1-\gamma)^2}\right) \tag{35}$$

Therefore, letting $\rho = \frac{1}{100e^k}$, setting

$$m = \frac{2\tilde{r}|\mathcal{S}_g||\mathcal{A}_g||\mathcal{S}_l||\mathcal{A}_l|k^{3.5+|\mathcal{S}_l||\mathcal{A}_l|}}{(1-\gamma)^5} \log\left(100|\mathcal{S}_g||\mathcal{A}_g||\mathcal{A}_l||\mathcal{S}_l|\right) \log\left(\frac{1}{(1-\gamma)^2}\right) \quad (36)$$

attains a Bellman error of $\epsilon_{k,m} \leq \tilde{O}(1/\sqrt{k})$ with probability at least $1 - \frac{1}{100e^k}$. Finally, the runtime of our learning algorithm is

$$O(mT|\mathcal{S}_g||\mathcal{S}_l||\mathcal{A}_g||\mathcal{A}_l|k^{|\mathcal{S}_l||\mathcal{A}_l|}) = \tilde{O}(|\mathcal{S}_g|^2|\mathcal{A}_g|^2|\mathcal{A}_l|^2|\mathcal{S}_l|^2 k^{3.5+2|\mathcal{S}_l||\mathcal{A}_l|}\tilde{r}),$$

which is still polynomial in $k$, proving the claim. $\qquad\square$

# H   Generalization to Stochastic Rewards

Suppose we are given two families of distributions, $\{\mathcal{G}_{s_g,a_g}\}_{s_g,a_g \in \mathcal{S}_g \times \mathcal{A}_g}$ and $\{\mathcal{L}_{s_i,s_g,a_i}\}_{s_i,s_g,a_i \in \mathcal{S}_l \times \mathcal{S}_g \times \mathcal{A}_l}$. Let $R(s,a)$ denote a stochastic reward of the form

$$R(s,a) = r_g(s_g, a_g) + \sum_{i \in [n]} r_l(s_i, s_g, a_i), \quad (37)$$

where the rewards of the global agent $r_g$ emerge from a distribution $r_g(s_g, a_g) \sim \mathcal{G}_{s_g,a_g}$, and the rewards of the local agents $r_l$ emerge from a distribution $r_l(s_i, s_g, a_i) \sim \mathcal{L}_{s_i,s_g,a_i}$. For $\Delta \subseteq [n]$, let $R_\Delta(s,a)$ be defined as:

$$R_\Delta(s,a) = r_g(s_g, a_g) + \sum_{i \in \Delta} r_l(s_i, s_g, a_i) \quad (38)$$

We make some standard assumptions of boundedness on $\mathcal{G}_{s_g,a_g}$ and $\mathcal{L}_{s_i,s_g,a_i}$.

**Assumption H.1.** Define

$$\bar{\mathcal{G}} = \bigcup_{(s_g,a_g) \in \mathcal{S}_g \times \mathcal{A}_g} \mathrm{supp}\left(\mathcal{G}_{s_g,a_g}\right)$$

$$\bar{\mathcal{L}} = \bigcup_{(s_i,s_g,a_i) \in \mathcal{S}_l \times \mathcal{S}_g \times \mathcal{A}_l} \mathrm{supp}\left(\mathcal{L}_{s_i,s_g,a_i}\right)$$

where for any distribution $\mathcal{D}$, $\mathrm{supp}(\mathcal{D})$ is the support (set of all random variables $\mathcal{D}$ that can be sampled with probability strictly larger than 0) of $\mathcal{D}$. Then, let $\hat{\mathcal{G}} = \sup\left(\bar{\mathcal{G}}\right), \hat{\mathcal{L}} = \sup\left(\bar{\mathcal{L}}\right), \check{\mathcal{G}} = \inf\left(\bar{\mathcal{G}}\right)$, and $\check{\mathcal{L}} = \inf\left(\bar{\mathcal{L}}\right)$. We assume that $\hat{\mathcal{G}} < \infty, \hat{\mathcal{L}} < \infty, \check{\mathcal{G}} > -\infty, \check{\mathcal{L}} > -\infty$, and that $\hat{\mathcal{G}}, \hat{\mathcal{L}}, \check{\mathcal{G}}, \check{\mathcal{L}}$ are all known in advance.

**Definition H.2.** Let the randomized empirical adapted Bellman operator be $\hat{\mathcal{T}}_{k,m}^{\text{random}}$ such that:

$$\hat{\mathcal{T}}_{k,m}^{\text{random}} \hat{Q}_{k,m}^t(s_g, s_1, F_{z_\Delta}, a_g, a_1) = R_\Delta(s,a) + \frac{\gamma}{m} \sum_{\ell \in [m]} \max_{a' \in \mathcal{A}} \hat{Q}_{k,m}^t(s_g^\ell, s_j^\ell, F_{z_{\Delta \setminus j}^\ell}, a_j^\ell, a_g^\ell), \quad (39)$$

**SUBSAMPLE-MFQ: Learning with Stochastic Rewards.** The proposed algorithm averages $\Xi$ samples of the adapted randomized empirical adapted Bellman operator, $\mathcal{T}_{k,m}^{\text{random}}$ and updates the $\hat{Q}_{k,m}$ function using with the average. One can show that $\mathcal{T}_{k,m}^{\text{random}}$ is a contraction operator with module $\gamma$. By Banach's fixed point theorem, $\mathcal{T}_{k,m}^{\text{random}}$ admits a unique fixed point $\hat{Q}_{k,m}^{\text{random}}$.

**Algorithm 4** SUBSAMPLE-MFQ: Learning with Stochastic Rewards

---

**Require:** A multi-agent system as described in Section 2. Parameter $T$ for the number of iterations in the initial value iteration step. Sampling parameters $k \in [n]$ and $m \in \mathbb{N}$. Discount parameter $\gamma \in (0,1)$. Oracle $\mathcal{O}$ to sample $s'_g \sim P_g(\cdot|s_g, a_g)$ and $s_i \sim P_l(\cdot|s_i, s_g, a_i)$ for all $i \in [n]$.

1: Set $\tilde{\Delta} = \{2, \ldots, k\}$.
2: Set $\mu_{k-1}(Z_l) = \{0, \frac{1}{k-1}, \frac{2}{k-1}, \ldots, 1\}^{|\mathcal{S}_l| \times |\mathcal{A}_l|}$.
3: Set $\hat{Q}^0_{k,m}(s_g, s_1, F_{z_{\tilde{\Delta}}}, a_g, a_1) = 0$, for $(s_g, s_1, F_{z_{\tilde{\Delta}}}, a_1, a_g) \in \mathcal{S}_g \times \mathcal{S}_l \times \mu_{k-1}(\mathcal{Z}_l) \times \mathcal{A}_g \times \mathcal{A}_l$.
4: **for** $t = 1$ to $T$ **do**
5:     **for** $(s_g, s_1, F_{z_{\tilde{\Delta}}}, a_g, a_1) \in \mathcal{S}_g \times \mathcal{S}_l \times \mu_{k-1}(\mathcal{Z}_l) \times \mathcal{A}_g \times \mathcal{A}_l$ **do**
6:         $\rho = 0$
7:         **for** $\xi \in \{1, \ldots, \Xi\}$ **do**
8:             $\rho = \rho + \hat{\mathcal{T}}^{\text{random}}_{k,m} \hat{Q}^t_{k,m}(s_g, s_1, F_{z_{\tilde{\Delta}}}, a_g, a_1)$
9:         $\hat{Q}^{t+1}_{k,m}(s_g, s_1, F_{z_{\tilde{\Delta}}}, a_g, a_1) = \rho/\Xi$
10: For all $(s_g, s_i, F_{s_{\Delta \setminus i}}) \in \mathcal{S}_g \times \mathcal{S}_l \times \mu_{k-1}(\mathcal{S}_l)$, let

$$\hat{\pi}^{\text{est}}_{k,m}(s_g, s_i, F_{s_{\Delta \setminus i}}) = \underset{(a_g, a_i, F_{a_{\Delta \setminus i}}) \in \mathcal{A}_g \times \mathcal{A}_l \times \mu_{k-1}(\mathcal{Z}_l)}{\arg\max} \hat{Q}^T_{k,m}(s_g, s_i, F_{z_{\Delta \setminus i}}, a_g, a_i).$$

---

**Theorem H.3** (Hoeffding's Theorem [Tsybakov, 2008]). *Let $X_1, \ldots, X_n$ be independent random variables such that $a_i \leq X_i \leq b_i$ almost surely. Then, let $S_n = X_1 + \cdots + X_n$. Then, for all $\epsilon > 0$,*

$$\mathbb{P}[|S_n - \mathbb{E}[S_n]| \geq \epsilon] \leq 2\exp\left(-\frac{2\epsilon^2}{\sum_{i=1}^n (b_i - a_i)^2}\right) \tag{40}$$

**Lemma H.4.** *When $\pi^{\text{est}}_{k,m}$ is derived from the randomized empirical value iteration operation and applying our online subsampling execution in Algorithm 2, we get*

$$\Pr\left[V^{\pi^*}(s_0) - V^{\tilde{\pi}_{k,m}}(s_0) \leq \tilde{O}\left(\frac{1}{\sqrt{k}}\right)\right] \geq 1 - \frac{1}{100\sqrt{k}}. \tag{41}$$

*Proof.*

$$\Pr\left[\left|\frac{\rho}{\Xi} - \mathbb{E}[R_\Delta(s,a)]\right| \geq \frac{\epsilon}{\Xi}\right] \leq 2\exp\left(-\frac{2\epsilon^2}{\sum_{i=1}^\Xi |\hat{\mathcal{G}} + \hat{\mathcal{L}} - \check{\mathcal{G}} - \check{\mathcal{L}}|^2}\right)$$

$$= 2\exp\left(-\frac{2\epsilon^2}{\Xi|\hat{\mathcal{G}} + \hat{\mathcal{L}} - \check{\mathcal{G}} - \check{\mathcal{L}}|^2}\right)$$

*Rearranging this, we get:*

$$\Pr\left[\left|\frac{\rho}{\Xi} - \mathbb{E}[R_\Delta(s,a)]\right| \leq \sqrt{\ln\left(\frac{2}{\delta}\right)\frac{|\hat{\mathcal{G}} + \hat{\mathcal{L}} - \check{\mathcal{G}} - \check{\mathcal{L}}|^2}{2\Xi^2}}\right] \geq 1 - \delta \tag{42}$$

Then, setting $\delta = \frac{1}{100\sqrt{k}}$, and setting $\Xi = 10|\hat{\mathcal{G}} + \hat{\mathcal{L}} - \check{\mathcal{G}} - \check{\mathcal{L}}|k^{1/4}\sqrt{\ln(200\sqrt{k})}$ gives:

$$\Pr\left[\left|\frac{\rho}{\Xi} - \mathbb{E}[R_\Delta(s,a)]\right| \leq \frac{1}{\sqrt{200k}}\right] \geq 1 - \frac{1}{100\sqrt{k}}$$

Then, applying the triangle inequality to $\epsilon_{k,m}$ allows us to derive a probabilistic bound on the optimality gap between $\hat{Q}^{\text{est}}_{k,m}$ and $Q^*$, where the gap is increased by $\frac{1}{\sqrt{200k}}$, and where the randomness is over the stochasticity of the rewards. Then, the optimality gap between $V^{\pi^*}$ and $V^{\pi^{\text{est}}_{k,m}}$, for when the policy $\hat{\pi}^{\text{est}}_{k,m}$ is learned using Algorithm 4 in the presence of stochastic rewards obeying Assumption H.1 follows

$$\Pr\left[V^{\pi^*}(s_0) - V^{\tilde{\pi}_{k,m}}(s_0) \leq \tilde{O}\left(\frac{1}{\sqrt{k}}\right)\right] \geq 1 - \frac{1}{100\sqrt{k}}, \tag{43}$$

which proves the lemma. $\qquad\square$

**Remark H.5.** First, note that that through the naïve averaging argument, the optimality gap above still decays to $0$ with probability decaying to $1$, as $k$ increases.

**Remark H.6.** This method of analysis could be strengthened by obtaining estimates of $\hat{\mathcal{G}}, \hat{\mathcal{L}}, \check{\mathcal{G}}, \check{\mathcal{L}}$ and using methods from order statistics to bound the errors of the estimates and use the deviations between estimates to determine an optimal online stopping time [Kleinberg, 2005] as is done in the online secretary problem. This, more sophisticated, argument would become an essential step in converting this to a truly online learning, with a stochastic approximation scheme. Furthermore, one could incorporate variance-based analysis in the algorithm, where we use information derived from higher-order moments to do a weighted update of the Bellman update [Jin et al., 2024], thereby taking advantage of the skewedness of the distribution, and allowing us the freedom to assign an optimism/pessimism score to our estimate of the reward.

# I   Partially Relaxing the Offline Learning assumption: Off-Policy Learning

A limitation of the planning Algorithm 1 is that it learns $\hat{Q}_k^*$ in an *offline* manner by assuming a generative oracle access to the transition functions $\mathbb{P}_g, \mathbb{P}_l$, and reward function $r(\cdot, \cdot)$. In certain realistic RL applications, such a generative oracle might not exist, and it is more desirable to perform off-policy learning where the agent continues to learn in an offline manner but from *historical data* Fujimoto et al. [2019]. In this setting, the agents learn the target policy $\hat{\pi}_k^*$ using data generated by a different behavior policy (the strategy it uses to explore the environment). There is a significant body of work on the theoretical guarantees in off-policy learning Chen et al. [2021b], Chen and Maguluri [2022b], Chen et al. [2021a, 2025].

In fact, these previous results are amenable to transforming guarantees about offline $Q$-learning to off-policy $Q$-learning, at the cost of $\log |\mathcal{S}_g||\mathcal{S}_l|^k|\mathcal{A}_g||\mathcal{A}_l|^k$ factors in the runtime. Therefore, this section is devoted to showing that our previous result satisfy the assumptions of transforming offline $Q$-learning to off-policy $Q$-learning for the subsampled $\hat{Q}_k$-function, and we further show that, in expectation, can maintain the decaying optimality gap of $\tilde{O}(1/\sqrt{k})$ of the learned policy $\pi_k$, where the randomness is over the heuristic exploration policy $\pi_b$.

The off-policy $\hat{Q}_k$-learning algorithm is an iterative algorithm to estimate the optimal $\hat{Q}_k$-function as follows: first, a sample trajectory $\{(s_g, s_\Delta, a_g, a_\Delta)\}$ is collected using a suitable behavior policy $\pi_{k,b}$. For simplicity of notation, let $S_\Delta = (s_g, s_\Delta)$ and $A_\Delta = (a_g, a_\Delta)$. Then, initialize $\hat{Q}_k^0 : |\mathcal{S}_g||\mathcal{S}_l|^k|\mathcal{A}_g||\mathcal{A}_l|^k \to \mathbb{R}$ and let $\alpha > 0$ be determined later. For each $t \geq 0$ and state-action pair $(S_\Delta, A_\Delta)$ that is updated to $S'_\Delta$, the iterate $\hat{Q}_k^t(S_\Delta, A_\Delta)$ is updated by

$$\hat{Q}_k^{t+1}(S_\Delta, A_\Delta) = (1 - \alpha)\hat{Q}_k^t(S_\Delta, A_\Delta) + \alpha_t \left( r(S_\Delta, A_\Delta) + \gamma \max_{A'_\Delta \in \mathcal{A}_g \times \mathcal{A}_l^k} \hat{Q}_k^t(S'_\Delta, A'_\Delta) \right). \quad (44)$$

Note that the update in Equation (44) does not include an expectation and can be computed in a single trajectory via historical data. We make the following ergodicity assumption:

**Assumption I.1.** The behavior policy $\pi_b$ satisfies $\pi_b(A_\Delta|S_\Delta) > 0$ for all $(S_\Delta, A_\Delta) \in \mathcal{S}_g \times \mathcal{S}_l^k \times \mathcal{A}_g \times \mathcal{A}_l^k$ and the Markov chain $\mathcal{M}_{S_\Delta} = \{S_\Delta^{(t)}\}_{t \geq 0}$ induced by $\pi_b$ is irreducible and aperiodic with stationary distribution $\mu$ and mixing time $t_\delta(\mathcal{M}) = \min\{t \geq 0 : \max_{S_\Delta \in \mathcal{S}_g \times \mathcal{S}_l^k} \|P^t(S_\Delta, \cdot) - \mu(\cdot)\|_{TV} \leq \delta\}$. There are many heuristics of such behavior policies [Fujimoto et al., 2019].

**Theorem I.2.** *Let $\pi_k$ be the policy learned through off-policy $\hat{Q}_k$-learning. Then, under Assumption I.1, we have that that with probability at least $1 - \frac{1}{100e^k}$,*

$$\mathbb{E}[V^{\pi^*}(s_0) - V^{\pi_k}(s_0)] \leq \frac{\tilde{r}}{(1-\gamma)^2} \sqrt{\frac{n-k+1}{2nk}} \sqrt{\ln \frac{40\tilde{r}|\mathcal{S}_l||\mathcal{A}_l||\mathcal{A}_g|k^{|\mathcal{A}_l|+\frac{1}{2}}}{(1-\gamma)^2}} + \frac{1}{10\sqrt{k}} + 2\epsilon_{k,m}$$

$$= \tilde{O}(1/\sqrt{k}),$$

*where the randomness in the expectation is over the stochasticity of the exploration policy $\pi_b$.*

**Proposition I.3.** *Recall that the following are true:*

1. $\|\hat{Q}_k(F_{S_\Delta}, F_{A_\Delta}) - \hat{Q}_k(F_{S'_\Delta}, F_{A_\Delta})\| \leq \frac{1}{1-\gamma}\|r_l(\cdot, \cdot)\|_\infty \cdot \|F_{S'_\Delta} - F_{S_\Delta}\|_1$, *for any* $S_\Delta, S'_\Delta \in \mathcal{S}_g \times \mathcal{A}_l^k$ *and* $A_\Delta \in \mathcal{A}_g \times \mathcal{A}_l^k$ *by Theorem E.3,*

2. $\|\hat{Q}_k\| \leq \frac{\tilde{r}}{1-\gamma}$ *by Lemma C.4,*

3. $\|\hat{Q}_k^{t+1}(S_\Delta, A_\Delta) - \tilde{Q}_k^{t+1}(S_\Delta, A_\Delta)\|_\infty \leq \gamma\|\hat{Q}_k^t - \tilde{Q}_k^t(S_\Delta, A_\Delta)\|_\infty$ *by Lemma C.6,*

4. *The Markov chain* $\mathcal{M}_S$ *enjoys a rapid mixing property from Assumption I.1,*

Then, by treating the single trajectory update of the $\hat{Q}_k$-function as a noisy addition to the expected update from the ideal Bellman operator, Chen et al. [2021b] uses Markovian stochastic approximation to bound $\mathbb{E}[\|\hat{Q}_k^T - \hat{Q}_k^*\|_\infty^2]$. We restate their result:

**Theorem I.4** (Theorem 3.1 in Chen et al. [2021b] adapted to our setting). *Suppose* $\alpha_t = \alpha$ *for all* $t \geq 0$, *where* $\alpha$ *is chosen such that* $\alpha t_\alpha(\mathcal{M}_{S_\Delta}) \leq c_{Q,0}\frac{(1-\beta_1)^2}{\log|\mathcal{S}_g||\mathcal{S}_l^k||\mathcal{A}_g|\mathcal{A}_l^k|}$, *where* $c_{Q,0}$ *is a numerical constant. Then, under Proposition I.3, for all* $t \geq t_\alpha(\mathcal{M}_{S_\Delta})$, *we have*

$$\mathbb{E}[\|\hat{Q}_k^t - \hat{Q}_k^*\|_\infty^2] \leq c_{Q,1}\left(1 - \frac{(1-\gamma)\alpha}{2}\right)^{t-t_\alpha(\mathcal{M}_{S_\Delta})} + c_{Q,2}\frac{\log|\mathcal{S}_g||\mathcal{S}_l|^k|\mathcal{A}_g||\mathcal{A}_l|^k}{(1-\gamma)^2}\alpha t_\alpha(\mathcal{M}_{S_\Delta}),$$

*where* $c_{Q,1} = 3(\frac{\tilde{r}}{1-\gamma} + 1)^2$, $c_{Q,2} = 912e(\frac{3\tilde{r}}{1-\gamma} + 1)^2$, *and where the randomness in the expectation is over the randomness of the stochasticity of the behavior/exploration policy* $\pi_b$.

**Corollary I.5** (Corollary 3.2 in Chen et al. [2021b] adapted to our setting). *To make* $\mathbb{E}[\|\hat{Q}_k^t - \hat{Q}_k^*\|_\infty \leq \frac{1}{100\sqrt{k}}]$ *for* $\epsilon > 0$, *we need*

$$t > \tilde{O}\left(\frac{10000k\log^2(100\sqrt{k})|\mathcal{S}_g||\mathcal{S}_l|^k|\mathcal{A}_g||\mathcal{A}_l|^k\log|\mathcal{S}_g||\mathcal{A}_g||\mathcal{S}_l|^k|\mathcal{A}_l|^k}{(1-\gamma)^5}\right).$$

With this sample complexity, by the triangle inequality we also recover an expected-value analog of Theorem F.8.

**Corollary I.6.** *For* $\delta \in (0,1)^2$, *with probability at least* $1 - \delta$, *we have*

$$\mathbb{E}[\hat{Q}_k^*(s_g, F_{S_\Delta}, a_g, F_{a_\Delta}) - Q_n^*(s_g, F_{s_{[n]}, a_g, F_{a_{[n]}}})] \leq \frac{\ln\frac{2|\mathcal{S}_l||\mathcal{A}_l|}{\delta}}{1-\gamma}\sqrt{\frac{n-k+1}{8kn}}\|r_l\|_\infty,$$

*where the randomness in the expectation is over the stochasticity of the exploration policy* $\pi_b$.

In turn, following the argument in the proof of Theorem G.3, it is straightforward to verify that this leads to a result on the expected performance difference using off-policy learning:

**Corollary I.7.** *With probability at least* $1 - \frac{1}{100e^k}$,

$$\mathbb{E}[V^{\pi^*}(s_0) - V^{\pi_k}(s_0)] \leq \frac{\tilde{r}}{(1-\gamma)^2}\sqrt{\frac{n-k+1}{2nk}}\sqrt{\ln\frac{40\tilde{r}|\mathcal{S}_l||\mathcal{A}_l||\mathcal{A}_g|k^{|\mathcal{A}_l|+\frac{1}{2}}}{(1-\gamma)^2}} + \frac{1}{10\sqrt{k}} + 2\epsilon_{k,m}$$
$$= \tilde{O}(1/\sqrt{k}),$$

*where the randomness in the expectation is over the stochasticity of the exploration policy* $\pi_b$.

## J Extension to Continuous State/Action Spaces

Multi-agent settings where each agent handles a continuous state and action space find many applications in optimization, control, and synchronization.

**Example J.1** (Quadcopter Swarm Drone [Preiss et al., 2017]). *Consider a system of drones with a global controller, where each drone has to chase the controller, and the controller is designed to follow a bounded trajectory. Here, the state of each local agent* $i \in [n]$ *is its position and velocity in the bounded region, and the state of the global agent* $g$ *is a signal on its position and direction. The action of each local agent is a velocity vector* $a_l \in \mathcal{A}_l \subset \mathbb{R}^3$ *which is a bounded subset of* $\mathbb{R}^3$, *and the action of the global agent is a vector* $a_g \in \mathcal{A}_g \subset \mathbb{R}^2$ *which is a bounded subset of* $\mathbb{R}^2$.

Hence, this section is devoted to extending our algorithm and theoretical results to the case where the state and action spaces can be continuous (and therefore have infinite cardinality).

**Preliminaries.** For a measurable space $(\mathcal{S}, \mathcal{B})$, where $\mathcal{B}$ is a $\sigma$-algebra on $\mathcal{S}$, let $\mathbb{R}^{\mathcal{S}}$ denote the set of all real-valued $\mathcal{B}$-measurable functions on $\mathcal{S}$. Let $(X, d)$ be a metric space, where $X$ is a set and $d$ is a metric on $X$. A set $S \subseteq X$ is *dense* in $X$ if every element of $X$ is either in $S$ or a limit point of $S$. A set is *nowhere dense* in $X$ if the interior of its closure in $X$ is empty. A set $S \subseteq X$ is of Baire first category if $S$ is a union of countably many nowhere dense sets. A set $S \subseteq X$ is of Baire second category if $X \setminus S$ is of first category.

**Theorem J.2** (Baire Category Theorem [Jin et al., 2020]). Let $(X, d)$ be a complete metric space. Then any countable intersection of dense open subsets of $X$ is dense.

**Definition J.3** (Linear MDP). $\mathrm{MDP}(S, A, \mathbb{P}, r)$ is a linear MDP with feature map $\phi : \mathcal{S} \times \mathcal{A} \to \mathbb{R}^d$ if there exist $d$ unknown (signed) measures $\mu = (\mu^1, \ldots, \mu^d)$ over $\mathcal{S}$ and a vector $\theta \in \mathbb{R}^d$ such that for any $(s, a) \in \mathcal{S} \times \mathcal{A}$, we have

$$\mathbb{P}(\cdot | s, a) = \langle \phi(s, a), \mu(\cdot) \rangle,$$
$$r(s, a) = \langle \phi(s, a), \theta \rangle$$

**Assumption J.4** (Bounded features). Without loss of generality, we assume boundedness of the features: $\|\phi(s, a)\| \leq 1$, for all $(s, a) \in S \times A$ and $\max\{\|\mu(S)\|, \|\theta\|\} \leq \sqrt{d}$.

We motivate our analysis by reviewing representation learning in RL via spectral decompositions. For instance, if $P(s'|s, a)$ admits a linear decomposition in terms of some spectral features $\phi(s, a)$ and $\mu(s')$, then the $Q(s, a)$-function can be linearly represented in terms of the spectral features $\phi(s, a)$.

Then, through a reduction from Ren et al. [2024] that uses function approximation to learn the spectral features $\phi_k$ for $\hat{Q}_k$, we derive a performance guarantee for the learned policy $\pi_k^{\mathrm{est}}$, where the optimality gap decays with $k$:

**Theorem J.5.** *When $\pi_k^{\mathrm{est}}$ is derived from the spectral features $\phi_k$ learned in $\hat{Q}_k$, and $M$ is the number of samples used in the function approximation, then*

$$\Pr\left[V^{\pi^*}(s) - V^{\pi_k^{\mathrm{est}}}(s) \leq \tilde{O}\left(\frac{1}{\sqrt{k}} + \frac{\|\phi_k\|^5 \log 2k^2}{\sqrt{M}} + \frac{2\gamma\tilde{r}}{(1-\gamma)\sqrt{k}}\|\phi_k\|\right)\right] \geq 1 + \frac{1}{50k} - \frac{201}{100\sqrt{k}}$$

We assume the system is a linear MDP, where $\mathcal{S}_g$ and $\mathcal{S}_l$ are infinite compact sets. By a reduction from Ren et al. [2024] and using function approximation to learn the spectral features $\phi_k$ for $\hat{Q}_k$, we derive a performance guarantee for the learned policy $\pi_k^{\mathrm{est}}$, where the optimality gap decays with $k$.

**Assumption J.6.** Suppose $\mathcal{S}_g \subset \mathbb{R}^{\sigma_g}, \mathcal{S}_l \subset \mathbb{R}^{\sigma_l}, \mathcal{A}_g \subset \mathbb{R}^{\alpha_g}, \mathcal{A}_l \subset \mathbb{R}^{\alpha_l}$ are bounded compact sets. From the Baire category theorem, the underlying field $\mathbb{R}$ can be replaced to any set of Baire first category, which satisfies the property that there exists a dense open subset. In particular, replace Assumption 2.1 with a boundedness assumption.

Multi-agent settings where each agent handles a continuous state/action space find many applications in optimization, control, and synchronization.

**Lemma J.7** (Proposition 2.3 in Jin et al. [2020]). For any linear MDP, for any policy $\pi$, there exist weights $\{w^\pi\}$ such that for any $(s, a) \in \mathcal{S} \times \mathcal{A}$, we have $Q^\pi(s, a) = \langle \phi(s, a), \mathbf{w}^\pi \rangle$.

**Lemma J.8** (Proposition A.1 in Jin et al. [2020]). For any linear MDP, for any $(s, a) \in \mathcal{S} \times \mathcal{A}$, and for any measurable subset $\mathcal{B} \subseteq \mathcal{S}$, we have that

$$\phi(s, a)^\top \mu(\mathcal{S}) = 1,$$
$$\phi(s, a)^\top \mu(\mathcal{B}) \geq 0.$$

**Property J.9.** Suppose that there exist spectral representation $\mu_g(s'_g) \in \mathbb{R}^d$ and $\mu_l(s'_i) \in \mathbb{R}^d$ such that the probability transitions $P_g(s'_g, |s_g, a_g)$ and $P_l(s'_i|s_i, s_g, a_i)$ can be linearly decomposed by

$$P_g(s'_g|s_g, a_g) = \phi_g(s_g, a_g)^\top \mu_g(s'_g)$$
$$P_l(s'_i|s_i, s_g, a_i) = \phi_l(s_i, s_g, a_i)^\top \mu_l(s'_i)$$

for some features $\phi_g(s_g, a_g) \in \mathbb{R}^d$ and $\phi_l(s_i, s_g, a_i) \in \mathbb{R}^d$. Then, the dynamics are amenable to the following spectral factorization, as in Ren et al. [2024].

**Lemma J.10.** $\hat{Q}_k$ admits a linear representation.

*Proof.* Given the factorization of the dynamics of the multi-agent system, we have:

$$\mathbb{P}(s'|s,a) = P_g(s'_g|s_g,a_g) \cdot \prod_{i=1}^{n} P_l(s'_i|s_i,s_g,a_i)$$

$$= \langle \phi_g(s_g,a_g), \mu_g(s'_g) \rangle \cdot \prod_{i=1}^{n} \langle \phi_l(s_i,s_g,a_i), \mu_l(s'_i) \rangle$$

$$= \phi_g(s_g,a_g), \mu_g(s'_g) \rangle \cdot \langle \otimes_{i=1}^{n} \phi_l(s_i,s_g,a_i), \otimes_{i=1}^{n} \mu_l(s'_i) \rangle$$

$$:= \langle \bar{\phi}_n(s,a), \bar{\mu}_n(s') \rangle$$

Similarly, for any $\Delta \subseteq [n]$ where $|\Delta| = k$, the subsystem consisting of $k$ local agents $\Delta$ has a subsystem dynamics given by

$$\mathbb{P}(s'_\Delta, s_g | s_\Delta, s_g, a_g, a_\Delta) = \langle \bar{\phi}_k(s_g, s_\Delta, a_g, a_\Delta), \bar{\mu}_k(s'_\Delta) \rangle.$$

Therefore, $\hat{Q}_k$ admits the linear representation:

$$\hat{Q}_k^{\pi_k}(s_g, F_{z_\Delta}, a_g) = r_g(s_g, a_g) + \frac{1}{k} \sum_{i \in \Delta} r_l(s_i, s_g, a_i) + \gamma \mathbb{E}_{s'_g, s_{\Delta'}} \max_{a'_g, a'_\Delta} \hat{Q}_k^{\pi_k}(s'_g, F_{z'_\Delta}, a'_g)$$

$$= r_g(s_g, a_g) + \frac{1}{k} \sum_{i \in \Delta} r_l(s_i, s_g, a_i) + (\mathbb{P}V^{\pi_k}(s_g, F_{s_\Delta}, a_g))$$

$$= \begin{bmatrix} r_\Delta(s,a) \\ \bar{\phi}_k(s_g, s_\Delta, a_g, a_\Delta) \end{bmatrix} \begin{bmatrix} 1 & \gamma \int_{s'_\Delta} \mu_k(s'_\Delta) V^{\pi_k}(s'_\Delta) ds'_\Delta \end{bmatrix},$$

proving the claim. $\square$

Therefore, $\bar{\phi}_k$ serves as a good representation for the $\hat{Q}_k$-function making the problem amenable to the classic linear functional approximation algorithms, consisting of *feature generation* and *policy gradients*.

In feature generation [Ren et al., 2024], we generate the appropriate features $\phi$, comprising of the local reward functions and the spectral features coming from the factorization of the dynamics. In applying the policy gradient, we perform a gradient step to update the local policy weights $\theta_i$ and update the new policy.

For this, we update the weight $w$ via the TD(0) target semi-gradient method (with step size $\alpha$), to get

$$w_{t+1} = w_t + \alpha(r_\Delta(s^t, a^t) + \gamma \bar{\phi}_k(s_\Delta^{t+1})^\top w_t - \bar{\phi}_k(s_\Delta^t)^\top w_t) \bar{\phi}_k(s_\Delta^t) \tag{45}$$

**Definition J.11.** Let the complete feature map be denoted by $\Phi : \mathcal{S} \times d \to \mathbb{R}$, and let the subsampled feature map be denoted by $\hat{\Phi}_k : \mathcal{S}_g \times \mathcal{S}_l^k \times d \in \mathbb{R}$, where we let

$$\hat{\Phi}_k = \begin{bmatrix} \bar{\Phi}_k(1) \\ \bar{\Phi}_k(2) \\ \vdots \\ \bar{\Phi}_k(|\mathcal{S}_g| \times |\mu_{k-1}(\mathcal{S}_l)|) \end{bmatrix} \in \mathbb{R}^{|\mathcal{S}_g| \times |\mu_{k-1}(\mathcal{S}_l)| \times d} \tag{46}$$

Here, the system's stage rewards are given by

$$r = [r(1) \quad \dots \quad r(S)] \in \mathbb{R}^S$$

and

$$r_k = [r_k(1), \dots, r_k(|\mathcal{S}_g| \times |\mu_{k-1}(\mathcal{S}_l)|] \in \mathbb{R}^{|\mathcal{S}_g| \times |\mu_{k-1}(\mathcal{S}_l)|},$$

where $d$ is a low-dimensional embedding we wish to learn.

Agnostically, the goal is to approximate the value function through the approximation

$$V_w \approx \hat{\Phi}_k w = \begin{bmatrix} \hat{\Phi}_k(1) \\ \vdots \\ \hat{\Phi}_k(|\mathcal{S}_g| \times |\mu_{k-1}(\mathcal{S}_l)) \end{bmatrix} w \in \text{span}(\hat{\Phi}_k)$$

This manner of updating the weights can be considered via the projected Bellman equation $\hat{\Phi}_k w = \Pi_\mu \mathcal{T}(\hat{\Phi}_k w)$, where $\Pi_\mu(v) = \arg\min_{z \in \hat{\Phi}_k w} \|z - v\|_\mu^2$. Notably, the fixed point of the the projected Bellman equation satisfies

$$w = (\hat{\Phi}_k^\top D_\mu \hat{\Phi}_k)^{-1} \hat{\Phi}_k^\top D_\mu (r + \gamma P^\pi \hat{\Phi}_k w),$$

where $D_\mu$ is a diagonal matrix comprising of $\mu$'s. In other words, $D_\mu = \text{diag}(\mu_1, \ldots, \mu_n)$. Then,

$$(\hat{\Phi}_k^\top D_\mu \hat{\Phi}_k) w = \hat{\Phi}_k^\top D_\mu (r + \gamma P^\pi \hat{\Phi}_k w)$$

In turn, this implies

$$\hat{\Phi}_k^\top D_\mu (I - \gamma P^\pi) \Phi w = \hat{\Phi}_k^\top D_\mu r.$$

Therefore, the problem is amenable to Algorithm 1 in Ren et al. [2024]. To bound the error of using linear function approximation to learn $\hat{Q}_k$, we use a result from Ren et al. [2024].

**Lemma J.12** (Policy Evaluation Error, Theorem 6 of Ren et al. [2024]). *Suppose the sample size $M \geq \log\left(\frac{4n}{\delta}\right)$, where $n$ is the number of agents and $\delta \in (0, 1)$ is an error parameter. Then, with probability at least $1 - 2\delta$, the ground truth $\hat{Q}_k^\pi(s, a)$ function and the approximated $\hat{Q}_k$-function $\hat{Q}_k^{\text{LFA}}(s, a)$ satisfies, for any $(s, a) \in \mathcal{S} \times \mathcal{A}$,*

$$\mathbb{E}\left[\left|\hat{Q}_k^\pi(s, a) - \hat{Q}_k^{\text{LFA}}(s, a)\right|\right] \leq O\left(\underbrace{\log\left(\frac{2k}{\delta}\right) \frac{\|\bar{\phi}_k\|^5}{\sqrt{M}}}_{\text{statistical error}} + \underbrace{2\epsilon_P \gamma \cdot \frac{\tilde{r}}{1 - \gamma} \cdot \|\bar{\phi}_k\|}_{\text{approximation error}}\right),$$

*where $\epsilon_P$ is the error in approximating the spectral features $\phi_g, \phi_l$.*

**Corollary J.13.** Therefore, when $\pi_{k,m}$ is derived from the spectral features learned in $\hat{Q}_k^{\text{LFA}}$, applying the triangle inequality on the Bellman noise and setting $\delta = \epsilon_P = \frac{1}{\sqrt{k}}$ yields[5]

$$\Pr\left[V^{\pi^*}(s_0) - V^{\pi_{k,m}}(s_0) \leq \tilde{O}\left(\frac{1}{\sqrt{k}} + \log\left(2k^2\right) \frac{\|\bar{\phi}_k\|^5}{\sqrt{M}} + \frac{2}{\sqrt{k}} \cdot \frac{\gamma \tilde{r}}{1 - \gamma} \|\bar{\phi}_k\|\right)\right] \geq 1 + \frac{1}{50k} - \frac{201}{100\sqrt{k}}$$

Using $\|\tilde{\phi}_k\| \leq 1$ and simplifying gives

$$\Pr\left[V^{\pi^*}(s_0) - V^{\pi_{k,m}}(s_0) \leq \tilde{O}\left(\frac{1}{\sqrt{k}} + \frac{\log\left(2k^2\right)}{\sqrt{M}} + \frac{2\gamma \tilde{r}}{\sqrt{k}(1 - \gamma)}\right)\right] \geq 1 - \frac{201}{100\sqrt{k}}$$

**Remark J.14.** Hence, in the continuous state/action space setting, as $k \to n$ and $M \to \infty$, $V^{\tilde{\pi}_k}(s_0) \to V^{\tilde{\pi}_n}(s_0) = V^{\pi^*}(s_0)$. Intuitively, as $k \to n$, the optimality gap diminishes following the arguments in Corollary G.7. As $M \to \infty$, the number of samples obtained allows for more effective learning of the spectral features.

# K Towards Deterministic Algorithms: Sharing Randomness

In large distributed systems, the random sampling in our communication network may be a bottleneck for "decentralized execution". In light of this, we have provided a practical derandomized heuristic where the agents can share some randomness by only sampling within pre-defined fixed blocks of size.

---

[5]Following Ren et al. [2024], the result easily generalizes to any positive-definite transition Kernel noise (e.g. Laplacian, Cauchy, Matérn, etc.

---

**Algorithm 5** SUB-SAMPLE-MFQ: Execution with weakly shared randomness

---

**Require:** A multi-agent system as described in Section 2. A distribution $s_0$ on the initial global state $s_0 = (s_g, s_{[n]})$. Parameter $T'$ for the number of iterations for the decision-making sequence. Hyperparameter $k \in [n]$. Discount parameter $\gamma$. Policy $\hat{\pi}_{k,m}^{\text{est}}(s_g, s_\Delta)$.

1: Sample $(s_g(0), s_{[n]}(0)) \sim s_0$.
2: Define groups $h_1, \ldots, h_x$ of agents where $x := \lceil \frac{n}{k} \rceil$ and $|h_1| = |h_2| = \cdots = |h_{x-1}| = k$ and $|h_x| = n \mod k$.
3: **for** $t = 0$ to $T' - 1$ **do**
4:    **for** $i \in [x]$ **do**
5:       Let $\Delta_i$ be a uniform sample of $\binom{[n] \backslash g_i}{k-1}$, and let $a_{g_i}(t) = [\hat{\pi}_{k,m}^{\text{est}}(s_g(t), s_{\Delta_i}(t))]_{1:k}$.
6:       Let $a_g(t) = \text{majority} \left( \left\{ [\hat{\pi}_{k,m}^{\text{est}}(s_g(t), s_{\Delta_i}(t))]_g \right\}_{i \in [x]} \right)$.
7:       Let $s_g(t+1) \sim P_g(\cdot | s_g(t), a_g(t))$
8:       Let $s_i(t+1) \sim P_l(\cdot | s_i(t)$
9:       $s_g(t), a_i(t))$, for all $i \in [n]$.
10:      Get reward $r(s, a) = r_g(s_g, a_g) + \frac{1}{n} \sum_{i \in [n]} r_l(s_i, a_i, s_g)$

---

**Algorithm 6** SUB-SAMPLE-MFQ: Execution with strongly shared randomness

---

**Require:** A multi-agent system as described in Section 2. A distribution $s_0$ on the initial global state $s_0 = (s_g, s_{[n]})$. Parameter $T'$ for the number of iterations for the decision-making sequence. Hyperparameter $k \in [n]$. Discount parameter $\gamma$. Policy $\hat{\pi}_{k,m}^{\text{est}}(s_g, s_\Delta)$.

1: Sample $(s_g(0), s_{[n]}(0)) \sim s_0$.
2: Define groups $h_1, \ldots, h_x$ of agents where $x := \lceil \frac{n}{k} \rceil$ and $|h_1| = |h_2| = \cdots = |h_{x-1}| = k$ and $|h_x| = n \mod k$.
3: **for** $t = 0$ to $T' - 1$ **do**
4:    **for** $i \in [x-1]$ **do**
5:       Let $a_{h_i}(t) = [\hat{\pi}_{k,m}^{\text{est}}(s_g(t), s_{h_i}(t))]_{1:k}$.
6:       Let $a_{h_i}^g(t) = [\hat{\pi}_{k,m}^{\text{est}}(s_g(t), s_{h_i}(t))]_g$.
7:    Let $\Delta_{\text{residual}} := \binom{[n] \backslash h_x}{k - (n - \lceil n/k \rceil k)}$.
8:    Let $a_{h_x}(t) = [\hat{\pi}_{k,m}^{\text{est}}(s_g(t), s_{h_x \cup \Delta_{\text{residual}}}(t))]_{1:n - \lceil n/k \rceil k}$
9:    Let $a_{h_x}^g(t) = [\hat{\pi}_{k,m}^{\text{est}}(s_g(t), s_{h_x \cup \Delta_{\text{residual}}}(t))]_g$
10:   Let $a_g(t) = \text{majority} \left( \{ a_{h_i}^g(t) : i \in [x] \} \right)$.
11:   Let $s_g(t+1) \sim P_g(\cdot | s_g(t), a_g(t))$, $s_i(t+1) \sim P_l(\cdot | s_i(t), s_g(t), a_i(t))$, for all $i \in [n]$.
12:   Get reward $r(s, a) = r_g(s_g, a_g) + \frac{1}{n} \sum_{i \in [n]} r_l(s_i, a_i, s_g)$

---

In this section, we propose algorithms RANDOM-SUB-SAMPLE-MFQ and RANDOM-SUB-SAMPLE-MFQ+, which shares randomness between agents through various careful groupings and using shared randomness within each groups. By implementing these algorithms, we derive simulation results, and establish progress towards derandomizing inherently stochastic approximately-optimal policies.

**Algorithm 5** (Execution with weakly shared randomness). The local agents are heuristically divided into arbitrary groups of size $k$. For each group $h_i$, the $k - 1$ local agents sampled are the same at each iteration. The global agent's action is then the majority global agent action determined by each group of local agents. At each round, this requires $O(\lceil n^2/k \rceil (n - k))$ random numbers.

**Algorithm 6** (Execution with strongly shared randomness). The local agents are randomly divided in to groups of size $k$. For each group, each agent uses the other agents in the group as the $k - 1$ other sampled states. Similarly, the global agent's action is then the majority global agent action determined by each group of local agents. Here, at each round, this requires $O(\lceil n^2/k \rceil)$ random numbers.

The probability of a bad event (the policy causing the $Q$-function to exceed the Lipschitz bound) scales with the $O(n^k)$ independent random variables, which is polynomially large. Agnostically, some randomness is always needed to avoid periodic dynamics that may occur when the underlying

Markov chain is not irreducible. In this case, an adversarial reward function could be designed such that the system does not minimize the optimality gap, thereby penalizing excessive determinism.

This ties into an open problem of interest, which has been recently explored [Larsen et al., 2024]. What is the minimum amount of randomness required for SUBSAMPLE-MFQ (or general mean-field or multi-agent RL algorithms) to perform well? Can we derive a theoretical result that demonstrates and balances the tradeoff between the amount of random bits and the performance of the subsampled policy when using the greedy action from the $k$-agent subsystem derived from $\hat{Q}_k^*$? We leave this problem as an exciting direction for future research.

