# OpenReview forum: "Mean-Field Sampling for Cooperative Multi-Agent Reinforcement Learning"
_NeurIPS.cc/2025/Conference — NeurIPS 2025 spotlight_

### Official Review · Reviewer_PDRU · 2025-07-01

**Clarity:** 4
**Significance:** 4
**Originality:** 3
**Rating:** 5
**Confidence:** 4

**Summary:**

This paper proposes a novel algorithm, \textbf{SUBSAMPLE-MFQ (Subsample-Mean-Field Q-learning)}, to address the scalability challenges in cooperative multi-agent reinforcement learning (MARL). Traditional MARL suffers from exponential complexity in the number of agents, and although mean-field MARL reduces this to polynomial complexity, it remains computationally prohibitive for large-scale systems.

The authors introduce a sampling-based approximation strategy: by selecting a small number $k \leq n$ of local agents and applying value iteration on this reduced system, they learn a policy that can be extended stochastically to the full agent population. The key theoretical contribution is showing that the performance gap between the learned policy and the global optimum decays at a rate of $\widetilde{O}(1/\sqrt{k})$, independent of the total number of agents $n$. Thus, by choosing $k = O(\log n)$, the algorithm achieves a \textit{polylogarithmic runtime} in $n$, representing an exponential speedup over previous approaches. The paper also extends this framework to off-policy learning and non-tabular settings.

**Questions:**

1. Can the authors provide more comprehensive experiments on common MARL benchmarks (e.g., MAgent, PettingZoo, SMAC) to evaluate practical performance?

2. How does the method scale or degrade with increasing agent heterogeneity? Can you support this with empirical results?

3. Could the authors discuss how the learned policies can be implemented with function approximators like neural networks in practical settings?

4. Is there any guidance on how to select $k$ in real-world tasks to balance performance and computational efficiency?
How does SUBSAMPLE-MFQ compare empirically to recent centralized training and decentralized execution (CTDE) methods?

**Ethical Concerns:**

["NO or VERY MINOR ethics concerns only"]

**Final Justification:**

The authors response the questions clearly, and it is a good thoery work in RL.

**Limitations:**

Yes.

**Paper Formatting Concerns:**

No.

**Quality:**

3

**Strengths And Weaknesses:**

Strengths

1. Theoretical contributions are strong and rigorous. The authors provide a provable convergence rate, detailed complexity bounds, and analysis for both tabular and non-tabular settings.

2. The approach is sound and novel, building a strong bridge between mean-field theory and sampling-based approximation in MARL.

3. The paper includes clear algorithmic details, including formal descriptions (Algorithm 1 & 2), and supportive mathematical analysis.

Weaknesses

1. The experimental section is minimal and primarily relegated to the appendix. There is no strong empirical validation of the claimed exponential speedup or policy performance on standard benchmarks.

2. While the paper discusses extension to heterogeneity, the current algorithm assumes mostly homogeneous agent types.

3. Real-world deployment constraints (e.g., communication or decentralized execution under limited observability) are not empirically studied.

---

> ### Author Rebuttal · Authors · 2025-07-29
>
> We thank reviewer PDRU for their feedback. Please find our response below:
>
> >While the paper discusses extension to heterogeneity, the algorithm assumes mostly homogeneous agent types.
>
> Indeed, while the current algorithm can only work for the homogeneous setting, we add two caveats:
> 1) As the reviewer points out, we mention an extension to a partially heterogeneous-agent setting by modeling types as a part of the state/action spaces of the local agents, in the “capturing heterogeneity” paragraph in page 4 and Remark 4.7.
> 2) There has been a line of recent work on extending mean-field MARL using a graphon-based framework that supports a continuum of distinct transition probabilities (which is also a form of heterogeneity). It may be possible to extend our subsampling framework to graphon mean-field MARL. Adapting this to our sampling framework may be highly non-trivial, and we leave this as an open problem for future works. Nonetheless, we can add a detailed discussion on this as part of Remark 4.7.
>
> >Can the authors provide more comprehensive experiments on common MARL benchmarks (e.g., MAgent, PettingZoo, SMAC) to evaluate practical performance?
>
> As our work is a theoretical framework to the NeurIPS theory track submission, we only provided sufficient experiments to numerically validate the theory. That being said, an even more thorough empirical study of cooperative multi-agent RL algorithms is an interesting avenue for future research for empirical research, however, this is outside of the scope of our current work. Moreover, the above benchmarks (MAgent, PettingZoo, SMAC) do not fit our MARL setting (for instance, SMAC requires heterogeneous agents, MAgent requires competitive agents, and PettingZoo does not support our global-agent/local-agent setup).
>
> >How does the method scale/degrade with increasing agent heterogeneity? Can you support this with empirical results?
>
> Increasing the agent heterogeneity using our “type-formulation” makes the algorithm somewhat more expensive since it factors into the query complexity via the state space of the local agents. **However**, since the optimality gap of the learned policy is on the order of $\tilde{O}(1/\sqrt{k})$ (modulo very small $\sqrt{\log(|\mathcal{S}_l| |\mathcal{A}_l|)}$ factors), **increasing the amount of agent heterogeneity does not degrade the quality of the learned policy as the theoretical bound does not get worse.**
>
> >Could the authors discuss how the learned policies can be implemented with function approximators like neural networks in practical settings?
>
> Learning the policy and sharing it to all the agents is the computational bottleneck in practical settings. After this, it is efficient to implement the learned policies since the empirical distribution only takes $O(k)$ time to compute, and this dependency is tight. In Appendix J, we extended our algorithm to continuous spaces. In this set-up, the learned policy could be  implemented with neural operators and other function approximators to parameterize and evaluate the policy.
>
> >Is there any guidance on how to select k in real-world tasks to balance performance and computational efficiency? How does SUBSAMPLE-MFQ compare empirically to recent centralized training and decentralized execution (CTDE) methods?
>
> Theoretically, $k = O(\log n)$ suffices to have a decaying error rate with best efficiency, as we mention in Remark 4.6, and as is validated by our experiments. To the best of our knowledge, there is no previous CTDE method that utilized a similar parameterized (with respect to $k$) method for learning a MARL policy in the full-observation setting.
>
> We hope this answers your questions, and look forward to further discussions.
>
> Thank you,
>
> Authors

---

> > ### Author Response · Authors · 2025-08-05
> > **Follow-up on Author Response**
> >
> > Dear Reviewer PDRU,
> >
> > We hope this message finds you well.
> >
> > As the author-reviewer discussion period is coming to an end, we would like to kindly ask whether we have adequately addressed your concerns. If our responses have adequately addressed your concerns, we kindly ask you to consider updating your score.
> >
> > If there are any remaining concerns, we would greatly appreciate it if you could share them with us, so we may have enough time to provide a comprehensive response. Thank you once again for your time and valuable feedback!
> >
> > Thanks,
> >
> > Authors

---

### Official Review · Reviewer_1oj9 · 2025-07-03

**Clarity:** 3
**Significance:** 4
**Originality:** 3
**Rating:** 5
**Confidence:** 4

**Summary:**

This paper addresses a challenging and important problem in cooperative multi-agent reinforcement learning (MARL) by proposing a Subsample-Mean-Field Q-learning algorithm that decouples the global dependence among agents through random subsampling. The paper establishes a performance gap of $O(1/\sqrt{k})$, where $k$ is the number of sampled agents with improved sample complexity. This is a strong and meaningful result in cooperative MARL under the homogeneous agent setting.

**Questions:**

1) The algorithm's sample complexity appears to be less relevant to the time horizon $T$, which is unusual in RL theory literature. If I understand correctly, it may be due to the large number of samples used in the Bellman updates. It would be beneficial to provide a detailed discussion on this topic.

2) In the learning period, the algorithm still seems to require a centralized way to do value iteration and (greedy) policy improvement; it would be better to explain how it can be implemented in a practical setting.

3) While the proposed method is designed for homogeneous agents, it would be valuable for the paper to discuss how the approach might be extended to heterogeneous settings, especially when the agent types are numerous or drawn from a continuous space. Even a discussion of potential challenges or directions would strengthen the broader applicability of the work.

**Ethical Concerns:**

["NO or VERY MINOR ethics concerns only"]

**Final Justification:**

I think it is a nice theory paper in MARL, and I am very positive about it.

**Quality:**

4

**Strengths And Weaknesses:**

Strengths
* The paper tackles an important and challenging problem in cooperative MARL. The proposed algorithm is intuitive, and the performance guarantee of $O(1/\sqrt{k})$ with improved sample complexity is a great theoretical contribution to cooperative MARL, which might be leveraged to understand more complex settings.

Weaknesses
* The paper may require a simulator to generate samples, and the number of samples per iteration ($m$) could be very large, scaling exponentially with the size of the local state and action spaces. The method might be limited to the homogeneous setting.

---

> ### Author Rebuttal · Authors · 2025-07-29
>
> We thank reviewer 1oj9 for their feedback. Please find our response below:
>
> >The paper may require a simulator to generate samples, and the number of samples per iteration could be very large, scaling exponentially with the size of the local state and action spaces. The method might be limited to the homogeneous setting.
>
> Our algorithm assumes the existence of a simulator to generate samples. As the reviewer points out, this adds an $O(1)$ cost per iteration, resulting in a large query complexity. However, this query complexity was previously exponential in the number of local agents and in the state/action spaces, whereas our work reduces this to a polynomial dependence with the number of local agents and a sub-exponential dependence on the state/action spaces. Moreover, Remark 4.5 and Appendix I details a version of the algorithm that relaxes the assumption of having a generative oracle by only utilizing historical data (by reducing to off-policy $Q$-learning).
>
> >The algorithm's sample complexity appears to be less relevant to the time horizon , which is unusual in RL theory literature. If I understand correctly, it may be due to the large number of samples used in the Bellman updates. It would be beneficial to provide a detailed discussion on this topic.
>
> In Appendix G.1, we show that a necessary condition of $T>\frac{2}{1-\gamma} \log \frac{\tilde{r}\sqrt{k}}{1-\gamma}$ is sufficient to allow the Bellman error to be on the order of $\tilde{O}(1/\sqrt{k})$. This enables Lemma 4.3 in the main body which allows us to abstract away the dependence on the time-horizon. We can add a parallel derivation that retains the $T$ factor via a $\gamma^T$ dependence in the final bound (which shows an exponentially decaying error with the time horizon $T$). Moreover, in Lemma G.10, we show that the query complexity is on the order of $O(mT|S_g||S_l|^k|A_g||A_l|^k)$, and we bound $T$ and set $m={\text{poly}}(1/(1-\gamma))$ to attain the $1/\sqrt{k}$ rate. We can make these dependencies more explicit in Theorem 4.2 in the final version of this paper.
>
> Additionally, we can explain that our polynomial dependence on $\frac{1}{1-\gamma}$ may be loose since we do not use more complicated variance reduction techniques as in [1, 2, 3, 4] to optimize the number of samples $m$ which is used, in turn, to bound the Bellman error $\epsilon_{k,m}$. This was primarily because our goal was to analyze the necessary number of sampled agents as well as to keep complex algorithms as simple as possible (and incorporating variance reduction would significantly complicate the algorithm and intuition).
>
> >In the learning period, the algorithm still seems to require a centralized way to do value iteration and (greedy) policy improvement; it would be better to explain how it can be implemented in a practical setting.
>
> Centralized learning is the only provable way for a multi-agent system to learn the true optimum policy in the full-information setting, as the optimum policy $\pi^*$ lives in a set whose cardinality depends on the joint state/action space. To clarify, there exists decentralized-training MARL algorithms with provable convergence guarantees to some equilibrium policy; however, these algorithms generally do not converge to the optimum policy. Since our algorithm only requires centralized training on $k\leq n$ agents, it is more scalable than previous optimum-searching policies. One practical suggestion that exploits the homogeneity of the local agents is to allow each local agent to compute the joint $Q$-function instead of having a controller share the learned policy with all the agents: this leads to a natural trade-off between the communication complexity and the sample complexity of the algorithm which can be optimized for in various practical settings, and we can add this as a discussion in Remark 4.8 in the main body.
>
> >While the proposed method is designed for homogeneous agents, it would be valuable for the paper to discuss how the approach might be extended to heterogeneous settings, especially when the agent types are numerous or drawn from a continuous space. Even a discussion of potential challenges or directions would strengthen the broader applicability of the work.
>
> Indeed, while the current algorithm can only work for the homogeneous setting, we add two caveats:
> 1) As the reviewer points out, we mention an extension to a partially heterogeneous-agent setting by modeling types as a part of the state/action spaces of the local agents, in the “capturing heterogeneity” paragraph in page 4 and Remark 4.7.
> 2) There has been a line of recent work on extending mean-field MARL using a graphon-based framework that supports a continuum of distinct transition probabilities (which is also a form of heterogeneity). It may be possible to extend our subsampling framework to graphon mean-field MARL. Adapting this to our sampling framework may be highly non-trivial, and we leave this as an open problem for future works. Nonetheless, we can add a detailed discussion on this as part of Remark 4.7.
>
> We hope this answers your questions, and look forward to further discussions.
>
> Thank you,
>
>
> Authors
>
> ====================
>
> References:
>
> [1] Variance Reduced Value Iteration and Faster Algorithms for Solving Markov Decision Processes (Sidford et. al., SODA 2018)
>
> [2] Near-Optimal Time and Sample Complexities for Solving Discounted Markov Decision Process with a Generative Model  (Sidford et. al., NeurIPS 2018)
>
> [3] Truncated Variance Reduced Value Iteration (Jin et. al., NeurIPS 2024)
>
> [4] Variance-reduced Q-learning is minimax optimal (Mainwright, arXiV 2019)

---

> > ### Comment · Reviewer_1oj9 · 2025-08-02
> >
> > Thank you for the detailed response! It address my concerns and I think it is a nice theory paper in MARL.

---

### Official Review · Reviewer_GLs2 · 2025-07-05

**Clarity:** 4
**Significance:** 3
**Originality:** 3
**Rating:** 4
**Confidence:** 4

**Summary:**

This paper addresses the fundamental challenge of scalability in cooperative multi-agent reinforcement learning (MARL), where the joint state-action space grows exponentially with the number of agents, n. The authors propose a novel algorithm, SUBSAMPLE-MFQ, that tackles this curse of dimensionality by leveraging mean-field theory and subsampling. The main contribution of the paper is a rigorous theoretical proof showing that the performance gap between the learned policy and the globally optimal policy decays at a rate of O(1/ square root of k). This error bound is independent of the total number of agents n, establishing a clear trade-off between computational complexity and sub-optimality.

**Questions:**

-  The theoretical results are very impressive. To better contextualize their impact, could you discuss how you expect SUBSAMPLE-MFQ to perform on standard, complex MARL benchmarks like the StarCraft Multi-Agent Challenge (SMAC)? How does the structural assumption of your model (a global agent and non-interacting local agents) map to such environments, and what challenges would arise in applying your method there?

- The paper proves an optimality gap of  O~(1/ square root of k). In the limitations, you correctly note the absence of a matching lower bound. Do you have any intuition on whether this rate is tight for this problem? Or could a different sampling or estimation strategy potentially achieve a faster rate, such as O(1/k)?

**Ethical Concerns:**

["NO or VERY MINOR ethics concerns only"]

**Limitations:**

yes

**Paper Formatting Concerns:**

No concerns

**Quality:**

3

**Strengths And Weaknesses:**

Strengths
- This paper presents a significant and original theoretical breakthrough for scalable MARL. The idea of combining mean-field approximation with agent subsampling to achieve complexity that is independent of the total number of agents n is a powerful and novel concept. This directly tackles the most critical bottleneck in MARL—the curse of dimensionality—and provides a new and viable path toward creating provably good algorithms for systems with massive numbers of agents.

-The main theoretical result (Theorem 4.4) is very strong. Proving an optimality gap that does not depend on n is a powerful guarantee. The proof appears rigorous and leverages a non-trivial combination of techniques, including a novel MDP sampling result based on an extension of the DKW inequality and a careful adaptation of the performance difference lemma to the multi-agent sampling context. The analysis is extensive, with the appendices covering extensions to stochastic rewards, off-policy learning, and even continuous spaces, demonstrating the robustness of the framework.

- For a deeply theoretical paper, the presentation is exceptionally clear. The authors do an excellent job of motivating the problem, providing high-level intuition for their approach, and then systematically building up the formalisms.

- The result establishes a fundamental trade-off between computation (polynomial in k) and optimality. By setting k=O(\log n), the paper demonstrates an exponential speedup over prior mean-field methods while maintaining a provably small, decaying optimality gap. This could inspire a new class of practical, scalable MARL algorithms.

Weaknesses
-The primary weakness of the paper is its experiments. The simulations are run on two relatively simple, synthetic environments ("Gaussian squeeze" and "constrained exploration") designed to illustrate the theory. The paper lacks evaluation on any standard, complex MARL benchmarks (e.g., StarCraft, Multi-Agent MuJoCo, etc.). While the results effectively demonstrate the claimed trade-off, they do not show how SUBSAMPLE-MFQ compares to or performs against established, practical MARL algorithms in more realistic settings.

-The theoretical results are derived for a specific cooperative MARL setting with homogeneous local agents and a structured reward and transition dynamic (agents do not directly interact with each other, only through the global state). While the authors mention that heterogeneity can be modeled by including a "type" in the state space, this may not be sufficient for all real-world scenarios. The restriction to purely cooperative tasks also limits the immediate applicability to competitive or mixed-motive games.

-The proposed online execution policy (Algorithm 2) requires each of the n agents to sample and observe the states of k-1 other agents at every timestep. While this is tractable from a computational perspective, it assumes a communication or observation network that can support this random sampling at scale. In a massively distributed system, this requirement could itself become a communication bottleneck, which complicates the "decentralized execution" claim.

---

> ### Author Rebuttal · Authors · 2025-07-29
>
> We thank reviewer GLs2 for their feedback on our work.
>
> >While this is tractable from a computational perspective, the decentralized execution assumes a communication or observation network that can support this random sampling at scale. In a massively distributed system, this requirement could itself become a communication bottleneck, which complicates the "decentralized execution" claim.
>
> Thanks for pointing this out. We can add some discussion on heuristics that derandomize the decentralized execution method, which may be more conductive for practical settings. Specifically, one practical implementation of the algorithm involves allowing the agents to share some randomness by only sampling within pre-defined fixed blocks of size $k$. We can add more detail on this and add this as a new section in the appendix.
>
>
> >To better contextualize their impact, could you discuss how you expect SUBSAMPLE-MFQ to perform on standard, complex MARL benchmarks like the StarCraft Multi-Agent Challenge (SMAC)?  How does the structural assumption of your model (a global agent and non-interacting local agents) map to such environments, and what challenges would arise in applying your method there?
>
> As the reviewer points out, the primary contributions of our work are theoretical, and we only provided simulations sufficient to numerically validate the theoretical result. For instance, the “Gaussian squeeze” experiment is similar to one conducted in the foundational mean-field MARL paper [1] which we ablate against.  An even more thorough empirical study of cooperative multi-agent RL algorithms would be an interesting avenue for future research for empirical research, however, this is outside of the scope of our current work which focuses on some theoretical bottlenecks of MARL. At the present, some structural assumptions which resist applications of our methods in some standard MARL benchmarks involve the fact that our method only incorporates weak heterogeneity, as well as the fact that it is designed for cooperative multi-agent learning rather than more competitive/game environments.
>
> >Regarding the optimality gap, do we have any intuition on whether this rate is tight for this problem? Or could a different sampling or estimation strategy potentially achieve a faster rate, such as O(1/k)?
>
> The algorithmic bottleneck in achieving a faster rate than $O(1/\sqrt{k})$ is actually from learning the $\hat{Q}_k$ function, rather than the online execution strategy; however, when $k = n$, our algorithm reduces to mean-field learning which has a rate of $O(1/\sqrt{n})$ (and is known to be tight in light of [1]). Thus, at least when $k =n$, our $O(1/\sqrt{k})$ rate is tight. This illustrates a natural obstacle against uniformly improving our $O(1/\sqrt{k})$ rate and perhaps suggests intuition for why our result might be tight.
>
> We would be happy to answer any further questions.
>
> Thank you,
>
> Authors
>
> ========================
>
> References:
>
> [1] Mean Field Multi-Agent Reinforcement Learning, Yang et. al. (ICML 2018)

---

> > ### Author Response · Authors · 2025-08-05
> > **Follow-up on Author Response**
> >
> > Dear Reviewer GLs2,
> >
> > We hope this message finds you well.
> >
> > As the author-reviewer discussion period is coming to an end, we would like to kindly ask whether we have adequately addressed your concerns. If our responses have adequately addressed your concerns, we kindly ask you to consider updating your score.
> >
> > If there are any remaining concerns, we would greatly appreciate it if you could share them with us, so we may have enough time to provide a comprehensive response. Thank you once again for your time and valuable feedback!
> >
> > Thanks,
> >
> > Authors

---

### Official Review · Reviewer_APLY · 2025-07-21

**Clarity:** 3
**Significance:** 3
**Originality:** 3
**Rating:** 5
**Confidence:** 2

**Summary:**

The paper introduces SUBSAMPLE-MFQ, a novel algorithm designed for cooperative Multi-Agent Reinforcement Learning (MARL). By subsampling $k \leq n$ agents and applying mean-field Q-learning to this reduced subset, the method achieves significant computational savings. The algorithm retains theoretical guarantees, showing that the learned policy converges to the optimal one with an optimality gap of $\tilde{O}(1/\sqrt{k})$, independent of the total number of agents $n$. When $k = O(\log n)$, SUBSAMPLE-MFQ offers exponential improvements over previous methods.

**Questions:**

* I wonder the feasibility of design a small experiment to validate the conclusion in this paper?

**Ethical Concerns:**

["NO or VERY MINOR ethics concerns only"]

**Limitations:**

The authors mentioned the work's limitation in the paper.

**Quality:**

3

**Strengths And Weaknesses:**

Strength:
1. The paper provided rigorous theoretical proof for the algorithm
2. The paper showed the given algorithm in clear details.

Weakness:
* Although this paper focuses more on the theory side, no experiments was given.

---

> ### Author Rebuttal · Authors · 2025-07-29
>
> We thank reviewer APLY for their feedback on our work.
>
> We did provide two experiments in Appendix B (page 16) corresponding to the motivating examples (Gaussian squeeze and constrained exploration) in Lines 126-145, to numerically validate the theoretical advantages of our SUBSAMPLE-MFQ algorithm. Across these experiments, we also provided figures that plot the log reward optimality gap as a function of $k$, the time taken for SUBSAMPLE-MFQ to compute its learned policy for varying $k$, and the cumulative discounted reward seen by the algorithm for different $k$.
>
> We could plan to use the extra page in the camera-ready version to move some of the experiments to the main body. We hope this eases any concerns about the experimental validity of the algorithm, and we would be happy to answer any further questions.
>
> Thank you,
>
> Authors

---

> > ### Author Response · Authors · 2025-08-06
> > **Follow-up on Author Response**
> >
> > Dear Reviewer APLY,
> >
> > We hope this message finds you well.
> >
> > As the author-reviewer discussion period is coming to an end, we would like to kindly ask whether we have adequately addressed your concerns. If our responses have adequately addressed your concerns, we kindly ask you to consider updating your score.
> >
> > If there are any remaining concerns, we would greatly appreciate it if you could share them with us, so we may have enough time to provide a comprehensive response. Thank you once again for your time and valuable feedback!
> >
> > Thanks,
> >
> > Authors

---

### Note · Authors · 2025-08-14

**Dear Reviewers, AC, and SAC,**

We sincerely thank you for your thoughtful reviews, and appreciate your suggestions for enhancing the impact of our work for the larger community. We are encouraged that _all_ the reviewers appreciated our theoretical contributions and the significance of our error bound being independent of the number of agents, $n$. We are also pleased with the reviewers’ positive comments on the clarity of our exposition.

We have addressed some specific concerns in individual rebuttals, which we summarize below:

- **Optimality of error bound** (gLs2): we have _strong evidence that the optimality gap of the learned policy of $\tilde{O}(1/\sqrt{k})$ is tight_: for instance, a hurdle to uniformly improving our bound is a smaller error in standard mean-field MARL, which is known to be tight.

- **Agent Heterogeneity** (1oj9): when the agent types are numerous or drawn from a continuous space, we believe that recent methods in graphon mean-field MARL are amenable to similar subsampling guarantees. We have committed to _adding a detailed discussion of this in Remark 4.7_.

- **Experiments** (GLs2 and PDRU): we only tested our algorithms on relatively synthetic benchmarks (of Gaussian squeeze and constrained exploration) so as to direct focus on the theoretical contributions (as it is a submission to the NeurIPS theory track, after all), while reflecting the style of prior theory works in the field [1, 2]. Indeed, this leaves open the task of a more detailed empirical study of SUBSAMPLE-MFQ for future work.

- **Online execution communication constraints** (GLs2 and PDRU): In large distributed systems, the random sampling in our communication network may be a bottleneck for ``decentralized execution’’. In light of this, we have _provided a practical derandomized heuristic where the agents can share some randomness by only sampling within pre-defined fixed blocks of size $k$_. We will add this to the manuscript. Moreover, our subsampling framework is a **relaxation of the vanilla centralized $Q$-learning setting which makes a much stronger assumption** that each agent has full observability in the communication network.

We hope that our responses have clarified any remaining concerns.

Thanks,

_Authors_

References:

1) Mean Field Multi-Agent Reinforcement Learning. Yang et. al. ICML, 2018

2) Mean-field multiagent reinforcement learning: A decentralized network approach. Gu et. al. Mathematics of Operations Research, 2021

---

### Decision · Program_Chairs · 2025-09-17

**Decision:**

Accept (spotlight)

**Comment:**

The paper “develop[s] and analyze[s] a scalable algorithm for multi-agent RL by sampling the mean-field distribution of the agents to overcome the curse of dimensionality.”  All reviewers rated the paper highly, universally stating that it is a good theory paper that delivers on its claims.  Some concerns were expressed about the limited experimental results.